# IGL-Bench: Establishing the Comprehensive Benchmark for Imbalanced Graph Learning

**Jiawen Qin**[1]*, **Haonan Yuan**[1]*, **Qingyun Sun**[1]*, **Lyujin Xu**[1], **Jiaqi Yuan**[1], **Pengfeng Huang**[1],
**Zhaonan Wang**[1], **Xingcheng Fu**[2], **Hao Peng**[1], **Jianxin Li**[1]†, **Philip S. Yu**[3]
[1]Beihang Univerisity    [2]Guangxi Normal University    [3]University of Illinois, Chicago
`{qinjw,yuanhn,sunqy,lijx}@buaa.edu.cn`

## Abstract

Deep graph learning has gained grand popularity over the past years due to its versatility and success in representing graph data across a wide range of domains. However, the pervasive issue of imbalanced graph data distributions, where certain parts exhibit disproportionally abundant data while others remain sparse, undermines the efficacy of conventional graph learning algorithms, leading to biased outcomes. To address this challenge, Imbalanced Graph Learning (IGL) has garnered substantial attention, enabling more balanced data distributions and better task performance. Despite the proliferation of IGL algorithms, the absence of consistent experimental protocols and fair performance comparisons pose a significant barrier to comprehending advancements in this field. To bridge this gap, we introduce `IGL-Bench`, a foundational comprehensive benchmark for imbalanced graph learning, embarking on **17** diverse graph datasets and **24** distinct IGL algorithms with uniform data processing and splitting strategies. Specifically, `IGL-Bench` systematically investigates state-of-the-art IGL algorithms in terms of *effectiveness*, *robustness*, and *efficiency* on node-level and graph-level tasks, with the scope of *class-imbalance* and *topology-imbalance*. Extensive experiments demonstrate the potential benefits of IGL algorithms on various imbalanced conditions, offering insights and opportunities in the IGL field. Further, we have developed an open-sourced and unified package to facilitate reproducible evaluation and inspire further innovative research, available at: `https://github.com/RingBDStack/IGL-Bench`.

## 1 Introduction

Graphs are widely acknowledged as powerful for representing networks such as social networks (Fan et al., 2019), citation networks (Sun et al., 2021; Li et al., 2023a; Sun et al., 2022a), e-commerce networks (Li et al., 2020; Yuan et al., 2023), *etc.* In graphs, nodes represent individual entities, and edges signify relationships between nodes. Graph representation learning seeks to embed the graph (nodes, edges, or entire graphs) into a low-dimensional space while retaining their structural semantics (Zhang et al., 2020). Graph Neural Networks (GNNs) (Kipf & Welling, 2016; Hamilton et al., 2017; Velickovic et al., 2018) have emerged as the dominant approach for graph representation learning owing to their exceptional ability to leverage both the graph topology and node properties.

Though GNNs achieve satisfying performance in various tasks, they are typically designed assuming that training data is comprehensive and balanced. However, real-world graph data often feature imbalanced distributions with some parts possessing abundant data while others are scarce (Qin et al., 2024), which greatly compromises task performance. The non-Euclidean nature of graph data precludes the use of traditional imbalance learning algorithms, presenting a considerable obstacle to the deployment of GNNs in real-world scenarios, which is also a heated research topic in the community. As graph learning enters the new era of large models, there are a number of graph foundation models (Liu et al., 2023a) that depend on a wide range of graph data for pre-training, enabling them to obtain base models that generalize across diverse domains and tasks. Unexpectedly, massive imbalanced graph data inevitably introduces intrinsic biases, presenting significant challenges for subsequent prompt-based fine-tuning for downstream applications.

---

*Equal contribution.
†Corresponding author.

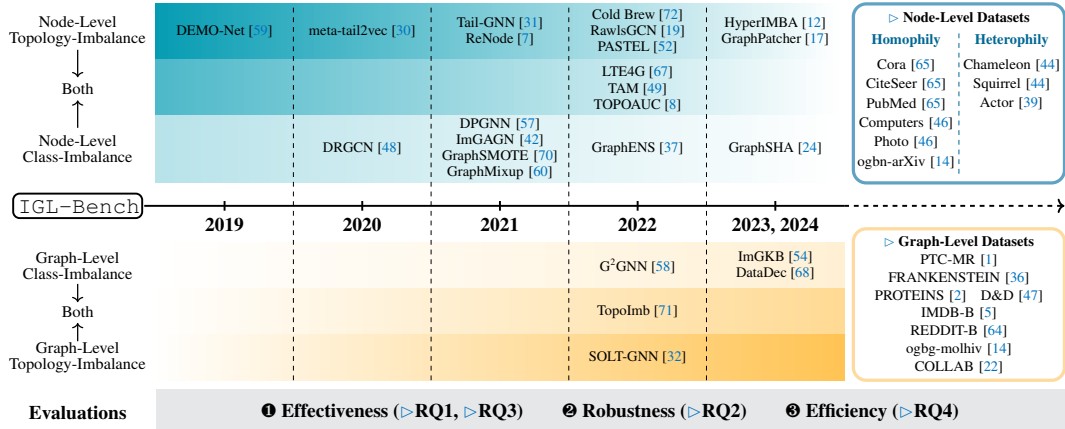

Figure 1: Overview of the established `IGL-Bench`. Both IGL algorithms and datasets are categorized into node-level and graph-level, where the algorithms are further divided into class-imbalance, topology-imbalance, or both. Click ▷ and link to the corresponding sections for in-depth analysis.

**Imbalanced Graph Learning (IGL).** To address the challenge of imbalance, a wide range of methods have been proposed in the realms of computer vision (Huang et al., 2016) and language (Li et al., 2019) on the broadly concerning *class-imbalance* learning issue (Johnson & Khoshgoftaar, 2019). Nevertheless, the non-Euclidean graph data presents distinct challenges due to its inherent non-*i.i.d.* and diverse topological nature. As a result, not only do these conventional methods become infeasible for graphs, but they also fail to address other distinctive *topology-imbalance* challenges intrinsic in graph data (Zhao et al., 2021; Liu et al., 2021; Chen et al., 2021; Li et al., 2024; 2025). To mitigate the aforementioned imbalanced issues, Imbalanced Graph Learning (IGL) has recently attracted considerable research interest, as highlighted in Figure 1. The increasing literature each year reflects the rising significance and profound effect of tackling IGL challenges, which are categorized into various kinds of research problems, presenting distinct characteristics, necessitating the creation of specialized techniques to handle the imbalance issues inherent to each scenario effectively.

Despite the emerging studies of IGL algorithms, there lacks a comprehensive and unified benchmark, which would significantly impede the understanding and progress of IGL for the following aspects. ❶ **Dataset preparation rule.** The use of different datasets, data processing approaches, and imbalanced data-splitting strategies in previous works makes many of the results incomparable. ❷ **Experiment conduction protocol.** The variability in experimental setups, including parameter settings, initialization procedures, and convergence criteria, hinders reproducibility and comparability across studies. ❸ **Performance evaluation standard.** The metric for evaluating task performance is not consistent. Apart from effectiveness, understanding the efficiency and complexity of each algorithm is imperative, yet often overlooked in the literature. Hence, there is an urgent necessity within the community for the creation of a comprehensive and open-sourced benchmark for IGL.

In this work, we establish a comprehensive **I**mbalanced **G**raph **L**earning **Bench**mark (`IGL-Bench`), which serves as the first open-sourced and unified benchmark for graph-specific imbalanced learning to the best of our knowledge. Through benchmarking existing IGL algorithms for **effectiveness**, **robustness**, and **efficiency**, we make the following contributions:

- **First Comprehensive IGL Benchmark.** `IGL-Bench` enables a fair and unified comparison among **19** state-of-the-art node-level and **5** graph-level IGL algorithms by unifying the experimental settings across **17** graph datasets of diverse characteristics, providing a comprehensive understanding of the *class-imbalance* and *topology-imbalance* problems in IGL for the first time.

- **Multi-faceted Evaluation and Analysis.** We conduct a systematic analysis of IGL methods from various dimensions, including effectiveness, efficiency, and complexity. Based on the results of extensive experiments, we uncover both the potential advantages and limitations of current IGL algorithms, providing valuable insights to guide future research endeavors.

- **Open-sourced Package.** To facilitate future IGL research, we develop an open-sourced benchmark package for public access. Users can evaluate their algorithms or datasets with less effort.

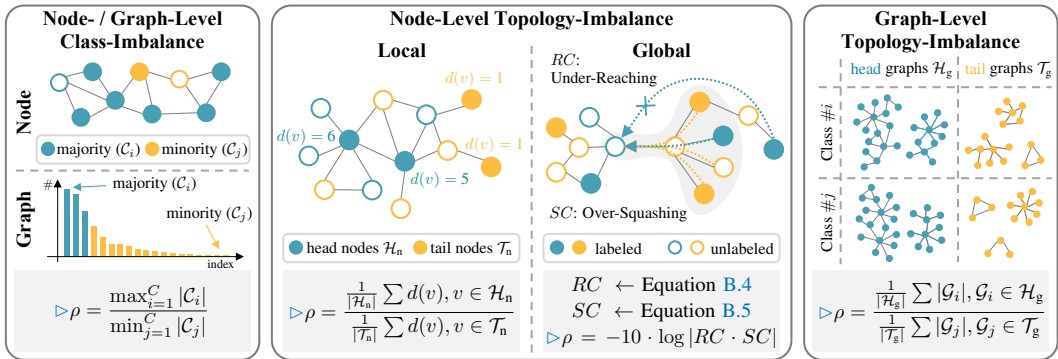

Figure 2: The research scope of the proposed `IGL-Bench`. Definitions of the imbalance ratio ($\rho$) corresponding to each imbalance issue are further concluded in Table 1. Click ▷ and check details.

## 2 PRELIMINARY AND PROBLEM FORMULATION

**Notations.** Let $\mathcal{G} = \{\mathcal{V}, \mathcal{E}, \mathbf{A}, \mathbf{X}\}$ be a graph, where $\mathcal{V}$ is the node set with $N$ nodes, $\mathcal{E}$ is the edge set, $\mathbf{A} \in \mathbb{R}^{N \times N}$ is the adjacency matrix, $\mathbf{X} \in \mathbb{R}^{N \times d}$ is the node feature matrix with $d$-dimension.

**Node-level Classification.** Given the labeled node set $\mathcal{V}_L$ and their labels $\mathbf{Y}_L \in \mathbb{R}^C$, where each node $v_i$ is associated with a label $\mathbf{y}_i$. Semi-supervised node classification aims to train a node classifier $f_{\boldsymbol{\theta}} : v \mapsto \mathbb{R}^C$ to predict the labels $\mathbf{Y}_U$ of the remaining nodes $\mathcal{V}_U = \mathcal{V} \setminus \mathcal{V}_L$.

**Graph-level Classification.** Denote $\mathbf{G}$ as the graph set. Given the labeled graph set $\mathbf{G}_L$ and their labels $\mathbf{Y}_L \in \mathbb{R}^C$, where each graph $\mathcal{G}_i$ is associated with a label $\mathbf{y}_i$. Graph classification task aims to train a graph classifier $\mathcal{F}_{\boldsymbol{\theta}} : \mathcal{G} \mapsto \mathbb{R}^C$ to predict the labels $\mathbf{Y}_U$ of the unlabeled graphs $\mathbf{G}_U = \mathbf{G} \setminus \mathbf{G}_L$.

We formulate the IGL problems into two categories: ***class-imbalance*** and ***topology-imbalance***, where the detailed categorizing motivations and problem descriptions can be found in Appendix B.1.

**Definition 1** (**Class-Imbalance**). There exists an imbalance in the number of labeled samples (nodes or graphs) across different classes, leading to the long-tailed quantity distribution (Ma et al., 2023).

**Definition 2** (**Topology-Imbalance**). Both node- and graph-level tasks encounter topology-imbalance. **For node-level tasks**, an imbalance exists in the topological distribution of labeled nodes, which is brought by two main aspects: ❶ **Local.** Imbalanced node degree distribution (Wu et al., 2019). ❷ **Global.** Imbalanced graph structures concerning the Under-reaching and Over-squashing phenomenon (Sun et al., 2022b). **For graph-level tasks**, the imbalance is facilitated by the uneven graph size (the number of nodes) distribution (Liu et al., 2022), which offers potentially biased structures.

## 3 IGL-BENCH: IMBALANCED GRAPH LEARNING BENCHMARK

In this section, we introduce the overview of the `IGL-Bench` with considerations of the datasets (Section 3.1), algorithms (Section 3.2), and the research questions that guide the benchmark study (Section 3.3). We provide additional details including further declarations in the **Appendix**.

### 3.1 BENCHMARK DATASETS

To comprehensively and effectively evaluate the performance of IGL algorithms, we have integrated **17** real-world datasets from various domains for both the node-level and graph-level tasks. We briefly introduce each category in the following sections. More details are provided in Appendix A.1.

**Node-level Classification Datasets.** We utilize **9** graph datasets covering different data scales and homophily, including three citation networks from Plantoid (Yang et al., 2016) (Cora, CiteSeer, PubMed), two co-occurrence networks in Amazon (Shchur et al., 2018) (Computers, Photo), the large-scale ogbn-arXiv (Hu et al., 2020), two page-page networks in Wikipedia (Rozemberczki et al., 2021) (Chameleon, Squirrel), and an actor-only induced subgraph of the film-director-actor-writer network Actor (Pei et al., 2019). Datasets range from strong homophily to strong heterophily.

Table 1: Definitions of the imbalance ratio ($\rho$) across different imbalance types.

| Imbalance Type | Definition | Explanation |
|---|---|---|
| Node-Level Class-Imbalance Graph-Level Class-Imbalance | $\rho = \dfrac{\max_{i=1}^{C} |\mathcal{C}_i|}{\min_{j=1}^{C} |\mathcal{C}_j|}$ | The imbalance ratio is set to the ratio between the number of samples ($|\mathcal{C}|$) in the majority and the minority class. |
| Node-Level Topology-Imbalance (local and global) | $\rho = \dfrac{\frac{1}{|\mathcal{H}_\mathrm{n}|} \sum d(v), v \in \mathcal{H}_\mathrm{n}}{\frac{1}{|\mathcal{T}_\mathrm{n}|} \sum d(v), v \in \mathcal{T}_\mathrm{n}}$ | The **local** imbalance ratio is set to the ratio of the average node degree ($d(v)$) of the head node set ($\mathcal{H}_\mathrm{n}$) to the average node degree of the tail node set ($\mathcal{T}_\mathrm{n}$). |
| | $\rho = -10 \cdot \log|RC \cdot SC|$ | The **global** imbalance ratio is set to the negative logarithm of the absolute value of the product of the Reaching Coefficient ($RC$) and the Squashing Coefficient ($SC$). |
| Graph-Level Topology-Imbalance | $\rho = \dfrac{\frac{1}{|\mathcal{H}_\mathrm{g}|} \sum |\mathcal{G}_i|, \mathcal{G}_i \in \mathcal{H}_\mathrm{g}}{\frac{1}{|\mathcal{T}_\mathrm{g}|} \sum |\mathcal{G}_j|, \mathcal{G}_j \in \mathcal{T}_\mathrm{g}}$ | The imbalance ratio is set to the ratio of the average graph size (number of nodes) of the head graph set ($\mathcal{H}_\mathrm{g}$) to the average graph size of the tail graph set ($\mathcal{T}_\mathrm{g}$). |

**Graph-level Classification Datasets.** We integrate **8** widely adopted real-world datasets. PTC-MR (Bai et al., 2019) and FRANKENSTEIN (Orsini et al., 2015) are molecule datasets, where each graph is a molecule with or without mutagenicity. PROTEINS (Dobson & Doig, 2003; Borgwardt et al., 2005) and D&D (Dobson & Doig, 2003; Shervashidze et al., 2011) are protein datasets marked as enzyme or non-enzyme. IMDB-B (Cai & Wang, 2018) and REDDIT-B (Yanardag & Vishwanathan, 2015) are social networks in movies and online discussions, respectively. The large-scale ogb-molhiv (Hu et al., 2020) is a benchmark dataset for predicting the biological activity of molecules, featuring various molecular structures represented as graphs along with corresponding labels. The scientific collaboration dataset COLLAB (Leskovec et al., 2005) for multi-class classification is derived from three publicly available collaboration datasets that represent distinct research fields.

## 3.2 BENCHMARK ALGORITHMS

Table A.3 conclude the overall **24** IGL algorithms integrated in `IGL-Bench` with their technique categorization, complexity analysis, and links to implementations (Details in Appendix A.2).

**Class-Imbalanced IGL Algorithms.** Node-level class-imbalanced IGL refers to the uneven allocation of labeled nodes among classes. The classifier prioritizes learning from classes abundant in labeled instances, potentially neglecting those with fewer instances. We implement 10 representative algorithms including DRGCN (Shi et al., 2020), DPGNN (Wang et al., 2021), ImGAGN (Qu et al., 2021), GraphSMOTE (Zhao et al., 2021), GraphENS (Park et al., 2021), GraphMixup (Wu et al., 2022), LTE4G (Yun et al., 2022), TAM (Song et al., 2022), TOPOAUC (Chen et al., 2022) and GraphSHA (Li et al., 2023b). Graph-level class-imbalanced IGL manifests in practical situations where the distribution of labeled graphs across classes is skewed, typically favoring the majority class with more labeled graphs. We select 4 typical algorithms including G$^2$GNN (Wang et al., 2022), TopoImb (Zhao et al., 2022), DataDec (Zhang et al., 2023), and ImGKB (Tang & Liang, 2023).

**Topology-Imbalanced IGL Algorithms.** Node-level topology-imbalanced IGL occurs when the node topology properties display an unequal distribution. An important metric is the node degree, which can reflect the proximity richness. We incorporate DEMO-Net (Wu et al., 2019), meta-tail2vec (Liu et al., 2020), Tail-GNN (Liu et al., 2021), Cold Brew (Zheng et al., 2021), LTE4G (Yun et al., 2022), RawlsGCN (Kang et al., 2022), and GraphPatcher (Ju et al., 2024a). Another profound topology imbalance is brought by the under-reaching and over-squashing problem (Sun et al., 2022b), which critically influences the label propagation process. We take ReNode (Chen et al., 2021), TAM (Song et al., 2022), PASTEL (Sun et al., 2022b), TOPOAUC (Chen et al., 2022), and HyperIMBA (Fu et al., 2023) as our investigation scope. Graph-level topology-imbalanced IGL stems from the intricate interconnections within graphs. This imbalance frequently presents as variations in graph sizes and topology groups. We implement SOLT-GNN (Liu et al., 2022) and TopoImb (Zhao et al., 2022).

## 3.3 RESEARCH QUESTIONS

We systematically design the `IGL-Bench` to comprehensively evaluate the existing IGL algorithms and inspire future research. In particular, we aim to investigate the following research questions.

**RQ1: How much progress has been made by the existing IGL algorithms?**

**Motivation and Experiment Design.** Existing IGL algorithms are conducted under inconsistent imbalance settings, making it unfair to compare the task performance. Given the fair data and experiment environment by `IGL-Bench`, **RQ1** aims to gain a deeper understanding of the strengths and weaknesses of IGL algorithms and identify directions that offer avenues for prospective improvements. To achieve this, we conduct node and graph classifications, where the train/val/test split satisfies the consistent ratio of 1:1:8. We facilitate dataset imbalance with the imbalance ratio $\rho$ follows definitions in Tabel 1, providing a fair comparison under the same imbalance degree. To perform an unbiased evaluation, we summarize all the metrics used in the original papers of the algorithms (Table C.1) and report results with Accuracy (Acc.), Balanced Accuracy (bAcc.), Macro-F1 (M-F1), and AUC-ROC. The consensus and focus for each metric are analyzed in Appendix C.2.

**RQ2: How effective are the IGL algorithms generalizing to the changing imbalance ratio?**

**Motivation and Experiment Design.** Since **RQ1** has already investigated the performance of IGL algorithms on datasets of certain fixed imbalance ratios, **RQ2** further explores the robustness of IGL algorithms as the degree of imbalance varies by quantitatively controlling the imbalance ratio of each dataset to study the diverse capabilities of IGL algorithms. To achieve this, we quantitatively set the imbalance ratio to exhibit a staggered distribution of imbalance levels from (relatively) balanced to extremely imbalanced under the predefined splitting constraints.

**RQ3: Does classifiers benefit from the IGL algorithms to learn clearer boundaries?**

**Motivation and Experiment Design.** Imbalanced data can cause unexpected shifts of classifier boundary, negatively impacting task performance. **RQ3** aims to investigate whether the performance improvement in downstream tasks results from clearer classification boundaries under the influence of the IGL algorithms. To achieve this, we compare changes in inter-class clustering coefficients by the Silhouette score (Rousseeuw, 1987). Additionally, we use t-SNE (Van der Maaten & Hinton, 2008) to visualize the learned embeddings, aiding in intuitively understanding boundary shifts.

**RQ4: How efficient are these IGL algorithms in terms of time and space?**

**Motivation and Experiment Design.** Existing IGL algorithms handle the imbalance issues generally by redistributing data at either the data level or algorithm level to achieve balance, a process that naturally incurs extra computational and spatial complexity compared to vanilla GNNs. However, the algorithm efficiency has been largely overlooked, where **RQ4** is proposed to understand the trade-off between efficiency and task performance. To achieve this, we evaluate the algorithm efficiency by reporting the training time and peak GPU memory consumption on consistent configurations.

## 4    EXPERIMENT RESULTS AND ANALYSIS

In this section, we compare IGL algorithms covering node-level and graph-level tasks, addressing *class-imbalance* and *topology-imbalance* issues. Detailed experiment settings and additional results on more metrics and backbones can be found in Appendix B, Appendix C, and Appendix D.

### 4.1    EFFECTIVENESS EVALUATIONS FOR IGL ALGORITHMS (RQ1)

#### 4.1.1    EFFECTIVENESS OF NODE-LEVEL CLASS-IMBALANCED ALGORITHMS

**Results** (Table 2). ❶ All algorithms surpass GCN on at least 5 datasets, showing a smaller performance gain on heterophilic graph datasets compared to homophilic ones. ❷ Compared to the resampling algorithms (*e.g.*, ImGAGN, GrapSMOTE, GraphENS, and GraphSHA), data-augmentation algorithms (*e.g.*, LTE4G and GraphMixup) achieve better performance on 6 out of 9 datasets. ❸ The loss-engineered algorithm TOPOAUC achieves optimal or near-optimal results in 5 out of 7 datasets, attributed to its tailored modules for handling class-imbalanced and global topology-imbalanced data. ❹ Over half of the algorithms fail on the large-scale ogbn-arXiv, while the others perform worse.

#### 4.1.2    EFFECTIVENESS OF NODE-LEVEL LOCAL TOPOLOGY-IMBALANCED ALGORITHMS

**Results** (Table 3). ❶ Most algorithms outperform GCN on 7 datasets, with DEMO-Net and Graph-Patcher surpassing GCN on all datasets. ❷ Neighbor-augmented algorithms (*e.g.*, Tail-GNN, Cold Brew, and GraphPatcher) achieve greater performance gains compared to model-modified algorithms (*e.g.*, DEMO-Net and RawlsGCN). ❸ Tail-GNN and GraphPatcher excel on high-homophily datasets, whereas Cold Brew performs better on high-heterophily ones. ❹ Cases on ogbn-arXiv are worse.

Table 2: **Accuracy** score (% ± standard deviation) of **node** classification on manipulated **class-imbalanced** graph datasets (**Low**) over 10 runs. "—" denotes out of memory or time limit. The best results are shown in **bold** and the runner-ups are underlined (the same for tables below).

| Algorithm | Cora 0.81 | CiteSeer 0.74 | PubMed 0.80 | Computers 0.78 | Photo 0.82 | ogbn-arXiv 0.65 | Chameleon 0.23 | Squirrel 0.22 | Actor 0.22 |
|---|---|---|---|---|---|---|---|---|---|
| GCN (bb.) [21] | 76.36±0.13 | 52.96±0.55 | 60.57±0.19 | 75.06±0.50 | 69.80±6.15 | 59.83±0.23 | 26.35±0.24 | 17.16±0.17 | 24.06±0.14 |
| DRGCN [48] | 71.35±0.77 | 55.22±1.82 | 62.59±4.62 | 67.71±3.10 | 85.67±5.30 | — | 26.40±0.35 | 17.11±0.81 | 25.03±0.23 |
| DPGNN [57] | 72.91±3.95 | 56.78±2.23 | 81.87±2.80 | 68.69±8.62 | 81.66±9.19 | — | 30.58±1.48 | **25.35±1.48** | 21.66±1.68 |
| ImGAGN [42] | 73.48±3.07 | 55.29±3.00 | 72.16±1.51 | 74.92±1.87 | 83.10±3.42 | — | 24.38±2.86 | 18.75±1.80 | 24.54±3.38 |
| GraphSMOTE [70] | 77.21±0.27 | 53.55±0.95 | 71.25±0.27 | 70.54±1.52 | 89.07±1.12 | — | 27.23±0.21 | 16.79±0.14 | 25.08±0.31 |
| GraphENS [37] | 79.34±0.49 | 61.98±0.76 | 80.84±0.17 | 80.72±0.68 | 90.38±0.37 | 53.23±0.52 | 24.34±1.62 | 20.05±1.61 | 25.03±0.38 |
| GraphMixup [60] | 79.88±0.43 | 62.66±0.70 | 75.94±0.09 | **86.15±0.47** | 89.69±0.31 | 56.08±0.31 | 30.95±0.40 | 17.83±0.32 | 24.75±0.37 |
| LTE4G [67] | 80.53±0.65 | 64.48±1.56 | **83.02±0.33** | 79.35±1.39 | 87.94±1.82 | — | 31.91±0.34 | 19.37±0.41 | **25.43±0.26** |
| TAM [49] | 80.69±0.27 | 64.16±0.24 | 81.47±0.15 | 81.30±0.53 | 90.35±0.42 | 53.49±0.54 | 23.27±1.38 | 21.17±0.95 | 24.53±0.33 |
| TOPOAUC [8] | **83.34±0.31** | **69.03±1.33** | — | 70.85±4.55 | 83.72±2.23 | | **33.60±1.51** | 21.38±1.03 | 25.16±0.46 |
| GraphSHA [24] | 80.03±0.46 | 60.51±0.61 | 77.94±0.36 | 82.71±0.40 | **91.55±0.32** | **60.30±0.13** | 23.73±1.97 | 20.05±1.61 | 23.59±1.01 |

Table 3: **Accuracy** score (% ± standard deviation) of **node** classification on manipulated **local topology-imbalanced** graph datasets (**Mid**) over 10 runs. "—" denotes out of memory or time limit.

| Algorithm | Cora 0.81 | CiteSeer 0.74 | PubMed 0.80 | Computers 0.78 | Photo 0.82 | ogbn-arXiv 0.65 | Chameleon 0.23 | Squirrel 0.22 | Actor 0.22 |
|---|---|---|---|---|---|---|---|---|---|
| GCN (bb.) [21] | 80.16±1.09 | 66.87±0.85 | 83.97±0.13 | 71.65±2.10 | 89.43±0.58 | 52.93±0.33 | 52.74±0.60 | 28.70±0.68 | 21.55±1.74 |
| DEMO-Net [59] | 80.37±0.52 | 69.73±1.31 | 84.11±0.20 | 79.38±0.98 | 88.09±1.30 | 65.81±0.11 | 55.51±0.87 | 39.45±0.62 | 29.12±0.30 |
| meta-tail2vec [30] | 32.17±0.68 | 29.97±3.61 | 59.82±2.86 | 68.17±1.07 | 79.82±1.02 | 33.71±1.16 | 38.78±0.44 | 24.90±0.25 | 26.09±0.07 |
| Tail-GNN [31] | 79.05±1.15 | 69.97±1.03 | 85.78±0.41 | **84.09±1.01** | 92.21±0.09 | — | 53.20±0.80 | 30.43±1.06 | 28.02±0.71 |
| Cold Brew [72] | 73.84±2.10 | 67.42±0.97 | **86.51±0.04** | 80.19±0.24 | 88.13±0.24 | **69.97±0.07** | **59.16±0.40** | **43.04±0.24** | **33.01±0.19** |
| LTE4G [67] | 82.54±0.46 | 70.55±0.54 | 84.77±0.78 | 81.32±2.21 | 91.09±0.19 | — | 55.84±2.86 | 32.43±3.31 | 24.00±0.49 |
| RawlsGCN [19] | 80.52±0.14 | 72.38±0.43 | 86.05±0.12 | 78.78±1.40 | 90.53±1.32 | 40.00±0.05 | 44.96±0.79 | 29.93±0.65 | 28.29±0.24 |
| GraphPatcher [17] | **83.25±0.42** | **73.38±0.42** | 85.60±0.16 | 83.68±0.69 | **92.28±0.06** | 66.74±0.04 | 55.19±0.41 | 36.94±0.11 | 23.85±0.92 |

Table 4: **Accuracy** score (% ± standard deviation) of **node** classification on manipulated **global topology-imbalanced** graph datasets (**High**) over 10 runs.

| Algorithm | Cora 0.81 | CiteSeer 0.74 | PubMed 0.80 | Computers 0.78 | Photo 0.82 | ogbn-arXiv 0.65 | Chameleon 0.23 | Squirrel 0.22 | Actor 0.22 |
|---|---|---|---|---|---|---|---|---|---|
| GCN (bb.) [21] | 79.10±1.28 | 68.37±1.73 | 83.44±0.16 | 75.02±2.20 | 86.32±1.90 | 51.04±0.18 | 33.90±0.70 | 23.27±0.82 | 22.40±0.68 |
| ReNode [7] | 79.91±1.52 | 69.89±0.73 | 82.97±0.12 | 77.95±1.71 | 87.80±0.52 | 50.68±0.15 | 32.92±0.98 | 23.80±0.59 | 22.39±0.62 |
| TAM [49] | 80.50±0.18 | **73.14±0.13** | 84.07±0.12 | 82.35±0.19 | 89.80±0.23 | **52.09±0.06** | 35.64±0.27 | 24.58±0.09 | 22.55±0.06 |
| PASTEL [52] | **80.91±0.36** | 72.73±0.26 | — | 83.24±0.85 | 89.10±0.41 | — | **47.12±2.82** | **33.15±0.66** | **27.56±1.04** |
| TOPOAUC [8] | 79.27±0.52 | 70.08±0.83 | — | 75.35±1.32 | 87.10±0.98 | — | 33.39±2.09 | 22.86±0.36 | 22.56±0.18 |
| HyperIMBA [12] | 79.81±0.78 | 71.78±0.40 | **84.75±0.30** | **83.43±0.65** | **90.65±0.14** | — | 38.30±2.70 | 29.97±1.79 | 25.30±2.56 |

### 4.1.3 Effectiveness of Node-Level Global Topology-Imbalanced Algorithms

**Results** (Table 4). ❶ Re-weighting IGL algorithms (*e.g.*, ReNode, TAM, and HyperIMBA) generally outperform vanilla GCN on highly homophilic datasets but struggle on heterophilic ones. ❷ Structure-refined PASTEL achieves optimal or near-optimal results on most datasets, showing significant improvements on highly heterophilic datasets due to its alleviation of both under-reaching and over-squashing phenomena. However, the structure learning mechanism introduces a heavy quadratic computational burden, making PASTEL challenging to adapt to large-scale graphs, *e.g.*, ogbn-arXiv. ❸ Though algorithms with sub-quadratic complexity (ReNode and TAM) are available on ogbn-arXiv, their performance is greatly weakened due to low-efficiency representation learning and imbalance debias. ❹ TOPOAUC has limited ability to address the global topology-imbalance problem and even performs worse than GCN on heterophilic datasets, which is caused by its homophily assumption.

### 4.1.4 Effectiveness of Graph-Level Class-Imbalanced Algorithms

**Results** (Table 5). ❶ DataDec achieves optimal or near-optimal results on all datasets. It identifies an informative subset for model training via dynamic sparse graph contrastive learning, which leverages abundant of unlabeled information to enhance the performance. ❷ G$^2$GNN generally outperforms GIN on binary classification datasets but fails to surpass GIN on multi-classification datasets. ❸ TopoImb and ImGKB show considerable instability across different datasets in class-imbalanced settings. Despite meticulous hyperparameter tuning detailed in Appendix C to ensure thorough and impartial evaluations, TopoImb cannot be consistently trained to outperform the backbones due to its sensitivity to dataset-specific characteristics. ❹ Half algorithms fail on the large-scale ogbg-molhiv.

Table 5: **Accuracy** score (% ± standard deviation) of **graph** classification on manipulated **class-imbalanced** graph datasets (**Low**) over 10 runs.

| Algorithm | PTC-MR | FRANKENSTEIN | PROTEINS | D&D | IMDB-B | REDDIT-B | ogbg-molhiv | COLLAB |
|---|---|---|---|---|---|---|---|---|
| GIN (bb.) [63] | 47.83±2.95 | 63.38±1.93 | 55.38±3.57 | 51.05±5.07 | 62.31±3.99 | 61.10±4.86 | 60.75±3.79 | 65.01±1.33 |
| G²GNN [58] | 51.88±6.23 | 61.13±1.05 | 63.61±5.03 | 56.29±7.30 | 63.87±4.64 | 69.58±3.59 | **65.00**±3.81 | 62.05±3.06 |
| TopoImb [71] | 44.86±3.52 | 49.49±7.14 | 52.12±10.51 | 49.97±7.24 | 59.95±5.19 | 59.67±7.30 | — | 65.88±0.75 |
| DataDec [68] | **55.72**±2.88 | **67.99**±0.75 | 66.58±1.35 | 63.51±1.62 | **67.92**±3.37 | **78.39**±5.01 | — | **71.48**±1.03 |
| ImGKB [54] | 50.11±5.95 | 40.83±0.02 | **66.60**±2.64 | **65.85**±3.70 | 47.74±0.29 | 67.50±2.70 | 48.57±2.14 | 51.21±0.10 |

Table 6: **Accuracy** score (% ± standard deviation) of **graph** classification on manipulated **topology-imbalanced** graph datasets (**Mid**) over 10 runs.

| Algorithm | PTC-MR | FRANKENSTEIN | PROTEINS | D&D | IMDB-B | REDDIT-B | ogbg-molhiv | COLLAB |
|---|---|---|---|---|---|---|---|---|
| GIN (bb.) [63] | 51.38±6.78 | 54.82±2.26 | 62.14±2.43 | 61.46±2.43 | 65.08±5.78 | 68.32±1.77 | 57.67±3.12 | 65.84±3.12 |
| SOLT-GNN [32] | **53.04**±3.91 | **68.71**±1.60 | **71.95**±2.36 | 63.33±1.86 | **69.38**±1.23 | **73.51**±1.14 | — | **69.69**±2.45 |
| TopoImb [71] | 51.59±4.30 | 54.52±0.87 | 64.03±4.43 | **65.99**±1.25 | 68.10±0.87 | 71.54±0.75 | — | 68.68±1.34 |

### 4.1.5 EFFECTIVENESS OF GRAPH-LEVEL TOPOLOGY-IMBALANCED ALGORITHMS

**Results** (Table 6). ❶ SOLT-GNN surpasses GIN in 5 datasets by transferring head graphs' knowledge to augment tail graphs, showcasing the effectiveness of knowledge transfer mechanisms in improving imbalanced classification. ❷ Though TopoImb is proposed primarily to address uneven sub-structure distribution, results also demonstrate its ability to alleviate topology-imbalance problems across several datasets. ❸ Despite recent advancements, a significant performance gap persists between current graph-level IGL algorithms and their node-level counterparts. This observation underscores the need for continued research into more effective strategies to bridge this disparity.

> **Key Insights for RQ1:** Node-level class-imbalance and topology-imbalance often coexist, posing unique challenges that can be simultaneous and orthogonal. For node-level classification, the homophily or heterophily property of the dataset (*i.e.*, the neighbor's label distribution) significantly impacts the learning on class-imbalanced and topology-imbalanced graphs. Currently, there is a lack of effective algorithms that address both types of imbalance in large-scale graphs without relying on homophily assumptions, underscoring the need for more robust and adaptable solutions.

## 4.2 ROBUSTNESS TO DIFFERENT IMBALANCE RATIOS (RQ2)

In this section, we quantitatively set the imbalance ratios of each dataset defined in Table 1 to further investigate the robustness of IGL algorithms as the degree of imbalance varies.

### 4.2.1 ROBUSTNESS OF NODE-LEVEL CLASS-IMBALANCED ALGORITHMS

**Settings.** We manipulate datasets following settings in Appendix B.2 to exhibit a staggered imbalance ratio from $\rho = 1$ to 100 (denoted as **Balanced** to **High**). We compare the algorithms' performance changes along with their relative decrease. The single bar chart reflects the algorithm's effectiveness, a set of bar charts further illustrates the robustness, and the line chart depicts the algorithm's ability to control the performance degradation in an imbalanced data distribution (the flatter, the better).

**Results** (Figure 3). ❶ As the imbalance ratio increases, all node-level IGL algorithms encounter greater challenges, resulting in a gradual decline in performance. ❷ Among the class-imbalanced IGL algorithms, those based on resampling demonstrate better robustness compared to algorithms based on re-weighting and data augmentation. ❸ For extreme class imbalance (High, $\rho = 100$), class-imbalance-specific IGL algorithms generally exhibit higher robustness and performance compared to GCN and global topology imbalance methods. Additionally, algorithms designed for both class- and topology-imbalance (*e.g.*, TAM and TOPOAUC) further enhance performance.

### 4.2.2 ROBUSTNESS OF NODE-LEVEL LOCAL TOPOLOGY-IMBALANCED ALGORITHMS

**Settings.** We manipulate datasets for the node classification following settings in Appendix B.3 with the local topology-imbalance ratios from **Low** to **High**. For each dataset, we randomly select training nodes to facilitate different imbalance ratios while ensuring an equal number of nodes per class.

**Results** (Figure 4(a)). ❶ IGL algorithms demonstrate greater robustness in various imbalanced cases, showing more stable performance compared to GCN by maintaining consistent performance levels.

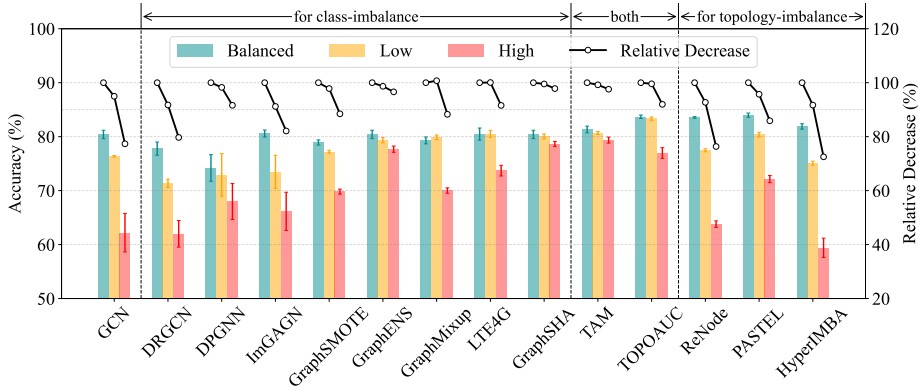

Figure 3: Robustness analysis of **node-level** algorithms under different **class-imbalance** levels on **Cora** (homophilic). Results are **Accuracy** and its relative decrease compared to the balanced split.

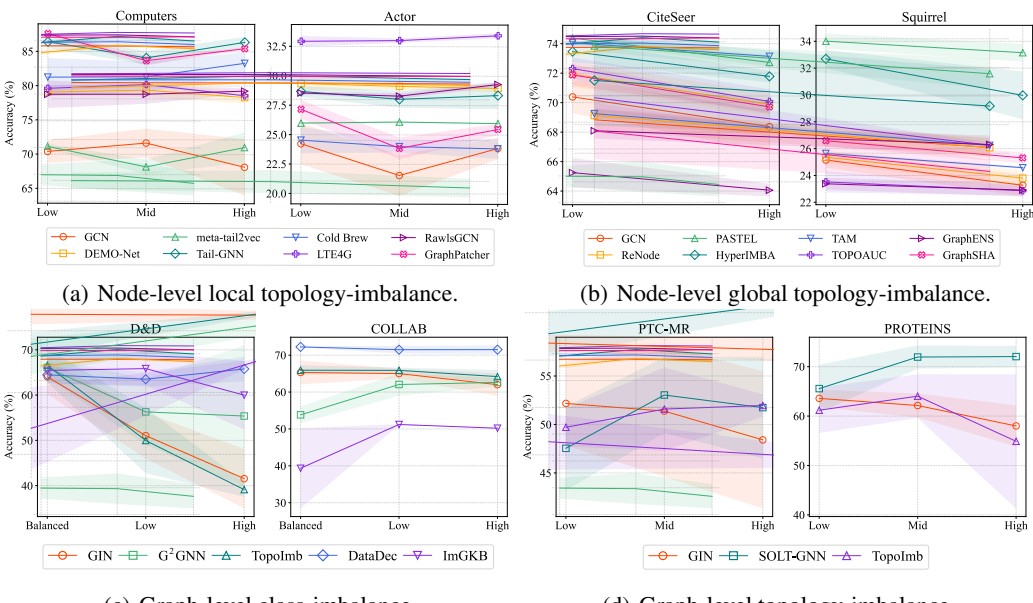

(a) Node-level local topology-imbalance.  (b) Node-level global topology-imbalance.

(c) Graph-level class-imbalance.  (d) Graph-level topology-imbalance.

Figure 4: Robustness analysis of the **node-level** and **graph-level** algorithms under different imbalance levels. Results are reported with the algorithm performance (**Accuracy**) with the standard deviation.

❷ Different algorithms display varying levels of robustness when facing different types of datasets. For example, neighbor-augmented algorithms are robust to extreme local topology-imbalance and they consistently boost performance in the homophilic dataset (*e.g.*, Computers) by a significant margin. Their advantages are even more prominent under higher topology-imbalance. However, they are relatively sensitive to different levels of imbalance in the heterophilic datasets (*e.g.*, Actor).

### 4.2.3 ROBUSTNESS OF NODE-LEVEL GLOBAL TOPOLOGY-IMBALANCED ALGORITHMS

**Settings.** We manipulate datasets for the node classification following settings in Appendix B.4 with different levels of the global topology-imbalance ratios from **Low** to **High**, concerning multiple degrees of the under-reaching and over-squashing phenomena to evaluate algorithm robustness.

**Results** (Figure 4(b)). ❶ All algorithms perform worse in highly imbalanced scenarios due to the difficulty in balancing the uneven topological distributions of training nodes. ❷ Topology-imbalanced IGL algorithms generally exhibit robustness across different imbalanced scenarios and tend to enhance performance on both homophilic and heterophilic datasets by utilizing structure learning to alleviate topological imbalance (*e.g.*, PASTEL and HyperIMBA). ❸ Class-imbalanced GraphSHA synthesizes nodes and connections with different labels, which promotes the global propagation of supervised signals and aids in addressing topological imbalance.

### 4.2.4 ROBUSTNESS OF GRAPH-LEVEL CLASS-IMBALANCED ALGORITHMS

**Settings.** Previous research emphasizes the impact of class-imbalance issues on the binary graph classification task. We manipulate datasets for both the binary and multi-class graph classification task following settings in Appendix B.5 with varying levels of class-imbalance to explore the robustness of IGL algorithms from **Balanced** ($\rho = 1$) to extremely imbalanced scenarios (**High**, $\rho = 100$).

**Results** (Figure 4(c)). ❶ IGL algorithms display varying degrees of robustness on different types of datasets. For example, with an increased imbalance ratio, the performance of IGL gradually decreases on binary classification datasets such as D&D. ❷ On the contrary, in the multi-class dataset COLLAB, IGL algorithms demonstrate strong robustness across varying levels of imbalance. This indicates that these algorithms can maintain their performance on imbalanced data, effectively handling the complexity and diversity of multiple classes. ❸ Among these IGL algorithms, DataDec stands out for its remarkable stability in different imbalanced scenarios. It consistently shows great performance gains across various datasets, highlighting its effectiveness and reliability.

### 4.2.5 ROBUSTNESS OF GRAPH-LEVEL TOPOLOGY-IMBALANCED ALGORITHMS

**Settings.** We manipulate datasets for the graph classification following settings in Appendix B.6 with different levels of the topology-imbalance ratios from **Low** to **High**, concerning multiple degrees of the graph size distribution to evaluate algorithm robustness.

**Results** (Figure 4(d)). ❶ SOLT-GNN demonstrated remarkable robustness across a spectrum of datasets and topology-imbalance scenarios, indicating its efficacy in handling varying levels of topology-imbalance. ❷ Contrarily, TopoImb did not consistently surpass GIN and exhibited notable variability in performance across different topology-imbalance degrees, suggesting that TopoImb may not be as reliable in maintaining performance stability for topology-imbalance changes. ❸ The results underscore the importance of algorithm choice in graph classification tasks, particularly in scenarios involving topology-imbalance, where robustness becomes a critical factor.

> **Key Insights for RQ2:** As the imbalance degree increases, the performance tends to degrade, especially under extreme conditions. Algorithms tailored to handle either issue demonstrate better robustness in respective contexts. Notably, class-imbalance and topology-imbalance do not seem to be entirely orthogonal issues. Future research should further investigate the impact of topology and class imbalance on each other in imbalanced graph learning by analyzing their intrinsic causes.

## 4.3 VISUALIZATIONS OF THE CLASSIFIER BOUNDARY (RQ3)

**Results** (Figure 5). Visualizations via t-SNE on Cora and COLLAB illustrate the classifier boundaries, with samples colored by predicted class labels. Quantitatively, the Silhouette score, which ranges from $-1$ to $1$ (higher values indicate better clustering), provides a clearer view of the clustering of sample embeddings. Results indicate that IGL algorithms effectively reduce class overlap and intuitively shift decision boundaries toward the minority class, enhancing the use of the minor class subspace. This is reflected in higher Silhouette scores for models like $G^2GNN$ and DataDec under graph-level class imbalance compared to GCN, particularly in challenging conditions.

> **Key Insights for RQ3:** Future research should focus more on exploring dynamic methods to adjust boundary sensitivity in response to imbalanced data, which could further enhance classification performance. Additionally, incorporating attention mechanisms or adversarial training techniques to improve boundary clarity under more extreme imbalanced conditions can offer stronger defenses against adversarial attacks and boost generalization to diverse biased graph data.

## 4.4 EFFICIENCY AND SCALABILITY ANALYSIS (RQ4)

**Results** (Figure 6). As we can observe, IGL algorithms generally have higher time or space complexity compared to backbones. Some algorithms (*e.g.*, GraphMixup, LTE4G and DataDec) can achieve relatively good performance improvement with less complexity increase. Besides, although some algorithms (*e.g.*, TOPOAUC, GraphPatcher and PASTEL) achieve remarkable effectiveness improvement, they largely increase the complexity of time and space. Additionally, the efficiency problem of IGL is specially pronounced on the large-scale dataset (ogbn-arXiv), as shown in Tables 2, 3, and 4, nearly half of IGL algorithms run out of memory. IGL algorithms struggle to achieve a satisfactory balance between performance and efficiency. Additional results are in Appendix D.

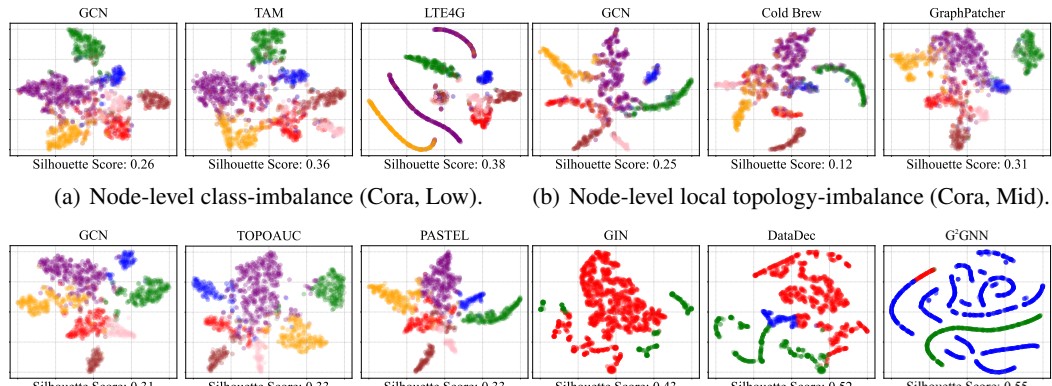

(a) Node-level class-imbalance (Cora, Low).    (b) Node-level local topology-imbalance (Cora, Mid).

(c) Node-level global topology-imbalance (Cora, High). (d) Graph-level class-imbalance (COLLAB, Low).

Figure 5: Visualization of node- and graph-level IGL algorithms in varying imbalanced scenarios.

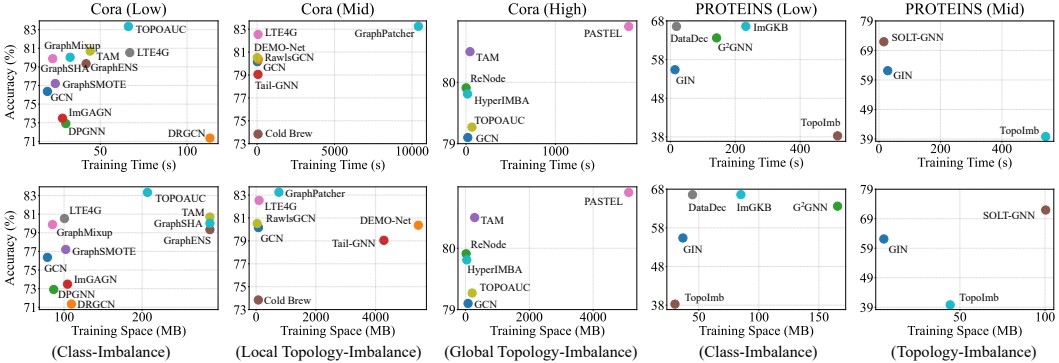

Figure 6: Time and space analysis of node- and graph-level IGL algorithms on Cora and Proteins.

> **Key Insights for RQ4:** Graph learning for ultra-large-scale data is a prominent research frontier in the community. The new paradigm of graph foundation models poses substantial challenges in memory-time-efficiently addressing imbalanced graph data and achieving high-quality representation learning. Investigating graph representation frameworks built on models like graph transformers, and state space models, *etc.*, presents a promising avenue for future development.

## 5 CONCLUSION AND FUTURE DIRECTIONS

This paper introduces the first comprehensive imbalanced graph learning benchmark, `IGL-Bench`, by integrating **24** methods across **17** graph datasets. We conduct extensive experiments to reveal the performance of IGL algorithms in terms of effectiveness, robustness and efficiency on node-level and graph-level tasks. We design and implement a package `IGL-Bench` (https://github.com/RingBDStack/IGL-Bench) that incorporates all the aforementioned protocols, baseline algorithms, processed datasets, and scripts to reproduce the results in this paper. Drawing upon our empirical analysis and insights, we point out some promising future directions for IGL community:

❶ **Unified Algorithm.** Class-imbalance and topology-imbalance simultaneously and widely exist in multi-domain graphs. Future research should revisit the optimization conflicts between two imbalance issues and develop a unified "one for both" IGL algorithm rooted in core nature of the problem.

❷ **Robustness and Generalization.** The practicality of IGL algorithms in real-world applications is essential. Future research should emphasize enhancing the robustness of IGL algorithms in extreme imbalance scenarios and improving their generalization to handle unseen testing domains or unprecedented distribution shifts, ensuring reliable performance in diverse real-world settings.

❸ **Efficiency and Scalability.** Empirical evidence suggests that current IGL algorithms struggle, or are infeasible to operate efficiently on large-scale graphs. As the size of graphs continues to grow exponentially, a key area of future research is the reduction of memory and computational complexity in IGL algorithms to ensure their efficient scalability and performance on large-scale graphs.

## ACKNOWLEDGMENTS

The corresponding author is Jianxin Li. This research was supported in part by the National Natural Science Foundation of China (NSFC) under Grants No. 62225202 and No. 62302023, the Fundamental Research Funds for the Central Universities, and the National Science Foundation (NSF) under Grants III-2106758 and POSE-2346158. We owe sincere thanks to all authors for their valuable efforts and contributions.

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

# Appendix

## Table of Contents

# A DATASETS AND ALGORITHMS

## A.1 BENCHMARK DATASETS

We adopt **17** benchmark datasets since ❶ they are extensively utilized for training and assessing IGL algorithms; ❷ they encompass a broad range of graph properties, spanning from small-scale to large-scale, from homophilic to heterophilic, and from node-level to graph-level; ❸ they cover diverse domains including citation networks, social networks, website networks, biochemicals, and co-occurrence networks. All the datasets integrated into our `IGL-Bench` are either published or publicly accessible. Table A.1 and Table A.2 provide the detailed statistics of the benchmark datasets, and their detailed descriptions are as follows.

- **Cora** (Yang et al., 2016) is a citation network dataset containing scientific publications classified into one of seven research areas. Each publication is represented by a feature vector indicating the presence or absence of words. The task is to predict the one-hot category label of a given publication. The dataset is licensed under Creative Commons 4.0.

- **CiteSeer** (Yang et al., 2016) is a citation network dataset, consisting of scientific publications, each labeled with one of six classes in the ont-hot vector form. It is commonly used for tasks such as document classification and citation prediction. The dataset is licensed under Creative Commons 4.0.

- **PubMed** (Yang et al., 2016) is a dataset of the biomedical literature, commonly used for tasks like document classification, information retrieval, and citation analysis. Each document is associated with a one-hot MeSH (Medical Subject Headings) topic label, which is used for document classification. The dataset is licensed under Creative Commons 4.0.

- **Computers** (Shchur et al., 2018) and **Photo** (Shchur et al., 2018) are Amazon products co-occurrence networks. Nodes represent goods and edges represent that two goods are frequently bought together. The task is to map goods to their respective product category. The datasets are licensed with MIT License.

- **ogbn-arXiv** (Hu et al., 2020) is a benchmark citation network derived from the arXiv website, consisting of a large number of nodes and edges, covering a wide range of research fields. Each node represents a paper, which is described by the word embeddings extracted from the title and abstract. Each directed edge indicates the citations between papers. It is used for tasks such as node classification and link prediction in academic citation networks. The dataset is licensed under ODC-BY.

- **Chameleon** (Rozemberczki et al., 2021) and **Squirrel** (Rozemberczki et al., 2021) are the Wikipedia page-page networks. Nodes represent web pages and edges represent hyperlinks between them. Node features represent several informative nouns on the Wikipedia pages. The task is to predict the average daily traffic of the web page. The datasets are licensed with GPL-3.0 License.

- **Actor** (Pei et al., 2019) is the actor-only induced subgraph of the film-director-actor-writer network. Each node corresponds to an actor, and the edge denotes co-occurrence on the same Wikipedia page. Node features represent keywords on the Wikipedia pages. The task is to classify nodes into five categories from the actor's Wikipedia. The dataset is made public with a license unspecified.

- **PTC-MR** (Bai et al., 2019) is a dataset of chemical compounds labeled with their mutagenic activity on bacteria. It has 344 molecules with a binary label indicating the carcinogenicity of compounds in rodents. It is used for tasks such as chemical compound classification and toxicity prediction. The dataset is made public with a license unspecified.

- **FRANKENSTEIN** (Orsini et al., 2015) is a set of molecular graphs with node features containing continuous values. A label denotes whether a molecule is a mutagen or non-mutagen. The dataset is made public with a license unspecified. The dataset is licensed under Creative Commons 1.0.

- **PROTEINS** (Borgwardt et al., 2005) is a set of macromolecules derived from Dobson and Doig, where nodes are structure elements. Edges denote nodes in an amino acid sequence or a close 3D space. The task is to predict whether a protein is an enzyme. The dataset is licensed under Creative Commons 4.0.

Table A.1: Statistics of benchmark datasets for node classification.

| Dataset | #Nodes | #Edges | #Classes | #Features | Avg. #Degree | #Homophily[1] |
|---|---|---|---|---|---|---|
| Cora [65] | 2,708 | 5,278 | 7 | 1,433 | 3.90 | 0.81 |
| CiteSeer [65] | 3,327 | 4,614 | 6 | 3,703 | 2.77 | 0.74 |
| PubMed [65] | 19,717 | 44,325 | 3 | 500 | 4.50 | 0.80 |
| Computers [46] | 13,752 | 245,861 | 10 | 767 | 35.76 | 0.78 |
| Photo [46] | 7,487 | 119,081 | 8 | 745 | 31.13 | 0.82 |
| ogbn-arXiv [14] | 169,343 | 1,157,799 | 40 | 767 | 13.67 | 0.65 |
| Chameleon [44] | 2,277 | 36,101 | 5 | 2,325 | 27.60 | 0.23 |
| Squirrel [44] | 5,201 | 217,073 | 5 | 2,089 | 76.33 | 0.22 |
| Actor [39] | 7,600 | 26,659 | 5 | 932 | 7.02 | 0.22 |

Table A.2: Statistics of benchmark datasets for graph classification.

| Dataset | #Graphs | Avg. #Nodes | Avg. #Edges | #Classes | #Features | Avg. #Degree | #$\mathcal{G}_{head}$[2] |
|---|---|---|---|---|---|---|---|
| PTC-MR [1] | 344 | 14.29 | 14.69 | 2 | 18 | 2.06 | 67 |
| FRANKENSTEIN [36] | 4,337 | 16.90 | 17.88 | 2 | 780 | 2.12 | 757 |
| PROTEINS [2] | 1,113 | 39.06 | 72.82 | 2 | 3 | 3.73 | 218 |
| D&D [47] | 1,178 | 284.32 | 715.66 | 2 | 89 | 5.03 | 234 |
| IMDB-B [5] | 1,000 | 19.77 | 96.53 | 2 | 65 | 9.77 | 194 |
| REDDIT-B [64] | 2,000 | 429.63 | 497.75 | 2 | 566 | 2.32 | 400 |
| ogbg-molhiv [14] | 41,127 | 25.51 | 27.50 | 2 | 9 | 4.29 | 8,225 |
| COLLAB [22] | 5000 | 74.49 | 2457.78 | 3 | 369 | 65.99 | 991 |

- **D&D** (Shervashidze et al., 2011) contains graphs of protein structures. A node represents an amino acid and edges are constructed if the distance of two nodes is less than 6Å. The label denotes whether a protein is an enzyme or a non-enzyme. The dataset is made public with a license unspecified.

- **IMDB-B** (Cai & Wang, 2018) is a movie collaboration dataset where actor/actress and genre information of different movies are collected. For each graph, nodes represent actors/actresses and there is an edge between them if they appear in the same movie. The dataset is licensed under Creative Commons 4.0.

- **REDDIT-B** (Yanardag & Vishwanathan, 2015) is a balanced dataset where each graph corresponds to an online discussion thread where nodes correspond to users, and there is an edge between two nodes if at least one of them responds to another's comment. The dataset is licensed under Creative Commons 4.0.

- **ogbg-molhiv** (Hu et al., 2020) is a natural imbalanced molecular dataset, consisting of a large number of graphs. Each graph represents a molecule, where nodes are atoms, and edges are chemical bonds. Node features contain atomic number and chirality, as well as other additional atom features. The dataset is made public with an MIT License.

- **COLLAB** (Leskovec et al., 2005) is the scientific collaboration dataset, deriving from three public collaboration datasets. The networks of researchers were generated from each field, and each was labeled as the researcher field. The task is to determine to which field the collaboration network of a researcher belongs. The dataset is licensed under Creative Commons 4.0.

## A.2 BENCHMARK ALGORITHMS

In our developed `IGL-Bench`, we integrate **24** state-of-the-art IGL algorithms, including **10** node-level class-imbalanced IGL algorithms: DRGCN (Shi et al., 2020), DPGNN (Wang et al., 2021),

---

[1]We report node homophily ratio that normalizes the edge homophily across neighborhoods (Pei et al., 2019).

[2]For each dataset, we divide graphs into head and tail with a predefined ratio based on the Pareto principle Sanders (1987) (also known as 20/80 rule) to employ the 20% largest graphs as head graphs, and the rest 80% as tail graphs.

Table A.3: Summary of representative Imbalanced Graph Representation Learning (IGL) algorithms integrated in `IGL-Bench` concerning the imbalance types, downstream tasks, method levels, and computational complexity. We also provide public access to the official algorithm implementations.

| Type | Algorithm | Task | Data-Level | | | Algorithm-Level | | | Computational Complexity[3] | Code |
|------|-----------|------|----|----|----|----|------|----|------|------|
| | | | IG | AG | PL | MR | Loss | RG | | |
| Node-Level Class-Imbalance | DRGCN [48] | NC | | ✓ | | | | ✓ | $\mathcal{O}(\|\mathcal{V}\| + \|\mathcal{E}\|)$ | link |
| | DPGNN [57] | NC | | | ✓ | ✓ | ✓ | | $\mathcal{O}(\|\mathcal{V}\| + \|\mathcal{E}\|)$ | link |
| | ImGAGN [42] | NC | | ✓ | | | | | $\mathcal{O}(\|\mathcal{V}\| + \|\mathcal{E}\|)$ | link |
| | GraphSMOTE [70] | NC | ✓ | | | | | | $\mathcal{O}(\|\mathcal{V}\|^2) + \mathcal{O}(\|\mathcal{E}\|)$ | link |
| | GraphENS [37] | NC | ✓ | | | | | | $\mathcal{O}(\|\mathcal{V}\| + \|\mathcal{E}\|)$ | link |
| | GraphMixup [60] | NC | ✓ | | | | | | $\mathcal{O}(\|\mathcal{V}\|^2) + \mathcal{O}(\|\mathcal{E}\|)$ | link |
| | LTE4G [67] | NC | | | | ✓ | | ✓ | $\mathcal{O}(\|\mathcal{V}\|^2) + \mathcal{O}(\|\mathcal{E}\|)$ | link |
| | TAM [49] | NC | | | | | ✓ | | $\mathcal{O}(\|\mathcal{V}\|C + \|\mathcal{E}\|)$ | link |
| | TOPOAUC [8] | NC | | | | | ✓ | ✓ | $\mathcal{O}(\|\mathcal{V}\| + \|\mathcal{E}\|)$ | link |
| | GraphSHA [24] | NC | ✓ | | | | | | $\mathcal{O}(\|\mathcal{V}\| + \|\mathcal{E}\|)$ | link |
| Node-Level Topology-Imbalance (local) | DEMO-Net [59] | NC | | | | ✓ | | | $\mathcal{O}(\|\mathcal{V}\| + \|\mathcal{E}\|)$ | link |
| | meta-tail2vec [30] | NC | | | | ✓ | ✓ | | $\mathcal{O}(\|\mathcal{V}\| + \|\mathcal{E}\|)$ | link |
| | Tail-GNN [31] | NC | | | | ✓ | ✓ | | $\mathcal{O}(\|\mathcal{V}\| + \|\mathcal{E}\|)$ | link |
| | Cold Brew [72] | NC | | | | ✓ | ✓ | | $\mathcal{O}(\|\mathcal{V}\| + \|\mathcal{E}\|)$ | link |
| | LTE4G [67] | NC | | | | ✓ | | ✓ | $\mathcal{O}(\|\mathcal{V}\|^2) + \mathcal{O}(\|\mathcal{E}\|)$ | link |
| | RawlsGCN [19] | NC | | | | | ✓ | | $\mathcal{O}(\|\mathcal{V}\| + \|\mathcal{E}\|)$ | link |
| | GraphPatcher [17] | NC | ✓ | | | | | ✓ | $\mathcal{O}(\|\mathcal{V}\| + \|\mathcal{E}\|)$ | link |
| Node-Level Topology-Imbalance (global) | ReNode [7] | NC | | | | | ✓ | | $\mathcal{O}(\|\mathcal{V}\| + \|\mathcal{E}\|)$ | link |
| | TAM [49] | NC | | | | | ✓ | | $\mathcal{O}(\|\mathcal{V}\|C + \|\mathcal{E}\|)$ | link |
| | PASTEL [52] | NC | | | | ✓ | ✓ | ✓ | $\mathcal{O}(\|\mathcal{V}^2\|) + \mathcal{O}(\|\mathcal{E}\|)$ | link |
| | TOPOAUC [8] | NC | | | | | ✓ | ✓ | $\mathcal{O}(\|\mathcal{V}\| + \|\mathcal{E}\|)$ | link |
| | HyperIMBA [12] | NC | | | | ✓ | ✓ | | $\mathcal{O}(\|\mathcal{V}\| + \|\mathcal{E}\|)$ | link |
| Graph-Level Class-Imbalance | G²GNN [58] | GC | ✓ | | | ✓ | ✓ | | $\mathcal{O}(\binom{\|\mathbf{G}\|}{2} \max \|\mathcal{V}^{\mathcal{G}_i}\|^3)$ | link |
| | TopoImb [71] | NC, GC | | | | | ✓ | | $\mathcal{O}(\sum(\|\mathcal{V}^{\mathcal{G}_i}\| + \|\mathcal{E}^{\mathcal{G}_i}\|))$ | link |
| | DataDec [68] | NC, GC | | | | ✓ | | | $\mathcal{O}(\sum(\|\mathcal{V}^{\mathcal{G}_i}\| + \|\mathcal{E}^{\mathcal{G}_i}\|))$ | link |
| | ImGKB [54] | GC | | | | ✓ | ✓ | | $\mathcal{O}(\sum(\|\mathcal{V}^{\mathcal{G}_i}\| + \|\mathcal{E}^{\mathcal{G}_i}\|))$ | link |
| Graph-Level Topology-Imbalance | SOLT-GNN [32] | GC | | | | | ✓ | ✓ | $\mathcal{O}(\sum(\|\mathcal{V}^{\mathcal{G}_i}\| + \|\mathcal{E}^{\mathcal{G}_i}\|))$ | link |
| | TopoImb [71] | NC, GC | | | | | ✓ | | $\mathcal{O}(\sum(\|\mathcal{V}^{\mathcal{G}_i}\| + \|\mathcal{E}^{\mathcal{G}_i}\|))$ | link |

ImGAGN (Qu et al., 2021), GraphSMOTE (Zhao et al., 2021), GraphENS (Park et al., 2021), GraphMixup (Wu et al., 2022), LTE4G (Yun et al., 2022), TAM (Song et al., 2022), TOPOAUC (Chen et al., 2022) and GraphSHA (Li et al., 2023b); **12** node-level topology-imbalanced IGL algorithms: DEMO-Net (Wu et al., 2019), meta-tail2vec (Liu et al., 2020), Tail-GNN (Liu et al., 2021), Cold Brew (Zheng et al., 2021), LTE4G (Yun et al., 2022), RawlsGCN (Kang et al., 2022), GraphPatcher (Ju et al., 2024a), ReNode (Chen et al., 2021), TAM (Song et al., 2022), PASTEL (Sun et al., 2022b), TOPOAUC (Chen et al., 2022), and HyperIMBA (Fu et al., 2023); **4** graph-level class-imbalanced IGL algorithms: G²GNN (Wang et al., 2022), TopoImb (Zhao et al., 2022), DataDec (Zhang et al., 2023), and ImGKB (Tang & Liang, 2023); **2** graph-level topology-imbalanced IGL algorithms: SOLT-GNN (Liu et al., 2022) and TopoImb (Zhao et al., 2022).

We conclude the aforementioned representative IGL algorithms in Tabel A.3 in terms of the downstream task, method level, and computational complexity. The **Task** column indicates the specific downstream tasks the algorithm can handle, where "NC" stands for node classification, and "GC" stands for graph classification. The **Data-Level** column implies the algorithm handles the imbalance issue from the training data perspective, where "IG" stands for generating samples by interpolating, "AG" stands for generating samples by adversarial training, and "PL" stands for generating pseudo labels for a large number of unlabeled nodes. The **Algorithm-Level** column suggests an algorithm-level contribution to solve the imbalance learning problems, where "MR" denotes refining GNN models for improving the representation learning process, "Loss" represents designing or engineering loss function for sample reweighting, *etc.*, and "RG" stands for utilizing extra regularizers for the imbalance recalibrating. We further introduce all the IGL algorithms as follows.

---

[3]For brevity, only the main bottlenecks of the algorithm's computational complexity are analyzed here, while the remaining negligible parts are uniformly ignored. The meanings of notations follow definitions in Section 2.

- **DRGCN** (Shi et al., 2020) is proposed to address the node-level class-imbalance issue. It employs a GNN-centric strategy, incorporating a conditioned generative adversarial network (GAN) to create synthetic nodes to balance redistribution. Additionally, it utilizes a KL-divergence constraint to harmonize the representation distribution of unlabeled nodes with that of labeled ones. The code is made available with a license unspecified.

- **DPGNN** (Wang et al., 2021) is proposed to address the node-level class-imbalance issue. It employs a class prototype-driven training approach to balance training loss across classes and transfer knowledge from head classes to tail classes, with the help of distance metric learning to accurately capture the relative positions of nodes concerning class prototypes, as well as smoothing representations of adjacent nodes while separating interclass prototypes. The code is made available with a license unspecified.

- **ImGAGN** (Qu et al., 2021) is proposed to address the node-level class-imbalance issue. It applies the generative adversarial network (GAN) to generate synthetic nodes, which simulates both the minority class nodes' attribute distribution and network topological structure distribution by generating a set of synthetic minority nodes such that the number of nodes in different classes can be balanced. The code is made available with a license unspecified.

- **GraphSMOTE** (Zhao et al., 2021) is proposed to address the node-level class-imbalance issue. It evolves around the generation of synthetic nodes to balance classes by a technique inspired by SMOTE (Chawla et al., 2002), which is the first data interpolation method on graphs by generating a synthetic minority node through interpolation between two real minority nodes in the embedding space. It pre-trains an edge predictor using a graph reconstruction objective on real nodes and existing edges to determine the connectivity between the synthetic node and existing nodes. The code is made available with a license unspecified.

- **GraphENS** (Park et al., 2021) is proposed to address the node-level class-imbalance issue. It creates a synthetic minority node by blending a real minority node with a randomly chosen target node. Notably, GraphENS (Park et al., 2021) prioritizes the neighbors of minority nodes, recognizing their significant informational value. To address this bias, it incorporates neighbor sampling and saliency-based node mixing techniques. The code is made available with an MIT License.

- **GraphMixup** (Wu et al., 2022) is proposed to address the node-level class-imbalance issue. GraphMixup (Wu et al., 2022) executes reinforcement mixup within the semantic space instead of the input or embedding space, thereby averting the creation of out-of-domain minority samples. It integrates two supplementary self-supervised learning objectives: local-path prediction and global-path prediction, aiming to encompass both local and global insights within the graph structure. The code is made available with an MIT License.

- **LTE4G** (Yun et al., 2022) is proposed to address the node-level class-imbalance issue. It takes into account the imbalance in both node classes and degrees. LTE4G (Yun et al., 2022) divides nodes into balanced subsets and assigns them to specialized Graph Neural Networks (GNNs) based on their similarity to each class prototype vector. The class with the highest similarity score is assigned to each node subset. Subsequently, LTE4G (Yun et al., 2022) utilizes knowledge distillation to train class-specific student models, thereby improving classification performance. The code is made available with a license unspecified.

- **TAM** (Song et al., 2022) is proposed to address the node-level class-imbalance issue and topology-imbalance issue simultaneously. TAM (Song et al., 2022) resolves the class-imbalance issue by integrating graph topology information into its loss function designs and addressing the decreased homogeneity among minority nodes. Particularly, TAM (Song et al., 2022) introduces connectivity- and distribution-aware margins to guide the model, highlighting class-wise connectivity and neighbor-label distribution in an innovative manner. The code is made available with an MIT License.

- **TOPOAUC** (Chen et al., 2022) is proposed to address the node-level class-imbalance and topology-imbalance issue simultaneously. It develops a multi-class AUC optimization work to deal with the class imbalance problem. With respect to topology imbalance, TOPOAUC (Chen et al., 2022) proposes a Topology-Aware Importance Learning mechanism (TAIL), which considers the topology of pairwise nodes and different contributions of topology information to pairwise node neighbors. The code is made available with a license unspecified.

- **GraphSHA** (Li et al., 2023b) is proposed to address the node-level class-imbalance issue. It aims to expand the decision boundaries of minority classes by generating more challenging synthetic samples from these classes. Additionally, GraphSHA (Li et al., 2023b) introduces a module named SemiMixup, which is designed to transfer the enlarged boundary information into the interior of the minority classes while preventing the leakage of information from the minority classes to their neighboring majority classes. This helps to enhance the separability of minority classes without compromising their integrity. The code is made available with an MIT License.

- **DEMO-Net** (Wu et al., 2019) is proposed to address the node-level topology-imbalance issue. Inspired by the Weisfeiler-Lehman graph isomorphism test, DEMO-Net (Wu et al., 2019) explicitly captures integrated graph topology and node attributes. It introduces multi-task graph convolution, where each task focuses on learning node representations for nodes with specific degree values, thereby preserving the degree-specific graph structure. Furthermore, DEMO-Net (Wu et al., 2019) devises a new graph-level pooling/readout scheme to learn graph representations, ensuring they reside in a degree-specific Hilbert kernel space. The code is made available with a license unspecified.

- **meta-tail2vec** (Liu et al., 2020) is proposed to address the node-level local topology-imbalance issue. It frames the objective of learning from imbalanced data, particularly focusing on learning embeddings for tail nodes, as a few-shot regression task, considering the limited connections associated with each tail node. Moreover, meta-tail2vec (Liu et al., 2020) recognizes that each node exists within its unique local context and therefore adapts the regression model individually for each tail node, personalizing the learning process. The code is made available with an MIT License.

- **Tail-GNN** (Liu et al., 2021) is proposed to address the node-level local topology-imbalance issue. While GNNs are capable of learning effective node representations, they often handle all nodes in a generic manner and do not specifically cater to the numerous tail nodes. Tail-GNN (Liu et al., 2021) leverages the innovative concept of transferable neighborhood translation to capture the diverse relationships between a node and its neighboring nodes. In essence, Tail-GNN (Liu et al., 2021) develops a node-specific adaptation technique that tailors the global translation to the individual needs of each node. The code is made available with a license unspecified.

- **Cold Brew** (Zheng et al., 2021) is proposed to address the node-level local topology-imbalance issue, with a particular focus on the most extreme cases in graphs where a node lacks any neighboring connections, known as the Strict Cold Start (SCS) problem (Qian et al., 2020). Cold Brew (Zheng et al., 2021) employs a teacher-student distillation framework to address the SCS issue and the challenge posed by noisy neighbors in the context of GNNs. Additionally, Cold Brew (Zheng et al., 2021) introduces the concept of feature contribution ratio, a metric that quantifies the performance of inductive GNNs in resolving the SCS problem. The code is made available with an Apache-2.0 License.

- **RawlsGCN** (Kang et al., 2022) is proposed to address the node-level local topology-imbalance issue. It approaches the issue of degree-related performance disparities through the lens of the Rawlsian difference principle, a concept derived from the theory of distributive justice. RawlsGCN (Kang et al., 2022) is designed to equalize the performance between nodes with low and high degrees while also optimizing for task-specific objectives, ensuring a fairer allocation of predictive utility across the graph. The code is made available with an MIT License.

- **GraphPatcher** (Ju et al., 2024a) is proposed to address the node-level local topology-imbalance issue. It suggests a test-time augmentation framework designed to improve the test-time generalization ability of any GNNs for low-degree nodes. In detail, GraphPatcher (Ju et al., 2024a) successively creates virtual nodes to repair the artificially generated low-degree nodes through corruptions, with the goal of incrementally reconstructing the target GNN's predictions across a series of progressively corrupted nodes. The code is made available with a license unspecified.

- **ReNode** (Chen et al., 2021) is proposed to address the node-level global topology-imbalance issue. ReNode (Chen et al., 2021) adjusts the weights of labeled nodes according to their proximity to class boundaries, thereby enhancing performance, especially for nodes near boundaries and those distant from them. Additionally, a metric is devised to measure this imbalance, utilizing influence conflict detection. ReNode (Chen et al., 2021) effectively addresses both class-imbalance and topology-imbalance challenges concurrently. The code is made available with an MIT License.

- **PASTEL** (Sun et al., 2022b) is proposed to address the node-level global topology-imbalance issue. PASTEL (Sun et al., 2022b) addresses topology imbalance by optimizing the paths of

information propagation. Its goal is to mitigate the under-reaching and over-squashing effects by improving intra-class connectivity and employing a position encoding mechanism. Additionally, PASTEL (Sun et al., 2022b) utilizes a class-wise conflict measure for edge weights to aid in node class separation. The code is made available with an MIT License.

- **HyperIMBA** (Fu et al., 2023) is proposed to address the node-level global topology-imbalance issue. HyperIMBA (Fu et al., 2023) employs hyperbolic geometric embedding to assess the hierarchy of labeled nodes. It then modifies label information propagation and adjusts the objective margin according to the node's hierarchy, effectively tackling issues arising from hierarchy imbalance. The code is made available with a license unspecified.

- **G²GNN** (Wang et al., 2022) is proposed to address the graph-level class-imbalance issue. It employs additional supervision at both global and local levels: globally, through neighboring graphs, and locally, via stochastic augmentations. G²GNN (Wang et al., 2022) constructs a Graph of Graphs (GoG) by utilizing kernel similarity and implements GoG propagation for information aggregation. Furthermore, it utilizes topological augmentation with self-consistency regularization at the local level. These combined strategies improve model generalizability and consequently enhance classification performance. The code is made available with a license unspecified.

- **TopoImb** (Zhao et al., 2022) is proposed to address the graph-level class-imbalance and topology-imbalance issues. Graph-level topology imbalance often stems from uneven motif distribution (*e.g.*, functional groups), resulting in a lack of training instances for minority groups. TopoImb (Zhao et al., 2022) tackles this challenge by dynamically updating the identification of topology groups and assigning importance weights to under-represented instances during training. This approach enhances the learning efficacy of minority topology groups and mitigates overfitting to majority groups. The code is made available with a license unspecified.

- **DataDec** (Zhang et al., 2023) is proposed to address both the node-level and graph-level class-imbalance issues. DataDec (Zhang et al., 2023) develops a unified data-model dynamic sparsity framework to address challenges brought by training upon massive class-imbalanced graph data. The key idea of DataDec (Zhang et al., 2023) is to identify the informative subset dynamically during the training process by adopting sparse graph contrastive learning. The code is made available with a license unspecified.

- **ImGKB** (Tang & Liang, 2023) is proposed to address the graph-level class-imbalance issue. It combines the restricted random walk kernel with the global graph information bottleneck (GIB) (Wu et al., 2020) to enhance the performance of imbalanced graph classification tasks. To prevent the dominant class graphs from introducing redundant information into the kernel outputs, ImGKB (Tang & Liang, 2023) frames the entire kernel learning process as a Markovian decision process. It then utilizes the global GIB (Wu et al., 2020) approach to optimize the learning, ensuring that the kernel effectively captures the relevant information for each class. The code is made available with a license unspecified.

- **SOLT-GNN** (Liu et al., 2022) is proposed to address both the graph-level topology-imbalance issues. Graphs with larger sizes (number of nodes) tend to possess more complex topological structures. To counter performance biases caused by the intricate topological structures, SOLT-GNN (Liu et al., 2022) enhances the performance of smaller graphs. It identifies co-occurrence patterns in larger graphs (or "head" graphs) and transfers this knowledge to augment smaller graphs, improving their performance. The code is made available with a license unspecified.

## B  DETAILS OF THE DATASET SETTINGS

### B.1  IMBALANCE RATIO DEFINITION

We provide additional explanations on the details of the imbalance ratio defined in Tabel 1.

- **Node/Graph-Level Class-Imbalance.** Given a set of labeled training node/graph classes $\mathcal{V}_L = \bigcup_{1 \leq i \leq C} \mathcal{C}_i$, the imbalance ratio is defined to be the ratio between the number of nodes/graphs in the majority class and the number of nodes/graphs in the minority class, *i.e.*,

$$\rho = \frac{\max_{i=1}^{C} |\mathcal{C}_i|}{\min_{j=1}^{C} |\mathcal{C}_j|}. \tag{B.1}$$

Table B.1: Definitions of the imbalance ratio ($\rho$) across different imbalance types.

| Imbalance Type | Definition | Explanation |
|---|---|---|
| Node-Level Class-Imbalance Graph-Level Class-Imbalance | $\rho = \dfrac{\max_{i=1}^{C} \|\mathcal{C}_i\|}{\min_{j=1}^{C} \|\mathcal{C}_j\|}$ | The imbalance ratio is set to the ratio between the number of samples ($\|\mathcal{C}\|$) in the majority and the minority class. |
| Node-Level Topology-Imbalance (local and global) | $\rho = \dfrac{\frac{1}{\|\mathcal{H}_n\|} \sum d(v), v \in \mathcal{H}_n}{\frac{1}{\|\mathcal{T}_n\|} \sum d(v), v \in \mathcal{T}_n}$ | The **local** imbalance ratio is set to the ratio of the average node degree ($d(v)$) of the head node set ($\mathcal{H}_n$) to the average node degree of the tail node set ($\mathcal{T}_n$). |
| | $\rho = -10 \cdot \log \|RC \cdot SC\|$ | The **global** imbalance ratio is set to the negative logarithm of the absolute value of the product of the Reaching Coefficient ($RC$) and the Squashing Coefficient ($SC$). |
| Graph-Level Topology-Imbalance | $\rho = \dfrac{\frac{1}{\|\mathcal{H}_g\|} \sum \|\mathcal{G}_i\|, \mathcal{G}_i \in \mathcal{H}_g}{\frac{1}{\|\mathcal{T}_g\|} \sum \|\mathcal{G}_j\|, \mathcal{G}_j \in \mathcal{T}_g}$ | The imbalance ratio is set to the ratio of the average graph size (number of nodes) of the head graph set ($\mathcal{H}_g$) to the average graph size of the tail graph set ($\mathcal{T}_g$). |

Node-level class-imbalance occurs when there is an uneven spread of labeled nodes among different classes. This can lead the model to prioritize learning from classes abundant in labeled instances, potentially neglecting those with fewer examples. Graph-level is similar to node-level class-imbalance. This issue frequently arises in practical contexts, such as imbalanced chemical compound classification, where the distributions of labeled graphs are skewed. Typically, this bias favors the majority class, which comprises more labeled graphs.

- **Node-Level Topology-Imbalance.**
  - **Local Imbalance.** Given a set of labeled nodes $\mathcal{V}_L = \{v_1, \cdots, v_N\}$ with the splits designating the top 20% of nodes by degree as high-degree head node set $\mathcal{H}_n$ and the rest 80% as low-degree tail node set $\mathcal{T}_n$ following the Pareto principle (also known as the 20/80 rule) (Sanders, 1987). The local node-level topology-imbalance is set to the ratio of the average node degree of the head training node set to the average node degree of the tail training node set, *i.e.*,

$$\rho = \frac{\frac{1}{\|\mathcal{H}_n\|} \sum d(v), v \in \mathcal{H}_n}{\frac{1}{\|\mathcal{T}_n\|} \sum d(v), v \in \mathcal{T}_n}, \tag{B.2}$$

  where $d(\cdot)$ denotes node degree and we require $d(v) \geq 1$. Node degrees frequently exhibit a long-tail distribution. Head nodes, which have high degrees, benefit from richer structural information, resulting in superior performance in downstream tasks such as node classification. In contrast, tail nodes with low degrees possess limited topological information, which hampers their performance (Liu et al., 2020; 2021).

  - **Global Imbalance.** The global imbalance is facilitated by two aspects: **Under-Reaching** and **Over-Squashing** (Sun et al., 2022b). Under-Reaching refers to the phenomenon that the influence from labeled nodes decays with the topology distance, resulting in the nodes being far away from labeled nodes lacking supervision information. Over-Squashing refers to the phenomenon of the supervision information of valuable labeled nodes being squashed when passing across the narrow path together with other useless information. The global node-level topology-imbalance ratio is set to the 10x negative logarithm of the absolute value of the product of the Reaching Coefficient ($RC$) and the Squashing Coefficient ($SC$) (Sun et al., 2022b), *i.e.*,

$$\rho = -10 \cdot \log \|RC \cdot SC\|. \tag{B.3}$$

    ○ **Reaching Coefficient ($RC$)** is the mean length of the shortest path from unlabeled to the labeled nodes of their corresponding classes, *i.e.*,

$$RC = \frac{1}{\|\mathcal{V}_U\|} \sum_{v_i \in \mathcal{V}_U} \frac{1}{\|\mathcal{V}_L^{\mathbf{y}_i}\|} \sum_{v_j \in \mathcal{V}_L^{\mathbf{y}_i}} \left(1 - \frac{\log \|\mathcal{P}_{sp}(v_i, v_j)\|}{\log D_{\mathcal{G}}}\right), \tag{B.4}$$

    where $\mathcal{V}_L^{\mathbf{y}_i}$ denotes the nodes in $\mathcal{V}_L$ whose label is $\mathbf{y}_i$, $\mathcal{P}_{sp}(v_i, v_j)$ denotes the shortest path between $v_i$ and $v_j$, and $\|\mathcal{P}_{sp}(v_i, v_j)\|$ denotes its length, and $D_{\mathcal{G}}$ is the graph diameter.
    ○ **Squashing Coefficient ($SC$)** is the mean Ricci curvature (Ollivier, 2009) of edges on the shortest path from unlabeled to the labeled nodes of their corresponding classes, *i.e.*,

Table B.2: Number of nodes for each class in node-level datasets under different $\rho$.

| Dataset | | # Node for Each Class |
|---|---|---|
| Cora [65] | $\rho = 20$ | **5**, 9, 15, 24, 40, 80, **100** |
| | $\rho = 100$ | **2**, 3, 6, 14, 17, 31, **200** |
| CiteSeer [65] | $\rho = 20$ | **7**, 13, 25, 46, 99, **140** |
| | $\rho = 100$ | **2**, 5, 12, 31, 80, **200** |
| PubMed [65] | $\rho = 20$ | **77**, 354, **1540** |
| | $\rho = 100$ | **17**, 254, **1700** |
| Computers [46] | $\rho = 20$ | **20**, 28, 39, 55, 76, 107, 150, 208, 297, **400** |
| | $\rho = 100$ | **5**, 9, 15, 25, 43, 71, 119, 199, 394, **500** |
| Photo [46] | $\rho = 20$ | **13**, 21, 32, 49, 75, 116, 194, **260** |
| | $\rho = 100$ | **3**, 7, 13, 26, 51, 98, 262, **300** |
| ogbn-arXiv [14] | $\rho = 20$ | **47**, 51, 56, 60, 65, 70, 76, 83, 89, 97, 105, 113, 123, 133, 144, 156, 168, 182, 197, 213, 231, 250, 271, 293, 317, 343, 371, 401, 434, 470, 509, 550, 596, 645, 697, 755, 817, 919, **940** |
| | $\rho = 100$ | **13**, 15, 17, 19, 22, 25, 28, 32, 36, 41, 46, 52, 59, 66, 75, 85, 96, 108, 122, 138, 156, 176, 199, 225, 254, 286, 323, 365, 412, 465, 525, 593, 669, 756, 853, 963, 1154, 1268, **1300** |
| Chameleon [44] | $\rho = 20$ | **6**, 12, 27, 60, **120** |
| | $\rho = 100$ | **1**, 5, 25, 94, **100** |
| Squirrel [44] | $\rho = 20$ | **14**, 29, 62, 135, **280** |
| | $\rho = 100$ | **3**, 11, 35, 171, **300** |
| Actor [39] | $\rho = 20$ | **20**, 43, 91, 206, **400** |
| | $\rho = 100$ | **5**, 16, 52, 187, **500** |

$$SC = \frac{1}{|\mathcal{V}_U|} \sum_{v_i \in \mathcal{V}_U} \frac{1}{|\mathcal{N}_{\mathbf{y}_i}(v_i)|} \sum_{v_j \in \mathcal{N}_{\mathbf{y}_i}(v_i)} \frac{\sum_{e_{kt} \in \mathcal{P}_{sp}(v_i, v_j)} Ric(v_k, v_t)}{|\mathcal{P}_{sp}(v_i, v_j)|}, \tag{B.5}$$

where $\mathcal{N}_{\mathbf{y}_i}(v_i)$ denotes the labeled nodes of class $\mathbf{y}_i$ that can reach the node $v_i$, $Ric(\cdot, \cdot)$ denotes the Ricci curvature, and $|\mathcal{P}_{sp}(v_i, v_j)|$ denotes the length of shortest path between node pair $v_i$ and $v_j$.

- **Graph-Level Topology-Imbalance.** Given a set of labeled graphs $\mathbf{G}_L = \{\mathcal{G}_1, \cdots, \mathcal{G}_N\}$ with the splits designating the top 20% of graphs by graph size (the number of nodes) as large-size head graph set $\mathcal{H}_g$ and the rest 80% as small-size tail graph set $\mathcal{T}_g$ following the Pareto principle (20/80 rule) (Sanders, 1987). The imbalance ratio is set to the ratio of the average graph size of the head graph set to the average graph size of the tail graph set, *i.e.*,

$$\rho = \frac{\frac{1}{|\mathcal{H}_g|} \sum |\mathcal{G}_i|, \mathcal{G}_i \in \mathcal{H}_g}{\frac{1}{|\mathcal{T}_g|} \sum |\mathcal{G}_j|, \mathcal{G}_j \in \mathcal{T}_g}. \tag{B.6}$$

The complex connections within graphs can result in topology imbalances across different graphs. This imbalance frequently appears as variations in graph sizes. Generally, graphs with larger sizes tend to be more expressive and thus produce better performance compared to smaller counterparts. This dynamic can introduce bias in applications like molecular or protein prediction.

## B.2 MANIPULATED CLASS-IMBALANCED DATASETS FOR NODE CLASSIFICATION

**Dataset Settings.** We perform the node classification task semi-supervised on **9** manipulated class-imbalanced datasets, where the train/val/test split satisfies the ratio of 1:1:8. Specifically, to construct the long-tailed distribution of the number of training nodes concerning varying imbalance ratio $\rho$ defined in Equation B.1, we assume that the number of nodes in each class in the training set grows exponentially, *i.e.*, $|\mathcal{C}_{i+1}| = \mu|\mathcal{C}_i|$, where $i$ is the class index, $|\mathcal{C}_i|$ is the number of $i$-th indexed class training samples and $\mu \in (0,1)$ is the coefficient. Therefore, given the total number of nodes in the training set and $\rho$, the number of nodes used for training in each class can be calculated deterministically. All nodes other than those used for training and validation are assigned to the test set. To provide a thorough evaluation, we consider three typical situations in `IGL-Bench`, *i.e.*, the **class-balanced** setting ($\rho = 1$ and each class has an equal number of training nodes), the **class-imbalanced** setting ($\rho = 20$), and the **extreme class-imbalanced** setting ($\rho = 100$).

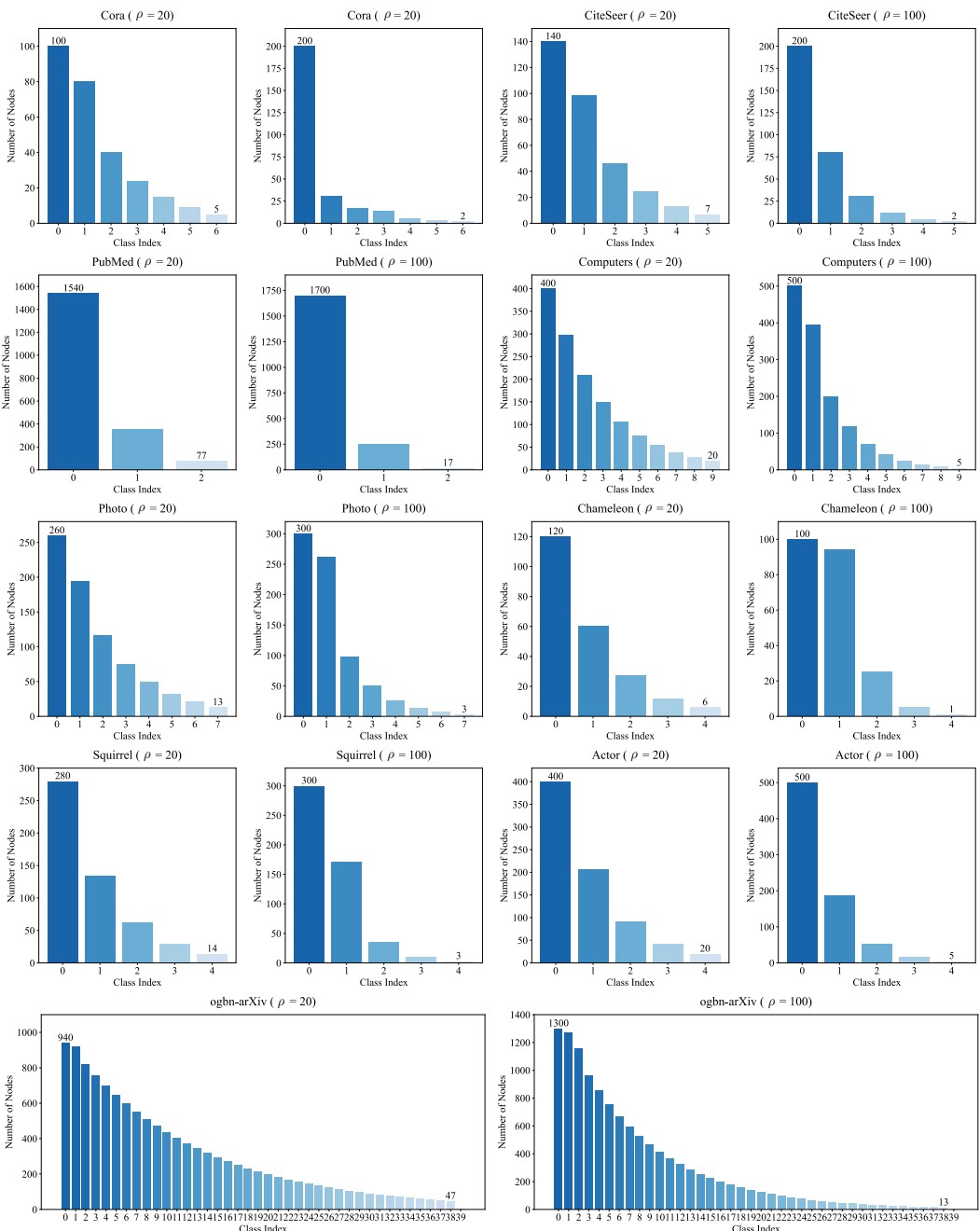

Figure B.1: Visualizations of the distribution of the number of nodes in the training sets for 9 benchmark datasets with different imbalance ratios. Note that, we calculate the imbalance ratio for the ogbn-arXiv (Hu et al., 2020) by the ratio between the number of nodes/graphs in the majority class and the number of nodes/graphs in the **sub**-minority class due to insufficient training nodes in some classes.

**Dataset Preview.** We present a visualization of the distribution of the number of nodes in the training sets for each dataset in Table B.2 and Figure B.1. It clearly reveals that the distribution follows a long-tail pattern. Notably, as the parameter $\rho$ increases, the number of nodes decreases more sharply, accentuating the long-tail effect. The higher the value of $\rho$, the more pronounced decline in node numbers, resulting in an even longer and more extended "tail". This trend indicates a significant imbalance, where a few classes are highly prevalent while the majority are sparsely represented.

Table B.3: Statistics of the manipulated local topology-imbalanced datasets (**training**) for node classification. The number of nodes for each class is equal (class-balanced), and the imbalance ratio $\rho$ is the ratio between the average degree of the head nodes and the average degree of the tail nodes.

| Dataset | Level | #Nodes per Class | #Head Nodes per Class | #Tail Nodes per Class | Avg. #Degree (Head Nodes) | Avg. #Degree (Tail Nodes) | Imbalance Ratio $\rho$ |
|---|---|---|---|---|---|---|---|
| Cora [65] | Low | 39 | 4 | 35 | 5.64 | 2.84 | 1.98 |
| | Mid | | 8 | 31 | 8.96 | 2.74 | 3.27 |
| | High | | 12 | 27 | 13.53 | 2.00 | 6.74 |
| CiteSeer [65] | Low | 55 | 6 | 49 | 4.47 | 1.83 | 2.44 |
| | Mid | | 12 | 43 | 6.78 | 1.62 | 4.18 |
| | High | | 18 | 37 | 9.61 | 1.17 | 8.21 |
| PubMed [65] | Low | 657 | 66 | 591 | 8.58 | 2.08 | 4.12 |
| | Mid | | 132 | 525 | 15.04 | 1.83 | 8.23 |
| | High | | 198 | 459 | 22.93 | 1.51 | 15.23 |
| Computers [46] | Low | 138 | 14 | 124 | 58.21 | 21.85 | 2.66 |
| | Mid | | 28 | 110 | 99.77 | 20.23 | 4.93 |
| | High | | 42 | 96 | 133.14 | 12.98 | 10.26 |
| Photo [46] | Low | 96 | 10 | 86 | 48.96 | 19.56 | 2.50 |
| | Mid | | 20 | 76 | 84.03 | 17.50 | 4.80 |
| | High | | 30 | 66 | 116.33 | 13.13 | 8.85 |
| ogbn-arXiv [14] | Low | 423 | 42 | 381 | 21.56 | 8.22 | 2.62 |
| | Mid | | 84 | 339 | 40.75 | 5.10 | 7.99 |
| | High | | 126 | 297 | 56.08 | 3.80 | 14.74 |
| Chameleon [44] | Low | 46 | 5 | 41 | 51.28 | 12.12 | 4.23 |
| | Mid | | 10 | 36 | 108.28 | 12.20 | 8.88 |
| | High | | 15 | 31 | 159.44 | 9.29 | 17.16 |
| Squirrel [44] | Low | 104 | 11 | 93 | 162.02 | 22.35 | 7.24 |
| | Mid | | 22 | 82 | 328.33 | 21.71 | 15.13 |
| | High | | 33 | 71 | 496.67 | 14.75 | 33.67 |
| Actor [39] | Low | 152 | 15 | 137 | 15.24 | 4.66 | 3.27 |
| | Mid | | 30 | 122 | 26.77 | 4.11 | 6.52 |
| | High | | 45 | 107 | 37.93 | 3.02 | 12.56 |

## B.3 MANIPULATED LOCAL TOPOLOGY-IMBALANCED DATASETS FOR NODE CLASSIFICATION

**Dataset Settings.** We conduct the semi-supervised node classification task on **9** manipulated locally topology-imbalanced datasets. The datasets are split into training, validation, and test sets with a ratio of 1:1:8. Local topology-imbalance is characterized by a long-tailed distribution in terms of node degree. Following the Pareto principle (the 20/80 rule) (Sanders, 1987), we designate the top 20% of nodes by degree as high-degree (head) nodes, and the remaining 80% as low-degree (tail) nodes. High-degree nodes benefit from more abundant structural information with superior performance in downstream tasks, while low-degree nodes suffer from limited topological information, which hinders their performance. To evaluate local topology-imbalance, we randomly select training and validation nodes according to the pre-defined splitting ratio (10%/10%) while ensuring an equal number of nodes per class for fairness. The remaining nodes are used for testing. To thoroughly assess the performance of the IGL algorithms, we create training sets with different imbalance ratios, as defined in Equation B.2. These ratios depend on the proportion of nodes selected from the head and tail sets. We repeat the node selection process multiple times, calculate the resulting imbalance ratios, and choose three groups of splits exhibiting significant variations in the imbalance ratio. These groups are categorized as **Low**, **Mid**, and **High**, based on their respective levels of local topology imbalance.

**Dataset Preview.** We conclude the statistics of the manipulated local topology-imbalanced datasets (training) for node classification in Table B.3. To guarantee a fair evaluation, we ensure the number of nodes for each class is equal (class-balanced). We also observe that the imbalance ratio corresponding to **Low**, **Mid**, and **High** roughly **doubles**, which can better simulate the various degrees of the imbalanced distribution of node degree.

Table B.4: Statistics of the manipulated global topology-imbalanced datasets (**training**) for node classification. The number of nodes for each class is equal (class-balanced), and the imbalance ratio $\rho$ is the 10x negative logarithm of the absolute value of the product of $RC$ and $SC$.

| Imbalance Level | Split | Dataset | Cora [65] | CiteSeer [65] | PubMed [65] | Computers [46] | Photo [46] | Chameleon [44] | Squirrel [44] | Actor [39] |
|---|---|---|---|---|---|---|---|---|---|---|
| Low | 1 | $RC$ | 0.60 | 0.84 | 0.62 | 0.73 | 0.54 | 0.49 | 0.46 | 0.56 |
| | | $SC$ | -0.62 | -0.41 | -0.81 | -0.87 | -0.63 | -0.66 | -0.53 | -0.71 |
| | | $\rho$ | **4.23** | **4.62** | **3.02** | **1.99** | **4.70** | **4.87** | **6.13** | **4.01** |
| | 2 | $RC$ | 0.60 | 0.84 | 0.61 | 0.72 | 0.54 | 0.49 | 0.45 | 0.56 |
| | | $SC$ | -0.62 | -0.41 | -0.81 | -0.87 | -0.63 | -0.67 | -0.53 | -0.71 |
| | | $\rho$ | **4.26** | **4.66** | **3.03** | **1.99** | **4.71** | **4.87** | **6.22** | **4.03** |
| | 3 | $RC$ | 0.60 | 0.84 | 0.61 | 0.72 | 0.54 | 0.48 | 0.45 | 0.56 |
| | | $SC$ | -0.62 | -0.41 | -0.81 | -0.87 | -0.63 | -0.67 | -0.53 | -0.71 |
| | | $\rho$ | **4.26** | **4.69** | **3.03** | **2.00** | **4.72** | **4.93** | **6.23** | **4.02** |
| | 4 | $RC$ | 0.60 | 0.83 | 0.61 | 0.72 | 0.54 | 0.48 | 0.45 | 0.56 |
| | | $SC$ | -0.62 | -0.41 | -0.81 | -0.87 | -0.63 | -0.67 | -0.54 | -0.70 |
| | | $\rho$ | **4.31** | **4.75** | **3.04** | **2.01** | **4.73** | **4.96** | **6.21** | **4.08** |
| | 5 | $RC$ | 0.59 | 0.83 | 0.61 | 0.72 | 0.54 | 0.46 | 0.43 | 0.55 |
| | | $SC$ | -0.62 | -0.40 | -0.81 | -0.87 | -0.63 | -0.67 | -0.53 | -0.71 |
| | | $\rho$ | **4.35** | **4.77** | **3.04** | **2.02** | **4.73** | **5.11** | **6.41** | **4.10** |
| Mid | 6 | $RC$ | 0.59 | 0.81 | 0.61 | 0.72 | 0.54 | 0.46 | 0.43 | 0.55 |
| | | $SC$ | -0.62 | -0.41 | -0.81 | -0.87 | -0.63 | -0.67 | -0.54 | -0.70 |
| | | $\rho$ | **4.35** | **4.80** | **3.05** | **2.03** | **4.74** | **5.11** | **6.36** | **4.16** |
| | 7 | $RC$ | 0.59 | 0.81 | 0.61 | 0.72 | 0.54 | 0.45 | 0.41 | 0.54 |
| | | $SC$ | -0.62 | -0.40 | -0.81 | -0.87 | -0.62 | -0.67 | -0.53 | -0.70 |
| | | $\rho$ | **4.35** | **4.87** | **3.08** | **2.04** | **4.75** | **5.20** | **6.56** | **4.16** |
| | 8 | $RC$ | 0.58 | 0.80 | 0.60 | 0.72 | 0.54 | 0.44 | 0.41 | 0.54 |
| | | $SC$ | -0.63 | -0.40 | -0.81 | -0.87 | -0.62 | -0.67 | -0.54 | -0.70 |
| | | $\rho$ | **4.38** | **4.94** | **3.11** | **2.04** | **4.76** | **5.32** | **6.63** | **4.19** |
| | 9 | $RC$ | 0.58 | 0.80 | 0.60 | 0.72 | 0.54 | 0.42 | 0.40 | 0.54 |
| | | $SC$ | -0.63 | -0.40 | -0.81 | -0.87 | -0.62 | -0.67 | -0.53 | -0.70 |
| | | $\rho$ | **4.39** | **4.96** | **3.12** | **2.05** | **4.77** | **5.47** | **6.73** | **4.20** |
| High | 10 | $RC$ | 0.58 | 0.80 | 0.59 | 0.72 | 0.53 | 0.41 | 0.41 | 0.54 |
| | | $SC$ | -0.62 | -0.39 | -0.81 | -0.87 | -0.62 | -0.68 | -0.52 | -0.69 |
| | | $\rho$ | **4.43** | **5.05** | **3.16** | **2.06** | **4.77** | **5.51** | **6.79** | **4.24** |

## B.4 MANIPULATED GLOBAL TOPOLOGY-IMBALANCED DATASETS FOR NODE CLASSIFICATION

**Dataset Settings.** We conduct the semi-supervised node classification task on **8** manipulated globally topology-imbalanced datasets. We select 10% nodes for training and 10% nodes for validation. For a fair comparison, we assign the same number of nodes for each class to guarantee the class-balance when evaluating the global topology-imbalance issue. The remaining nodes are used for testing. The global topology-imbalance issue is facilitated by both the under-reaching and over-squashing phenomenon, which are quantified with the metrics of the Reaching Coefficient ($RC$) and the Squashing Coefficient ($SC$). Considering that $RC$ and $SC$ reflect two aspects of the causes of global topology-imbalance simultaneously, and both variables change monotonically, the negative logarithm of their product is used to define the imbalance ratio according to Equation B.3 (since $RC$ is positive and $SC$ is negative, the purpose of 10x and taking the negative logarithm is to amplify the observable variation of the imbalance ratio). Note that, larger $RC$ means better reachability and larger $SC$ means lower squashing. Consequently, the lower the degree of global topology-imbalance ratio. We randomly generate 100 groups of training splits and calculate the imbalance ratio for each. We select 10 groups with the minimum, maximum, and uniformly varying imbalance ratios within the range to simulate the change in the degree of global topology imbalance from **High** to **Low**.

**Dataset Preview.** We conclude the statistics of the manipulated local topology-imbalanced datasets (training) for node classification in Table B.4. To guarantee a fair evaluation, we ensure the number of nodes for each class is equal (class-balanced). It can be observed that as the imbalance degree increases from low to high, the imbalance ratio also increases from small to large.

Table B.5: Statistics of the manipulated class-imbalanced datasets for graph classification. The imbalance ratio for the graph-level class-imbalance problem is set to the ratio between the number of graphs in the majority and the number of graphs in the minority class. The number of graphs for each class is equal in the validation set for fair evaluation.

| Dataset | Task | Level | #Graphs (val.) per Class | #Graphs (Majority Class) | #Graphs (Minority Class) | Imbalance Ratio $\rho$ |
|---|---|---|---|---|---|---|
| PTC-MR [1] | Binary | Balanced | | 17 | 17 | 1.0 (5:5) |
| | | Low | 17 | 23 | 11 | 2.3 (7:3) |
| | | High | | 30 | 4 | 9.0 (9:1) |
| FRANKENSTEIN [1] | Binary | Balanced | | 216 | 216 | 1.0 (5:5) |
| | | Low | 216 | 302 | 130 | 2.3 (7:3) |
| | | High | | 388 | 44 | 9.0 (9:1) |
| PROTEINS [2] | Binary | Balanced | | 55 | 55 | 1.0 (5:5) |
| | | Low | 55 | 77 | 33 | 2.3 (7:3) |
| | | High | | 99 | 11 | 9.0 (9:1) |
| D&D [47] | Binary | Balanced | | 58 | 58 | 1.0 (5:5) |
| | | Low | 58 | 80 | 36 | 2.3 (7:3) |
| | | High | | 104 | 12 | 9.0 (9:1) |
| IMDB-B [5] | Binary | Balanced | | 50 | 50 | 1.0 (5:5) |
| | | Low | 50 | 70 | 30 | 2.3 (7:3) |
| | | High | | 90 | 10 | 9.0 (9:1) |
| REDDIT-B [64] | Binary | Balanced | | 100 | 100 | 1.0 (5:5) |
| | | Low | 100 | 140 | 60 | 2.3 (7:3) |
| | | High | | 180 | 20 | 9.0 (9:1) |
| ogbg-molhiv [14] | Binary | — | 400 | 38,884 | 643 | 60.4 |
| COLLAB [22] | Multi-Class | Balanced | | 167 | 167 | 1 |
| | | Low | 167 | 380 | 19 | 20 |
| | | High | | 400 | 4 | 100 |

## B.5 MANIPULATED CLASS-IMBALANCED DATASETS FOR GRAPH CLASSIFICATION

**Dataset Settings.** We conduct the graph classification task on the **7** manipulated class-imbalanced graph datasets, which are split into training, validation, and test sets with a ratio of 1:1:8. We also evaluate IGL algorithms on the **naturally imbalanced** *ogbg-molhiv* dataset, consisting of a large number of graphs. Our manipulations involve three different types of processing methods. For ❶ **balanced datasets with binary classification**, we randomly sample 10%/10% graphs for training and validation, and the rest are for testing to ensure the sufficiency of the minority class instances in both training and validation set given the skewed imitative data distribution. According to Equation B.1, the imbalance ratio for the graph-level class-imbalance problem is set to the ratio between the number of graphs in the majority and the number of graphs in the minority class. To construct graph datasets with different imbalance ratios, we select the class with a larger number of graphs as the majority class, and the remaining class as the minority class. We then create training datasets with different imbalance ratios by adjusting the training sample ratios to 9:1 ($\rho = 9.0$), 7:3 ($\rho = 2.3$), and 5:5 ($\rho = 1.0$, **class-balanced**), while ensuring that the number of training samples constitutes 10% of the total. In the validation set, an equal number of samples are allocated for each class for fairness. All remaining samples are then assigned to the test set. For ❷ **imbalanced dataset with binary classification**, to make validation/test sets balanced, we sample the same number of graphs from each class for validation/test sets. Then, the remaining graphs are assigned to the training set. For ❸ **multi-class classification**, situations are similar to manipulations defined in Section B.2. We hypothesize that the number of graphs in each class within the training dataset multiplies exponentially. Given the total number of graphs in the training dataset and $\rho$, the number of graphs allocated for training in each class can be determined with certainty. Any graphs not allocated for training or validation are assigned to the test set. For a thorough performance evaluation, we consider three scenarios within `IGL-Bench`: a class-balanced scenario ($\rho = 1$), a class-imbalanced scenario ($\rho = 20$), and an extreme class-imbalanced scenario ($\rho = 100$).

**Dataset Preview.** We conclude the statistics of the manipulated class-imbalanced datasets for graph classification in Table B.5. It can be observed that the constructed datasets can not only evaluate the ideal class-balanced ($\rho = 1$) scenario but also comprehensively assess the performance of IGL algorithms under the general class-imbalanced and extremely class-imbalanced conditions.

Table B.6: Statistics of the manipulated topology-imbalanced datasets (**training**) for graph classification. The number of nodes for each class is equal (class-balanced), and the imbalance ratio $\rho$ is the ratio between the average size of the head graphs and the average size of the tail graphs.

| Dataset | Level | #Graphs per Class | #Head Graphs per Class | #Tail Graphs per Class | Avg. #Size (Head Graphs) | Avg. #Size (Tail Graphs) | Imbalance Ratio $\rho$ |
|---|---|---|---|---|---|---|---|
| PTC-MR [1] | Low | 17 | 2 | 15 | 29.50 | 18.40 | 1.60 |
| | Mid | | | | 34.25 | 11.80 | 2.90 |
| | High | | | | 56.00 | 11.37 | 4.93 |
| FRANKENSTEIN [1] | Low | 217 | 22 | 195 | 33.09 | 21.19 | 1.56 |
| | Mid | | | | 32.43 | 13.78 | 2.35 |
| | High | | | | 77.55 | 13.78 | 5.63 |
| PROTEINS [2] | Low | 56 | 6 | 50 | 93.50 | 45.81 | 2.04 |
| | Mid | | | | 90.25 | 22.26 | 4.05 |
| | High | | | | 276.00 | 22.76 | 12.13 |
| D&D [47] | Low | 59 | 6 | 53 | 548.00 | 350.58 | 1.56 |
| | Mid | | | | 524.83 | 207.70 | 2.53 |
| | High | | | | 1765.00 | 198.11 | 8.91 |
| IMDB-B [5] | Low | 50 | 5 | 45 | 37.50 | 22.92 | 1.64 |
| | Mid | | | | 34.30 | 16.29 | 2.11 |
| | High | | | | 70.30 | 15.81 | 4.45 |
| REDDIT-B [64] | Low | 100 | 10 | 90 | 1180.75 | 423.23 | 2.79 |
| | Mid | | | | 1097.75 | 223.04 | 4.92 |
| | High | | | | 2442.70 | 222.73 | 10.97 |
| ogbg-molhiv [14] | Low | 200 | 20 | 180 | 37.69 | 23.12 | 1.63 |
| | Mid | | | | 41.87 | 15.86 | 2.64 |
| | High | | | | 89.30 | 15.64 | 5.71 |
| COLLAB [22] | Low | 167 | 17 | 150 | 141.41 | 80.14 | 1.76 |
| | Mid | | | | 147.22 | 50.04 | 2.94 |
| | High | | | | 309.41 | 50.77 | 6.09 |

## B.6 MANIPULATED TOPOLOGY-IMBALANCED DATASETS FOR GRAPH CLASSIFICATION

**Dataset Settings.** We conduct the graph classification task on the **7** manipulated topology-imbalanced graph datasets. For 6 class-balanced datasets, we divide them into training, validation, and test sets with a ratio of 1:1:8. To ensure fairness, we maintain an equal number of graphs per class within each set, achieving a class-balanced scenario. For the naturally imbalanced ogbg-molhiv dataset, considering that one category contains a small number of graphs, we randomly sample a specified number of graphs from each category for training and validation, reserving the remainder for testing. Equation B.6 defines the imbalance ratio to be the ratio of the average graph size in the head graph set to the average graph size in the tail graph set. Specifically, the head graph set consists of the top 20% of graphs in terms of size (measured by the number of nodes each graph contains), while the remaining 80% comprise the tail graph set (Sanders, 1987). Typically, larger graphs are more expressive due to their complex structures and richer information content. This expressiveness often translates to improved performance in graph classification tasks compared to smaller graphs. However, this advantage can also introduce biases in applications such as molecular or protein prediction, where larger graphs might inherently contain more predictive features, overshadowing the smaller graphs. We create training datasets with varying degrees of imbalance. The degree of imbalance is manipulated by altering the proportion of graphs selected from the head and tail sets. We select graphs multiple times, each time computing the resulting imbalance ratios. From these computations, we identify three distinct sets of splits that exhibit significant variations in imbalance levels. These sets are categorized and labeled as **Low**, **Mid**, and **High** to reflect their respective levels of local topology imbalance. By systematically varying the imbalance levels, we aim to simulate diverse real-world scenarios. This approach allows us to rigorously test the robustness and adaptability of IGL algorithms under different degrees of topology imbalance. Ultimately, this comprehensive evaluation provides a deeper understanding of the performance of IGL algorithms across datasets with varying characteristics, highlighting their strengths and potential areas for improvement.

**Dataset Preview.** We conclude the statistics of the manipulated class-imbalanced datasets for graph classification in Table B.5. To guarantee a fair evaluation, we ensure the number of graphs for each class is equal (class-balanced). We also observe that the imbalance ratio corresponds to Low, Mid, and High roughly **doubles**, which can better simulate the various degrees of the imbalanced distribution of graph sizes.

Table C.1: Different evaluation metrics for each algorithm in original papers.

| Algorithm | Accuracy | Balanced Accuracy | Macro-F1 | AUC-ROC | Others |
|---|---|---|---|---|---|
| DRGCN [48] | ✓ | | | ✓ | |
| DPGNN [57] | | | ✓ | | Weighted-F1, Micro-F1 |
| ImGAGN [42] | | | | ✓ | Precision, Recall |
| GraphSMOTE [70] | ✓ | | ✓ | ✓ | |
| GraphENS [37] | ✓ | ✓ | ✓ | | |
| GraphMixup [60] | ✓ | | ✓ | ✓ | |
| LTE4G [67] | | ✓ | ✓ | | G-Means |
| TAM [49] | | ✓ | ✓ | | |
| TOPOAUC [8] | | | ✓ | ✓ | Weighted-F1 |
| GraphSHA [24] | ✓ | ✓ | ✓ | | |
| DEMO-NET [59] | ✓ | | | | |
| meta-tail2vc [30] | ✓ | | | | Micro-F1 |
| Tail-GNN [31] | ✓ | | | | Micro-F1 |
| Cold Brew [72] | ✓ | | | | |
| RawlsGCN [19] | ✓ | | | | Bias |
| GraphPatcher [17] | ✓ | | | | |
| ReNode [7] | | | ✓ | | Weighted-F1 |
| PASTEL [52] | | | ✓ | | Weighted-F1 |
| HyperIMBA [12] | | | | | Weighted-F1, Micro-F1 |
| G$^2$GNN [58] | | | ✓ | | Micro-F1 |
| Topolmb [71] | | | ✓ | ✓ | |
| DataDec [68] | | | ✓ | | Micro-F1 |
| ImGKB [54] | | | ✓ | ✓ | Recall |
| SOLT-GNN [32] | ✓ | | | | |

## C   DETAILS OF THE EXPERIMENTAL SETTINGS

### C.1   GENERAL EXPERIMENTAL CONFIGURATIONS

The number of training epochs for optimizing all IGL algorithms is set to 1000. We adopt the early stopping strategy, *i.e.*, stop training if the performance on the validation set does not improve for 50 epochs. All parameters are randomly initiated. We adopt Adam (Kingma & Ba, 2015) with an appropriate learning rate and weight decay for the best performance on the validation split. We randomly run all the experiments 10 times, and report the average results with standard deviations.

### C.2   EVALUATION METRICS

To perform an unbiased evaluation, we summarize all the metrics used in the original papers of the algorithms in Table C.1. We can conclude that, Accuracy (Acc.), Balanced Accuracy (bAcc.), Macro-F1, and AUC-ROC are four commonly used metrics for evaluating imbalanced graph learning performance. Though metrics like Weighted-F1 and Micro-F1 are also popular for evaluation, their difference lies in considering the imbalance between classes. However, this aligns similarly with the difference between Accuracy and Balanced Accuracy, so we have not taken metrics like Weighted-F1 and Micro-F1 into the evaluation. However, as we have saved the weights for each algorithm under all experiment settings, it is easy to include more metrics in our benchmark in a short updating time for all algorithms under various experiment settings.

We briefly introduce and analyze the evaluation metrics employed to assess the performance of IGL algorithms including Accuracy (Acc.), Balanced Accuracy (bAcc.), Macro-F1, and AUC-ROC.

**Accuracy** (Kipf & Welling, 2016). It reflects the ratio of correctly predicted instances to the total number of instances. It is formally defined as:

$$\text{Accuracy} = \frac{TP + TN}{TP + TN + FP + FN},\tag{C.1}$$

where $TP$ denotes true positives, $TN$ denotes true negatives, $FP$ denotes false positives, and $FN$ denotes false negatives. ❶ **Advantages:** Accuracy is simple and ease of interpretation. Further, it provides an immediate, overall performance measure of the algorithm. ❷ **Disadvantages:** In the imbalanced datasets, Accuracy can be misleading as it tends to favor the majority class, and fails to account for the distribution of classes, underrepresenting the performance of minority classes.

**Balanced Accuracy** (Brodersen et al., 2010). Balanced Accuracy adjusts the conventional Accuracy to account for class imbalance. It is the average of recall obtained in each class. For multi-class classification, it is defined as:

$$\text{Balanced Accuracy} = \frac{1}{N} \sum_{i=1}^{N} \frac{TP_i}{TP_i + FN_i},\tag{C.2}$$

where $N$ is the number of classes. ❶ **Advantages:** Accuracy accounts for class imbalance, providing a more equitable evaluation, and it reflects performance across all classes more accurately than standard accuracy. ❷ **Disadvantages:** May be sensitive to noise and outliers, particularly in minority classes. In addition, it is potentially less intuitive to interpret compared to simple accuracy.

**Macro-F1** (Xia et al., 2014). The Macro-F1 score is the harmonic mean of precision and recall, calculated independently for each class and then averaged. It is expressed as:

$$\text{Macro-F1} = \frac{1}{N} \sum_{i=1}^{N} \frac{2 \cdot \text{Precision}_i \cdot \text{Recall}_i}{\text{Precision}_i + \text{Recall}_i},\tag{C.3}$$

where $\text{Precision}_i = \frac{TP_i}{TP_i + FP_i}$ and $\text{Recall}_i = \frac{TP_i}{TP_i + FN_i}$. ❶ **Advantages:** Macro-F1 emphasizes both precision and recall, ensuring consideration of both false positives and false negatives. Moreover, it provides a balanced view of the classification performance across all classes. ❷ **Disadvantages:** It can be disproportionately affected by very small classes and does not account for the prevalence of different classes.

**AUC-ROC** (Bradley, 1997). AUC-ROC (Area Under the Receiver Operating Characteristic Curve) measures the area under the ROC curve, which plots the true positive rate (recall) against the false positive rate (fall-out) at various threshold settings. For binary classification, it is defined as:

$$\text{AUC-ROC} = \int_0^1 \text{ROC}(t) \, \mathrm{d}t.\tag{C.4}$$

For multi-class problems, an average of the AUC-ROC scores for each class against the rest can be employed. ❶ **Advantages:** AUC-ROC evaluates the algorithm's performance across all possible classification thresholds. ❷ **Disadvantages:** It is computationally intensive, particularly for large datasets. Further, it does not provide a clear threshold for decision-making, focusing instead on overall ranking performance.

**Analysis.** When evaluating node-level and graph-level tasks under class-imbalance and topology-imbalance conditions, selecting the appropriate evaluation metric is crucial. ❶ Accuracy is often unsuitable for imbalanced scenarios due to its tendency to favor the majority class, potentially providing a false sense of model performance when minority classes are present. ❷ Balanced Accuracy and Macro-F1 are more appropriate for imbalanced datasets as they offer a more equitable assessment of performance across classes. Macro-F1, in particular, is informative in tasks where both precision and recall are critical. ❸ AUC-ROC is advantageous in ranking-based scenarios and for evaluating models across different thresholds. Its robustness to class imbalance is beneficial, though its interpretation can be less straightforward in multi-class problems.

In summary, while no single metric is universally optimal, a combination of these metrics can provide a comprehensive evaluation of imbalanced graph learning algorithms. Accuracy offers a general overview, while Balanced Accuracy and Macro-F1 provide insights into class-specific performance. AUC-ROC, on the other hand, offers a threshold-independent evaluation, particularly useful in highly imbalanced scenarios.

Table C.2: Hyperparameter search space for **node**-level **class**-imbalanced IGL algorithms.

| Algorithm | Hyperparameter | Search Space |
|---|---|---|
| General Settings | dropout
weight decay
number of max training epochs
learning rate | 0.2, 0.3, 0.4, 0.5, 0.6
0, 5e–6, 5e–5, 5e–4, 5e–3
500, 1000, 2000
0.005, 0.0075, 0.01, 0.015 |
| GCN [21] | number of layers
hidden size | 1, 2, 3
32, 64, 128 |
| DRGCN [48] | $\alpha$ for loss trade-off | 0.5, 0.6, 0.7, 0.8, 0.9 |
| DPGNN [57] | $\lambda_1$ for $\mathcal{L}_{\text{ssl}_p}$
$\lambda_2$ for $\mathcal{L}_{\text{ssl}_s}$
threshold $\eta$ for the hard pseudo label | 1, 10
1, 10
0, 1, 2, 3, 4, 5, 6 |
| ImGAGN [42] | $\lambda_1$ for minority nodes ratio
$\lambda_2$ for discriminator training steps | 0.1, 0.2, 0.3, 0.4, 0.5, 0.6
20, 30, 40, 50, 60 |
| GraphSMOTE [70] | $\lambda$ for $\mathcal{L}_{edge}$ | 1e–6, 2e–6, 4e–6 |
| GraphENS [37] | number of warming up epochs
$k$ for feature masking
$\tau$ for temperature | 1, 5
1, 5, 10
1, 2 |
| LTE4G [67] | $\alpha$ for the focal loss
$\gamma$ for curve shape controlling | 0.1, 0.2, 0.3, 0.4, 0.5, 0.6, 0.7, 0.8, 0.9
0, 1, 2 |
| TAM [49] | $\phi$ for the class–wise temperature
$\alpha$ for the ACM term of node $v$
$\beta$ for the ADM term of node $v$
the base model | 0.8, 1.2
0.25, 0.5, 1.5, 2.5
0.125, 0.25, 0.5
GraphENS [37], ReNode [7] |
| GraphSHA [24] | sampled $\beta$-distribution | $\beta(1,100)$, $\beta(1,10)$ |

## C.3 HYPERPARAMETER

We meticulously optimize hyperparameters to guarantee a rigorous and unbiased assessment of the integrated IGL methods. In cases where the original paper or source code for a specific algorithm lacks guidance on hyperparameter selection, we perform the hyperparameter tuning through Bayesian search on the Weights & Biases (wandb) platform[4]. The hyperparameter search space for all IGL algorithms is detailed in Table C.2 (for node-level class-imbalanced IGL algorithms), Table C.3 (for node-level topology-imbalanced IGL algorithms), and Table C.4 (for graph-level class-imbalanced and topology-imbalanced IGL algorithms). For interpretations of these hyperparameters, please consult the respective papers. More detailed and comprehensive hyperparameter configurations for all algorithms are accessible within our publicly released GitHub package.

---

[4]https://wandb.ai/

Table C.3: Hyperparameter search space for **node**-level **topology**-imbalanced IGL algorithms.

| Algorithm | Hyperparameter | Search Space |
|---|---|---|
| General Settings | dropout
weight decay
number of max training epochs
learning rate | 0.2, 0.3, 0.4, 0.5, 0.6
0, 5e–6, 5e–5, 5e–4, 5e–3
500, 1000, 2000
0.005, 0.0075, 0.01, 0.015, |
| GCN [21] | number of layers
hidden size | 1, 2, 3
32, 64, 128 |
| Tail-GNN [31] | $\mu$ for $\mathcal{L}_m$
$\eta$ for $\mathcal{L}_d$ | 0.01, 0.001
0.1, 1.0 |
| Cold Brew [72] | $\alpha$ for mixing coefficient
number of propagations | 0.01, 0.1, 0.5, 0.9, 0.99
10, 20, 50, 100, 200 |
| LTE4G [67] | $\alpha$ for the focal loss
$\gamma$ for curve shape controlling | 0.1, 0.2, 0.3, 0.4, 0.5, 0.6, 0.7, 0.8, 0.9
0, 1, 2 |
| RawlsGCN [19] | $\alpha$ for probability scalar | 0.1, 0.2, 0.3, 0.4, 0.5, 0.6, 0.7, 0.8, 0.9 |
| GraphPatcher [17] | batch size
number of accumulation steps
number of patching steps
augmentation length | 4, 8, 16, 64
16, 32, 64
3, 4, 5
0.1, 0.2, 0.3 |
| ReNode [7] | PageRank teleport probability
lower bound of reweighting
upper bound of reweighting | 0.05, 0.1, 0.15, 0.2
0.25, 0.5, 0.75
1.25, 1.5, 1.75 |
| TAM [49] | $\phi$ for the class–wise temperature
$\alpha$ for the ACM term of node v
$\beta$ for the ADM term of node v | 0.8, 1.2
0.25, 0.5, 1.5, 2.5
0.125, 0.25, 0.5 |
| PASTEL [52] | $\lambda_1$ for structure mixing
$\lambda_2$ for structure mixing | 0.7, 0.8, 0.9
0.7, 0.8, 0.9 |

Table C.4: Hyperparameter search space for both the **graph**-level class-imbalanced and topology-imbalanced IGL algorithms.

| Algorithm | Hyperparameter | Search Space |
|---|---|---|
| General Settings | dropout
weight decay
number of max training epochs
learning rate | 0.2, 0.3, 0.4, 0.5, 0.6
0, 5e–6, 5e–5, 5e–4, 5e–3
500, 1000, 2000
0.001,0.005,0.01,0.0125,0.05 |
| GCN [21], GIN [63] | number of layers
hidden size | 2, 3, 4
32, 64, 128 |
| G$^2$GNN [58] | $k$ for the number of neighboring graphs
drop edge ratio
mask node ratio | 1, 2, 3
5e–5,1e–4,5e–4,1e–3,5e–5
5e–5,1e–4,5e–4,1e–3,5e–5 |
| TopoImb [71] | $\alpha$ for $\mathcal{L}_{RE}$ | 0.2, 0.3, 0.4, 0.5, 0.6 |
| ImGKB [54] | $\beta$ for compression coefficient
$k$ for the number of neighboring graphs | 0.3, 0.4, 0.5, 0.6
2, 4, 6, 8, 10 |
| SOLT-GNN [32] | $\alpha$ for loss trade-off
$\mu_1$ for $\mathcal{L}_{rel}^{node}$
$\mu_2$ for $\mathcal{L}_{rel}^{subg}$ | 0.1, 0.15, 0.3
0, 0.5, 1, 1.5, 2
0, 0.5, 1, 1.5, 2 |

## C.4 Computation Resouces

We conduct the experiments with the following resources and configurations:

- Operating System: Ubuntu 20.04 LTS.
- CPU: Intel(R) Xeon(R) Platinum 8358 CPU@2.60GHz with 1TB DDR4 of Memory.
- GPU: NVIDIA Tesla A100 SMX4 with 40GB of Memory.
- Software: CUDA 10.1, Python 3.8.12, PyTorch (Paszke et al., 2019) 1.9.1, PyTorch Geometric (Fey & Lenssen, 2019) 2.0.1.

# D   Additional Experimental Results

## D.1   Additional Results for Algorithm Effectiveness (**RQ1**)

### D.1.1   Effectiveness of Node-Level Class-Imbalanced Algorithms

Table D.1: **Accuracy** score (% ± standard deviation) of **node** classification on manipulated **class-imbalanced** graph datasets with changing imbalance levels over 10 runs. "—" denotes out of memory or time limit. The best results are shown in **bold** and the runner-ups are underlined.

| Algorithm | Cora 0.81 | CiteSeer 0.74 | PubMed 0.80 | Computers 0.78 | Photo 0.82 | ogbn-arXiv 0.65 | Chameleon 0.23 | Squirrel 0.22 | Actor 0.22 |
|---|---|---|---|---|---|---|---|---|---|
| $\rho = 1$ (Balanced) | | | | | | | | | |
| GCN (bb.) [21] | 80.41±0.78 | 66.39±0.86 | 82.88±0.15 | 79.29±0.66 | 88.30±0.56 | 45.92±0.48 | 26.76±1.89 | 21.46±0.87 | 23.07±0.38 |
| DRGCN [48] | 77.78±1.22 | 66.99±1.70 | 81.37±2.96 | 63.71±26.04 | 85.67±0.93 | — | 33.81±1.64 | 24.90±0.48 | 24.19±1.55 |
| DPGNN [57] | 74.20±2.47 | 62.07±3.03 | 80.96±3.09 | 66.15±11.96 | 87.58±2.77 | — | 32.06±2.41 | 25.00±1.32 | 22.49±2.80 |
| ImGAGN [42] | 80.58±0.65 | 66.27±0.60 | 82.78±0.11 | 76.67±1.15 | 86.62±0.37 | — | 29.19±2.26 | 21.61±0.84 | 22.55±0.94 |
| GraphSMOTE [70] | 78.92±0.48 | 65.50±0.42 | 81.85±0.19 | 79.46±0.60 | 86.89±0.66 | — | 25.05±2.01 | 21.32±0.22 | 25.96±0.31 |
| GraphENS [37] | 80.41±0.78 | 66.42±0.86 | 82.87±0.15 | 78.71±0.98 | 88.63±1.44 | 46.68±0.68 | 26.76±1.89 | 21.46±0.87 | 23.07±0.38 |
| GraphMixup [60] | 79.30±0.64 | 69.95±1.09 | 83.85±0.11 | **82.16±1.18** | **90.56±0.35** | 43.88±1.02 | 35.05±0.34 | 24.59±0.32 | 24.29±1.16 |
| LTE4G [67] | 80.48±1.12 | 67.77±2.25 | **84.27±0.30** | 74.23±4.72 | 88.48±3.83 | — | 35.71±0.53 | 24.62±0.47 | 24.88±1.11 |
| TAM [49] | 81.33±0.62 | 66.26±0.52 | 74.56±0.78 | 78.76±0.78 | 88.49±1.57 | 46.66±0.57 | 28.56±1.24 | 21.51±0.94 | 23.54±0.50 |
| TOPOAUC [8] | **83.69±0.32** | **73.41±0.46** | — | 69.79±3.93 | 82.85±2.33 | — | **37.14±0.95** | **25.24±0.46** | **26.25±1.22** |
| GraphSHA [24] | 80.41±0.78 | 66.40±0.85 | 82.87±0.14 | 78.88±0.88 | 88.61±4.99 | **47.32±0.39** | 26.76±1.89 | 21.46±0.87 | 23.07±0.38 |
| $\rho = 20$ (Low) | | | | | | | | | |
| GCN (bb.) [21] | 76.36±0.13 | 52.96±0.55 | 60.57±0.19 | 75.06±0.50 | 69.80±6.15 | 59.83±0.23 | 26.35±0.24 | 17.16±0.17 | 24.06±0.14 |
| DRGCN [48] | 71.35±0.77 | 55.22±1.82 | 62.59±4.62 | 67.71±3.10 | 85.67±5.30 | — | 26.40±0.35 | 17.11±0.81 | 25.03±0.23 |
| DPGNN [57] | 72.91±3.95 | 56.78±2.23 | 81.87±2.80 | 68.69±8.62 | 81.66±9.19 | — | 30.58±1.48 | **25.35±1.48** | 21.66±1.68 |
| ImGAGN [42] | 73.48±3.07 | 55.29±3.00 | 72.16±1.51 | 74.92±1.87 | 83.10±3.42 | — | 24.38±2.86 | 18.75±1.80 | 24.54±3.38 |
| GraphSMOTE [70] | 77.21±0.27 | 53.55±0.95 | 71.25±0.27 | 76.04±1.52 | 89.07±1.12 | — | 27.23±0.21 | 16.79±0.14 | 25.08±0.31 |
| GraphENS [37] | 79.34±0.49 | 61.98±0.76 | 80.84±0.17 | 80.72±0.68 | 90.38±0.37 | 53.23±0.52 | 24.34±1.62 | 20.05±1.61 | 25.03±0.38 |
| GraphMixup [60] | 79.88±0.43 | 62.66±0.70 | 75.94±0.09 | **86.15±0.47** | 89.69±0.31 | 56.08±0.31 | 30.95±0.40 | 17.83±0.32 | 24.75±0.37 |
| LTE4G [67] | 80.53±0.65 | 64.48±1.56 | 83.02±0.33 | 79.35±1.39 | 87.94±1.82 | — | 31.91±0.34 | 19.37±0.41 | **25.43±0.26** |
| TAM [49] | 80.69±0.27 | 64.16±0.24 | 81.47±0.15 | 81.30±0.53 | 90.35±0.42 | 53.49±0.54 | 23.27±1.38 | 21.17±0.95 | 24.53±0.33 |
| TOPOAUC [8] | **83.34±0.31** | **69.03±1.33** | — | 70.85±4.55 | 83.72±2.23 | — | **33.60±1.51** | 21.38±1.03 | 25.16±0.46 |
| GraphSHA [24] | 80.03±0.46 | 60.51±0.61 | 77.94±0.36 | 82.71±0.40 | **91.55±0.32** | **60.30±0.13** | 23.73±1.97 | 20.05±1.61 | 23.59±1.01 |
| $\rho = 100$ (High) | | | | | | | | | |
| GCN (bb.) [21] | 62.20±3.57 | 42.48±0.24 | 47.31±0.72 | 58.04±0.98 | 46.54±0.33 | **60.35±0.24** | 25.68±0.12 | 15.17±0.10 | 21.70±0.22 |
| DRGCN [48] | 61.99±2.46 | 45.69±2.79 | 49.80±4.33 | 66.02±1.48 | 73.58±5.44 | — | 25.79±0.44 | 15.32±0.43 | 23.03±0.59 |
| DPGNN [57] | 67.98±3.35 | 51.10±3.06 | 76.29±3.38 | 70.04±8.56 | 87.70±0.34 | — | 28.82±1.83 | **23.91±1.90** | 22.37±0.74 |
| ImGAGN [42] | 66.16±3.54 | 53.60±3.32 | 64.03±0.62 | 66.89±4.29 | 74.92±5.89 | — | 23.72±2.85 | 17.30±3.16 | 24.10±1.26 |
| GraphSMOTE [70] | 69.81±0.46 | 45.72±0.80 | 69.12±0.22 | 56.55±1.29 | 44.97±0.22 | — | 25.60±0.12 | 15.41±0.10 | 21.76±0.21 |
| GraphENS [37] | 77.68±0.58 | 62.85±0.72 | 76.69±0.31 | 80.99±0.76 | 90.31±0.33 | 54.13±0.49 | 26.26±2.42 | 20.65±2.30 | 20.67±2.47 |
| GraphMixup [60] | 70.01±0.50 | 49.63±0.28 | 63.47±0.08 | 79.34±0.42 | 73.02±4.01 | 57.40±0.35 | 26.41±0.08 | 15.75±0.16 | 23.39±0.37 |
| LTE4G [67] | 73.70±0.99 | 57.14±1.28 | 70.58±15.32 | 79.59±0.94 | 89.53±0.74 | — | 27.88±0.60 | 16.18±0.34 | 24.76±0.42 |
| TAM [49] | **79.36±0.56** | 64.30±0.46 | **80.53±0.18** | **85.77±0.41** | 90.28±0.32 | 54.25±0.70 | 23.47±1.73 | 23.48±1.24 | 21.92±0.18 |
| TOPOAUC [8] | 76.97±0.99 | **67.31±2.02** | — | — | 82.74±3.10 | — | **30.66±0.48** | 17.67±1.29 | **25.35±1.04** |
| GraphSHA [24] | 78.66±0.46 | 57.63±0.82 | 70.68±2.42 | 80.79±0.65 | **91.27±0.25** | 60.17±0.17 | 24.14±1.30 | 20.78±2.19 | 20.82±2.65 |

Table D.2: **Balanced Accuracy** score (% ± standard deviation) of **node** classification on manipulated **class-imbalanced** graph datasets with changing imbalance levels over 10 runs. "—" denotes out of memory or time limit. The best results are shown in **bold** and the runner-ups are underlined.

| Algorithm | Cora 0.81 | CiteSeer 0.74 | PubMed 0.80 | Computers 0.78 | Photo 0.82 | ogbn-arXiv 0.65 | Chameleon 0.23 | Squirrel 0.22 | Actor 0.22 |
|---|---|---|---|---|---|---|---|---|---|
| $\rho = 1$ (Balanced) | | | | | | | | | |
| GCN (bb.) [21] | 79.98±0.63 | 62.62±0.60 | 83.13±0.06 | 87.20±0.26 | 89.70±0.26 | 45.93±0.56 | 28.04±1.60 | 21.47±0.87 | 22.65±0.33 |
| DRGCN [48] | 77.20±0.78 | 63.05±0.82 | 82.22±2.10 | 73.23±23.82 | 88.22±0.71 | — | 34.36±1.20 | 24.89±0.48 | 23.33±0.34 |
| DPGNN [57] | 75.79±2.44 | 58.96±3.21 | 81.42±3.08 | 74.75±10.80 | 88.15±2.56 | — | 33.07±1.92 | 24.99±1.32 | 22.13±1.79 |
| ImGAGN [42] | 80.20±0.41 | 62.55±0.38 | 83.11±0.08 | 84.20±0.36 | 88.80±0.27 | — | 29.65±1.78 | 21.61±0.84 | 22.57±0.60 |
| GraphSMOTE [70] | 78.04±0.44 | 62.16±0.41 | 82.11±0.14 | 80.07±0.54 | 88.89±0.26 | — | 26.83±2.01 | 21.34±0.22 | 22.04±0.32 |
| GraphENS [37] | 79.98±0.63 | 62.65±0.60 | 83.12±0.08 | 86.26±0.51 | 89.85±0.64 | 45.98±0.49 | 28.04±1.60 | 21.47±0.87 | 22.65±0.33 |
| GraphMixup [60] | 82.41±0.24 | 67.48±1.01 | 84.72±0.07 | **88.96**±0.35 | **91.90**±0.24 | 43.97±0.49 | 35.60±0.33 | 24.59±0.32 | 23.50±0.52 |
| LTE4G [67] | 81.97±1.02 | 65.10±2.07 | **85.21**±0.13 | 83.59±2.06 | 88.83±3.48 | — | 35.13±0.53 | 24.61±0.47 | 24.88±0.73 |
| TAM [49] | 80.98±0.33 | 62.99±0.26 | 77.26±0.62 | 86.06±0.52 | 89.77±0.74 | 46.05±0.77 | 28.82±0.76 | 21.51±0.94 | 22.93±0.45 |
| TOPOAUC [8] | **84.86**±0.18 | **69.90**±0.42 | — | 77.23±1.73 | 85.24±1.32 | — | **37.92**±0.68 | **25.23**±0.46 | **25.68**±0.41 |
| GraphSHA [24] | 79.98±0.63 | 62.63±0.59 | 83.13±0.07 | 86.11±0.44 | 89.81±0.63 | **46.40**±0.83 | 28.04±1.60 | 21.47±0.87 | 22.65±0.33 |
| $\rho = 20$ (Low) | | | | | | | | | |
| GCN (bb.) [21] | 69.17±0.26 | 47.61±0.48 | 52.40±0.15 | 40.86±0.77 | 49.87±7.16 | 37.36±0.31 | 26.75±0.22 | 20.83±0.17 | 20.62±0.10 |
| DRGCN [48] | 63.04±0.99 | 49.86±1.68 | 56.40±3.91 | 43.92±2.58 | 74.82±9.35 | — | 26.79±0.35 | 19.98±0.45 | 22.10±0.21 |
| DPGNN [57] | 67.64±3.32 | 51.34±2.01 | **81.94**±2.85 | 76.17±9.32 | 82.20±9.18 | — | 30.72±1.49 | **26.52**±1.59 | 21.47±0.80 |
| ImGAGN [42] | 67.78±3.46 | 50.40±3.03 | 67.34±1.14 | 73.92±0.82 | 78.14±2.13 | — | 24.50±2.71 | 20.14±0.74 | 23.83±1.73 |
| GraphSMOTE [70] | 70.54±0.42 | 48.27±0.91 | 70.54±0.21 | 51.46±4.33 | 80.21±1.46 | — | 27.54±0.20 | 20.63±0.13 | 21.73±0.31 |
| GraphENS [37] | 78.54±0.55 | 58.76±0.95 | 79.47±0.27 | 86.03±0.25 | **90.26**±0.24 | 41.83±0.79 | 24.80±1.64 | 21.03±1.06 | 25.64±0.49 |
| GraphMixup [60] | 72.63±0.69 | 56.76±0.68 | 72.40±0.10 | 82.91±0.65 | 81.21±0.48 | 39.67±0.36 | 31.21±0.36 | 20.81±0.25 | 21.68±0.36 |
| LTE4G [67] | 75.42±1.26 | 58.52±1.35 | 81.68±0.22 | 72.29±3.90 | 87.99±1.34 | — | 32.00±0.34 | 22.37±0.34 | 23.11±0.33 |
| TAM [49] | **80.29**±0.37 | 60.88±0.26 | 81.20±0.18 | 86.19±0.24 | 90.19±0.21 | **41.94**±0.53 | 23.82±1.46 | 21.11±0.49 | **25.84**±0.30 |
| TOPOAUC [8] | 79.98±0.33 | **63.69**±0.93 | — | 77.02±2.60 | 85.79±1.62 | — | **33.87**±1.28 | 23.17±0.80 | 24.24±0.24 |
| GraphSHA [24] | 77.11±0.40 | 56.98±0.74 | 75.18±0.39 | 77.04±0.64 | 88.83±0.28 | 35.92±0.48 | 24.17±2.16 | 21.03±1.00 | 22.54±0.82 |
| $\rho = 100$ (High) | | | | | | | | | |
| GCN (bb.) [21] | 47.96±5.26 | 38.66±0.20 | 43.02±0.55 | 22.83±2.07 | 25.06±0.36 | 30.20±0.41 | 27.02±0.12 | 20.62±0.09 | 20.22±0.12 |
| DRGCN [48] | 49.11±3.52 | 41.40±2.34 | 44.87±3.23 | 35.80±1.98 | 54.85±6.26 | — | 27.11±0.44 | 20.57±0.38 | 21.03±0.36 |
| DPGNN [57] | 58.09±3.37 | 46.01±2.71 | 74.95±3.08 | 76.51±7.76 | 85.55±1.77 | — | 29.74±2.13 | **25.88**±1.23 | 21.18±0.92 |
| ImGAGN [42] | 57.01±4.30 | 48.70±3.16 | 55.56±0.44 | 60.84±6.59 | 69.88±3.24 | — | 24.76±2.93 | 19.81±0.63 | 22.21±1.01 |
| GraphSMOTE [70] | 58.91±0.55 | 41.60±0.74 | 64.02±0.20 | 21.05±0.88 | 23.91±0.12 | — | 26.91±0.13 | 20.67±0.14 | 20.27±0.12 |
| GraphENS [37] | 73.61±0.32 | 58.08±0.59 | 73.62±0.43 | 85.72±0.42 | 90.19±0.31 | 40.42±0.59 | 27.12±2.61 | 22.03±1.49 | 20.99±1.43 |
| GraphMixup [60] | 56.43±0.59 | 44.63±0.22 | 57.86±0.13 | 49.37±0.63 | 54.60±4.88 | 34.44±0.19 | 27.93±0.09 | 20.50±0.15 | 21.25±0.27 |
| LTE4G [67] | 62.22±1.22 | 51.16±1.21 | 67.89±10.50 | 72.49±2.92 | 83.28±1.91 | — | 29.69±0.51 | 21.08±0.29 | 22.79±0.51 |
| TAM [49] | **75.11**±0.39 | 59.10±0.43 | **78.98**±0.27 | **85.77**±0.41 | **90.20**±0.24 | **40.61**±0.55 | 23.96±1.94 | 22.74±0.84 | 21.97±0.15 |
| TOPOAUC [8] | 71.10±1.30 | **61.13**±2.02 | — | — | 85.13±2.23 | — | **32.10**±0.42 | 21.22±0.46 | **24.05**±0.61 |
| GraphSHA [24] | 73.05±0.35 | 53.92±0.69 | 65.28±0.56 | 72.59±1.36 | 87.06±0.58 | 28.12±0.32 | 24.80±1.36 | 22.05±1.49 | 20.84±1.34 |

Table D.3: **Macro-F1** score (% ± standard deviation) of **node** classification on manipulated **class-imbalanced** graph datasets with changing imbalance levels over 10 runs. "—" denotes out of memory or time limit. The best results are shown in **bold** and the runner-ups are underlined.

| Algorithm | Cora 0.81 | CiteSeer 0.74 | PubMed 0.80 | Computers 0.78 | Photo 0.82 | ogbn-arXiv 0.65 | Chameleon 0.23 | Squirrel 0.22 | Actor 0.22 |
|---|---|---|---|---|---|---|---|---|---|
| $\rho = 1$ (Balanced) | | | | | | | | | |
| GCN (bb.) [21] | 78.26±0.91 | 61.83±0.78 | 82.03±0.16 | 68.40±0.49 | 85.03±0.47 | 27.50±0.17 | 23.99±1.98 | 18.80±1.62 | 21.91±0.36 |
| DRGCN [48] | 75.46±0.97 | 62.65±1.09 | 80.46±3.08 | 55.55±22.56 | 83.23±0.69 | — | 32.85±2.01 | 23.41±0.95 | 22.30±0.63 |
| DPGNN [57] | 72.73±2.46 | 58.77±2.86 | 80.49±2.78 | 55.27±10.24 | 83.37±2.87 | — | 30.06±3.08 | 21.67±1.65 | 17.38±2.51 |
| ImGAGN [42] | 78.60±0.61 | 61.70±0.54 | 81.96±0.12 | 66.70±1.20 | 83.79±0.37 | — | 26.45±2.53 | 18.76±1.74 | 21.55±0.86 |
| GraphSMOTE [70] | 77.03±0.42 | 61.20±0.44 | 80.93±0.18 | 65.39±0.90 | 83.76±0.53 | — | 22.93±1.31 | 14.71±1.38 | 21.13±0.71 |
| GraphENS [37] | 78.26±0.91 | 61.86±0.77 | 82.03±0.16 | 68.39±0.89 | 85.64±1.22 | 27.92±0.16 | 23.99±1.98 | 18.80±1.62 | 21.91±0.35 |
| GraphMixup [60] | 77.54±0.73 | 66.34±0.98 | 82.92±0.10 | **70.81**±0.92 | **86.84**±0.40 | 26.14±0.38 | 33.76±0.38 | **23.78**±0.26 | 22.25±0.97 |
| LTE4G [67] | 78.45±1.10 | 64.35±2.04 | **83.19**±0.32 | 62.28±3.47 | 84.09±4.41 | — | 34.00±0.67 | 23.45±0.31 | 22.75±1.17 |
| TAM [49] | 79.34±0.61 | 61.95±0.33 | 74.23±0.77 | 68.53±0.72 | 85.48±1.39 | 27.93±0.22 | 25.40±1.86 | 18.43±2.12 | 22.10±0.54 |
| TOPOAUC [8] | **81.95**±0.36 | **69.28**±0.45 | — | 57.76±1.85 | 79.31±2.25 | — | **35.85**±0.79 | 23.70±0.44 | **24.43**±0.69 |
| GraphSHA [24] | 78.26±0.91 | 61.84±0.77 | 82.02±0.15 | 68.61±0.67 | 85.62±1.21 | **28.20**±0.16 | 23.99±1.98 | 18.80±1.62 | 21.91±0.36 |
| $\rho = 20$ (Low) | | | | | | | | | |
| GCN (bb.) [21] | 71.15±0.30 | 43.71±0.54 | 45.71±0.16 | 39.11±0.43 | 48.99±8.51 | **33.94**±0.26 | 16.67±0.52 | 9.83±0.49 | 12.18±0.67 |
| DRGCN [48] | 64.43±1.52 | 47.50±1.46 | 53.83±4.78 | 41.55±2.42 | 74.51±8.46 | — | 17.22±0.44 | 11.01±0.88 | 16.43±0.57 |
| DPGNN [57] | 68.70±3.61 | 50.06±2.10 | 81.54±2.42 | 66.97±8.96 | 79.53±8.70 | — | 26.03±1.83 | **21.53**±2.14 | 18.48±1.50 |
| ImGAGN [42] | 68.69±3.93 | 48.40±3.58 | 67.80±1.41 | 72.51±1.28 | 75.49±1.64 | — | 16.94±2.14 | 13.91±2.77 | 21.04±3.16 |
| GraphSMOTE [70] | 72.71±0.31 | 45.21±0.95 | 68.93±0.21 | 49.30±4.76 | 79.64±0.98 | — | 18.72±0.26 | 8.36±0.36 | 16.26±0.84 |
| GraphENS [37] | 77.16±0.50 | 57.80±0.96 | 79.71±0.24 | 77.89±0.69 | 88.20±0.28 | 30.16±0.37 | 19.58±0.99 | 16.73±1.73 | 23.30±0.38 |
| GraphMixup [60] | 74.03±0.62 | 55.31±0.69 | 73.63±0.11 | **81.54**±0.58 | 80.83±0.63 | 32.38±0.17 | 25.57±0.76 | 13.17±0.79 | 18.35±0.54 |
| LTE4G [67] | 76.46±1.17 | 57.35±1.49 | **82.12**±0.27 | 69.02±3.78 | 85.23±1.34 | — | 25.96±0.35 | 15.13±0.80 | 21.27±0.48 |
| TAM [49] | 78.83±0.32 | 60.12±0.31 | 80.75±0.15 | 78.10±0.65 | 88.16±0.30 | 30.30±0.29 | 19.99±1.21 | 17.29±0.94 | **24.01**±0.54 |
| TOPOAUC [8] | **80.61**±0.30 | **62.95**±1.21 | — | 67.15±5.38 | 81.69±2.38 | — | **29.06**±2.35 | 18.77±1.36 | 23.61±0.24 |
| GraphSHA [24] | 77.66±0.46 | 55.76±0.85 | 76.17±0.37 | 75.43±0.47 | **89.04**±0.27 | 32.09±0.23 | 19.64±1.26 | 16.73±1.73 | 20.36±0.97 |
| $\rho = 100$ (High) | | | | | | | | | |
| GCN (bb.) [21] | 43.97±7.75 | 30.77±0.21 | 34.08±0.77 | 20.44±1.45 | 16.99±0.64 | 31.11±0.50 | 14.79±0.11 | 8.27±0.17 | 9.01±0.66 |
| DRGCN [48] | 47.47±4.00 | 34.83±2.80 | 36.44±4.29 | 33.60±1.15 | 52.58±6.68 | — | 15.02±0.26 | 8.83±1.09 | 12.38±1.05 |
| DPGNN [57] | 58.66±3.44 | 41.53±3.59 | 75.47±3.04 | 68.30±8.53 | 84.82±1.94 | — | 23.96±2.16 | 18.85±2.36 | 19.62±1.15 |
| ImGAGN [42] | 55.03±5.32 | 43.92±4.61 | 50.74±1.93 | 57.30±5.51 | 67.27±4.58 | — | 14.12±2.75 | 10.62±4.19 | 18.84±2.87 |
| GraphSMOTE [70] | 58.93±0.54 | 35.40±0.93 | 63.27±0.19 | 17.90±1.20 | 16.34±0.04 | — | 15.13±0.16 | 9.31±0.29 | 9.05±0.76 |
| GraphENS [37] | 72.09±0.22 | 56.58±0.60 | 74.36±0.40 | 78.75±0.59 | 88.53±0.25 | 30.89±0.38 | 20.80±0.64 | 17.96±1.89 | 18.47±2.45 |
| GraphMixup [60] | 55.91±0.61 | 38.36±0.24 | 55.24±0.21 | 46.92±0.61 | 53.54±5.72 | **33.18**±0.16 | 20.02±0.11 | 10.07±0.17 | 14.46±0.78 |
| LTE4G [67] | 62.11±1.29 | 45.67±1.91 | 66.79±15.94 | 70.80±2.70 | 83.24±2.67 | — | 21.02±1.06 | 10.61±0.51 | 17.88±0.67 |
| TAM [49] | **75.07**±0.57 | 57.67±0.46 | **79.34**±0.21 | 78.97±0.29 | 88.44±0.23 | 30.91±0.45 | 20.37±1.49 | **20.28**±1.48 | 21.33±0.31 |
| TOPOAUC [8] | 70.01±0.95 | **58.92**±2.94 | — | — | 80.80±3.23 | — | **24.41**±1.51 | 13.62±2.00 | **22.40**±0.40 |
| GraphSHA [24] | 73.38±0.25 | 51.99±0.68 | 64.66±1.02 | 72.46±1.10 | 88.36±0.43 | 27.92±0.33 | 19.96±1.82 | 17.87±1.99 | 18.05±2.33 |

Table D.4: **AUC-ROC** score (% ± standard deviation) of **node** classification on manipulated **class-imbalanced** graph datasets with changing imbalance levels over 10 runs. "—" denotes out of memory or time limit. The best results are shown in **bold** and the runner-ups are underlined.

| Algorithm | Cora 0.81 | CiteSeer 0.74 | PubMed 0.80 | Computers 0.78 | Photo 0.82 | ogbn-arXiv 0.65 | Chameleon 0.23 | Squirrel 0.22 | Actor 0.22 |
|---|---|---|---|---|---|---|---|---|---|
| | | | | $\rho = 1$ **(Balanced)** | | | | | |
| GCN (bb.) [21] | 96.08±0.47 | 89.74±0.11 | 94.55±0.04 | 97.61±0.04 | 98.73±0.04 | 93.39±0.10 | 63.48±0.66 | 51.32±0.95 | 53.22±0.58 |
| DRGCN [48] | 95.01±0.30 | 89.32±0.22 | 94.73±0.41 | 93.43±11.13 | 98.49±0.12 | — | 64.93±0.49 | 55.40±0.32 | 54.28±0.45 |
| DPGNN [57] | 93.84±0.91 | 86.15±1.58 | 92.59±2.58 | 95.06±2.35 | 97.85±0.63 | — | 63.30±1.93 | 55.05±1.31 | 52.51±2.20 |
| ImGAGN [42] | 96.49±0.17 | 89.97±0.13 | 94.52±0.03 | 97.12±0.13 | 98.53±0.08 | — | 64.90±0.86 | 51.28±0.53 | 53.66±1.05 |
| GraphSMOTE [70] | 95.88±0.35 | 89.48±0.13 | 93.94±0.06 | 97.18±0.09 | 98.64±0.04 | — | 60.33±3.34 | 51.21±0.33 | 53.70±0.36 |
| GraphENS [37] | 96.08±0.47 | 89.74±0.11 | 94.55±0.04 | 97.67±0.07 | 98.76±0.11 | 93.33±0.07 | 63.48±0.66 | 51.32±0.95 | 53.22±0.58 |
| GraphMixup [60] | 96.42±0.25 | 89.83±0.94 | 95.16±0.04 | **97.96±0.13** | **98.79±0.03** | 92.64±0.09 | 63.16±0.65 | 54.01±0.31 | 53.82±0.76 |
| LTE4G [67] | 96.54±0.39 | 89.00±0.67 | **95.61±0.05** | 95.88±3.08 | 98.17±0.56 | — | 65.42±0.49 | 55.22±0.40 | **57.75±0.64** |
| TAM [49] | 96.85±0.04 | 89.77±0.08 | 92.62±0.12 | 97.69±0.06 | 98.71±0.16 | 93.34±0.06 | 60.11±1.56 | 51.67±0.78 | 51.84±0.46 |
| TOPOAUC [8] | **97.56±0.04** | **91.89±0.23** | — | 91.26±2.37 | 94.11±1.46 | — | **67.70±0.88** | **55.43±0.09** | 57.17±0.53 |
| GraphSHA [24] | 96.08±0.47 | 89.74±0.11 | 94.54±0.04 | 97.69±0.06 | 98.76±0.11 | 93.50±0.08 | 63.48±0.66 | 51.32±0.95 | 53.22±0.58 |
| | | | | $\rho = 20$ **(Low)** | | | | | |
| GCN (bb.) [21] | 95.04±0.08 | 86.57±0.19 | 92.23±0.09 | 94.41±0.53 | 94.35±0.96 | **93.54±0.09** | 57.70±0.42 | 51.70±0.08 | 52.05±0.36 |
| DRGCN [48] | 92.68±0.30 | 84.92±0.63 | 92.04±0.35 | 93.84±0.50 | 98.22±0.49 | — | 56.19±0.66 | 49.42±0.68 | 55.20±0.28 |
| DPGNN [57] | 92.41±1.08 | 81.04±1.44 | 93.24±1.77 | 95.03±1.93 | 96.30±2.20 | — | 60.45±2.11 | **55.66±1.56** | 51.20±0.71 |
| ImGAGN [42] | 92.89±0.45 | 83.52±0.51 | 90.53±1.69 | 94.55±1.38 | 95.25±1.19 | — | 53.11±2.76 | 49.98±0.74 | 54.92±1.99 |
| GraphSMOTE [70] | 95.14±0.16 | 86.59±0.27 | 91.77±0.09 | 96.51±0.41 | 98.28±0.31 | — | 56.92±0.23 | 51.71±0.06 | 53.79±0.27 |
| GraphENS [37] | 96.32±0.13 | 87.46±0.43 | 93.26±0.11 | 97.70±0.09 | 98.71±0.05 | 93.04±0.08 | 58.54±2.61 | 52.33±0.77 | **56.10±0.13** |
| GraphMixup [60] | 96.23±0.13 | 85.86±0.30 | 93.81±0.05 | **98.24±0.05** | 98.27±0.09 | 92.75±0.09 | 61.05±0.37 | 53.78±0.39 | 52.70±0.12 |
| LTE4G [67] | 95.67±0.33 | 86.14±0.99 | **94.90±0.29** | 96.85±0.25 | 98.69±0.25 | — | 63.31±0.60 | 53.83±0.29 | 54.36±0.44 |
| TAM [49] | 96.74±0.07 | 88.41±0.19 | 93.71±0.07 | 97.69±0.10 | 98.71±0.04 | 93.07±0.07 | 58.71±1.96 | 51.17±1.18 | 55.72±0.17 |
| TOPOAUC [8] | **97.09±0.16** | **88.62±0.59** | — | 91.04±1.72 | 93.96±1.62 | — | **65.86±0.82** | 53.47±0.49 | 54.78±0.25 |
| GraphSHA [24] | 96.27±0.05 | 87.36±0.22 | 93.14±0.10 | 97.78±0.06 | **98.74±0.06** | 93.39±0.11 | 58.19±2.52 | 52.33±0.77 | 52.48±1.05 |
| | | | | $\rho = 100$ **(High)** | | | | | |
| GCN (bb.) [21] | 91.55±0.80 | 80.26±0.32 | 79.13±1.41 | 87.24±1.56 | 76.98±1.42 | **92.87±0.12** | 57.86±0.65 | 51.16±0.06 | 50.79±0.79 |
| DRGCN [48] | 89.80±0.54 | 79.58±1.24 | 84.19±2.66 | 92.09±0.75 | 96.50±0.63 | — | 55.79±1.06 | 49.24±0.39 | **54.87±0.25** |
| DPGNN [57] | 87.41±2.09 | 78.59±1.51 | 90.02±2.99 | 95.34±1.58 | 97.09±0.57 | — | 59.14±2.36 | **54.25±1.11** | 50.35±0.72 |
| ImGAGN [42] | 88.38±1.39 | 81.37±0.69 | 87.52±1.15 | 92.75±1.38 | 94.82±0.72 | — | 52.73±2.84 | 49.44±0.71 | 53.60±1.51 |
| GraphSMOTE [70] | 93.29±0.14 | 82.56±0.37 | 78.88±1.27 | 89.52±1.39 | 84.03±1.44 | — | 56.82±0.53 | 51.49±0.11 | 52.51±2.31 |
| GraphENS [37] | 95.55±0.13 | 85.91±0.32 | 91.38±0.14 | 97.74±0.06 | **98.65±0.03** | 92.83±0.09 | 58.15±1.92 | 52.87±0.55 | 50.91±1.27 |
| GraphMixup [60] | 92.31±0.22 | 82.50±0.12 | 90.58±0.09 | 97.06±0.17 | 93.85±0.69 | 92.03±0.09 | 58.73±0.27 | 53.24±0.08 | 52.46±0.14 |
| LTE4G [67] | 92.82±0.56 | 83.87±1.13 | 90.77±2.91 | 97.19±1.93 | 97.26±0.46 | — | 59.10±0.89 | 52.60±0.18 | 53.96±0.67 |
| TAM [49] | **96.24±0.04** | **87.22±0.25** | **93.14±0.05** | **97.75±0.05** | **98.65±0.03** | 92.84±0.10 | 56.81±2.37 | 52.99±0.46 | 52.42±0.14 |
| TOPOAUC [8] | 90.42±1.64 | 86.50±0.50 | — | — | 93.14±0.96 | — | **61.71±0.80** | 52.17±0.28 | 53.31±0.34 |
| GraphSHA [24] | 95.73±0.11 | 85.10±0.16 | 90.09±3.59 | 97.51±0.14 | 98.62±0.06 | 92.18±0.13 | 56.00±2.91 | 52.87±0.55 | 50.72±0.98 |

## D.1.2 EFFECTIVENESS OF NODE-LEVEL LOCAL TOPOLOGY-IMBALANCED ALGORITHMS

Table D.5: **Accuracy** score (% ± standard deviation) of **node** classification on manipulated **local topology-imbalanced** graph datasets with changing imbalance levels over 10 runs. "—" denotes out of memory or time limit. The best results are shown in **bold** and the runner-ups are underlined.

| Algorithm | Cora 0.81 | CiteSeer 0.74 | PubMed 0.80 | Computers 0.78 | Photo 0.82 | ogbn-arXiv 0.65 | Chameleon 0.23 | Squirrel 0.22 | Actor 0.22 |
|---|---|---|---|---|---|---|---|---|---|
| **Imbalance Ratio: Low** | | | | | | | | | |
| GCN (bb.) [21] | 80.16±0.97 | 66.99±1.78 | 83.97±0.14 | 70.39±1.68 | 87.28±2.77 | 55.45±0.06 | 50.42±1.22 | 30.87±0.60 | 24.23±1.74 |
| DEMO-Net [59] | 81.40±0.43 | 68.42±0.72 | 82.70±0.51 | 79.07±0.53 | 88.06±1.81 | 65.31±0.15 | 56.18±0.62 | 38.86±0.78 | 29.34±0.59 |
| meta-tail2vec [30] | 38.11±1.36 | 24.77±2.39 | 59.93±1.95 | 71.13±0.41 | 77.16±0.33 | 33.61±0.25 | 41.45±0.46 | 24.47±0.18 | 25.98±0.00 |
| Tail-GNN [31] | 80.16±0.64 | 70.37±0.57 | 84.56±0.30 | 86.40±0.92 | 92.79±0.19 | — | 51.87±0.79 | 31.89±0.93 | 28.67±0.47 |
| Cold Brew [72] | 75.37±1.07 | 65.16±0.59 | **86.11±0.05** | 79.61±0.40 | 85.94±0.23 | 68.50±0.07 | **58.78±0.19** | **39.57±0.19** | **32.93±0.45** |
| LTE4G [67] | 82.10±0.56 | 69.17±0.96 | 84.64±0.30 | 81.24±2.79 | 92.47±0.21 | — | 58.61±0.98 | 25.74±2.58 | 24.53±1.12 |
| RawlsGCN [19] | 79.95±0.29 | 72.20±0.39 | 85.97±0.12 | 78.74±2.01 | 87.89±0.10 | 41.70±0.23 | 44.91±1.15 | 29.68±0.83 | 28.54±0.12 |
| GraphPatcher [17] | **84.00±0.62** | **72.34±0.32** | 85.58±0.13 | **87.60±0.23** | **93.20±0.32** | 66.35±0.09 | 55.77±1.04 | 35.16±0.22 | 27.15±0.80 |
| **Imbalance Ratio: Mid** | | | | | | | | | |
| GCN (bb.) [21] | 80.16±1.09 | 66.87±0.85 | 83.97±0.13 | 71.65±2.10 | 89.43±0.58 | 52.93±0.33 | 52.74±0.60 | 28.70±0.68 | 21.55±1.74 |
| DEMO-Net [59] | 80.37±0.52 | 69.73±1.31 | 84.11±0.20 | 79.38±0.98 | 88.09±1.30 | 65.81±0.11 | 55.51±0.87 | 39.45±0.62 | 29.12±0.30 |
| meta-tail2vec [30] | 32.17±0.68 | 29.97±3.61 | 59.82±2.86 | 68.17±1.07 | 79.82±1.02 | 33.71±1.16 | 38.78±0.44 | 24.90±0.25 | 26.09±0.07 |
| Tail-GNN [31] | 79.05±1.15 | 69.97±1.03 | 85.78±0.41 | **84.09±1.01** | 92.21±0.09 | — | 53.20±0.80 | 30.43±1.06 | 28.02±0.71 |
| Cold Brew [72] | 73.84±2.10 | 67.42±0.97 | **86.51±0.04** | 80.19±0.24 | 88.13±0.24 | 69.97±0.07 | **59.16±0.40** | **43.04±0.24** | **33.01±0.19** |
| LTE4G [67] | 82.54±0.46 | 70.55±0.54 | 84.77±0.78 | 81.32±2.21 | 91.09±0.19 | — | 55.84±2.86 | 32.43±3.31 | 24.00±0.49 |
| RawlsGCN [19] | 80.52±0.14 | 72.38±0.43 | 86.05±0.12 | 78.78±1.40 | 90.53±1.32 | 40.00±0.05 | 44.96±0.79 | 29.93±0.65 | 28.29±0.24 |
| GraphPatcher [17] | **83.25±0.42** | **73.38±0.42** | 85.60±0.16 | 83.68±0.69 | 92.28±0.06 | 66.74±0.04 | 55.19±0.41 | 36.94±0.11 | 23.85±0.92 |
| **Imbalance Ratio: High** | | | | | | | | | |
| GCN (bb.) [21] | 78.70±1.05 | 65.07±0.81 | 83.87±0.32 | 68.15±4.13 | 89.42±1.24 | 50.72±0.30 | 53.33±1.09 | 29.56±2.72 | 23.86±0.90 |
| DEMO-Net [59] | 78.23±1.32 | 67.11±0.44 | 83.51±0.29 | 78.34±0.88 | 88.08±0.30 | 65.76±0.18 | 54.08±1.41 | 36.98±1.27 | 28.96±0.30 |
| meta-tail2vec [30] | 38.16±1.42 | 21.62±1.70 | 58.39±2.25 | 71.03±2.20 | 66.37±2.96 | 35.31±0.21 | 37.94±0.52 | 25.18±0.30 | 25.98±0.03 |
| Tail-GNN [31] | 81.20±0.55 | 69.69±0.55 | 84.95±0.37 | **86.39±0.82** | 92.55±0.40 | — | 53.00±0.89 | 31.08±0.91 | 28.36±1.16 |
| Cold Brew [72] | 75.44±2.31 | 66.12±0.71 | **86.44±0.02** | 78.59±0.10 | 86.83±0.27 | 70.32±0.08 | 59.47±0.14 | 40.16±0.16 | 33.44±0.22 |
| LTE4G [67] | 81.93±1.43 | 67.09±0.73 | 84.30±0.49 | 83.33±1.59 | 92.12±0.32 | — | 56.39±2.69 | 30.16±4.09 | 23.83±0.85 |
| RawlsGCN [19] | 81.66±0.17 | 69.88±0.74 | 85.72±0.07 | 79.27±0.41 | 87.99±1.16 | 39.14±0.16 | 42.22±0.37 | 28.54±0.79 | 29.30±0.17 |
| GraphPatcher [17] | 80.77±0.23 | **73.13±0.48** | 85.74±0.14 | 85.47±0.16 | **93.57±0.13** | 67.38±0.06 | 56.74±0.25 | 37.12±0.18 | 25.48±0.49 |

Table D.6: **Balanced Accuracy** score (% ± standard deviation) of **node** classification on manipulated **local topology-imbalanced** graph datasets with changing imbalance levels over 10 runs. "—" denotes out of memory or time limit. Best results are shown in **bold** and the runner-ups are underlined.

| Algorithm | Cora 0.81 | CiteSeer 0.74 | PubMed 0.80 | Computers 0.78 | Photo 0.82 | ogbn-arXiv 0.65 | Chameleon 0.23 | Squirrel 0.22 | Actor 0.22 |
|---|---|---|---|---|---|---|---|---|---|
| **Imbalance Ratio: Low** | | | | | | | | | |
| GCN (bb.) [21] | 81.19±0.89 | 63.72±2.11 | 84.31±0.14 | 77.82±1.32 | 86.91±2.30 | 22.36±0.15 | 50.82±0.89 | 30.86±0.60 | 23.02±1.14 |
| DEMO-Net [59] | 82.53±0.35 | 65.58±0.71 | 83.84±0.08 | 84.87±0.25 | 90.06±0.82 | 41.57±1.20 | 56.18±0.45 | 38.85±0.78 | 26.59±0.61 |
| meta-tail2vec [30] | 27.64±1.24 | 19.99±1.80 | 53.78±1.75 | 55.68±4.36 | 80.39±0.35 | 7.46±0.43 | 40.58±0.42 | 24.46±0.16 | 20.12±0.15 |
| Tail-GNN [31] | 82.10±0.28 | 66.78±0.30 | 85.42±0.41 | **90.80±0.53** | 93.57±0.77 | — | 51.96±0.99 | 31.89±0.93 | 28.28±0.30 |
| Cold Brew [72] | 77.41±0.62 | 62.25±1.02 | 86.11±0.05 | 85.58±0.35 | 88.67±0.18 | 47.29±0.12 | 58.05±0.20 | 39.56±0.19 | **29.01±1.40** |
| LTE4G [67] | 82.68±0.43 | 67.45±0.40 | 85.15±0.25 | 87.35±2.31 | 92.86±0.44 | — | 58.15±0.91 | 25.73±2.58 | 24.31±0.62 |
| RawlsGCN [19] | 81.80±0.21 | **68.80±0.27** | **86.76±0.13** | 84.67±0.64 | 90.59±0.31 | 13.32±0.08 | 45.15±0.83 | 29.69±0.83 | 28.13±0.12 |
| GraphPatcher [17] | **84.66±0.47** | 68.56±0.16 | 85.68±0.19 | 90.73±0.18 | **93.61±0.26** | 43.97±0.09 | 55.28±1.02 | 35.15±0.22 | 22.46±0.46 |
| **Imbalance Ratio: Mid** | | | | | | | | | |
| GCN (bb.) [21] | 81.54±0.39 | 63.00±0.81 | 84.46±0.13 | 78.91±2.60 | 89.02±0.58 | 18.12±0.38 | 51.74±0.65 | 28.69±0.68 | 21.25±0.33 |
| DEMO-Net [59] | 81.75±0.26 | 65.52±1.00 | 84.99±0.08 | 86.24±0.63 | 89.43±0.90 | 42.88±0.66 | 55.37±1.14 | 39.45±0.62 | 26.70±0.17 |
| meta-tail2vec [30] | 32.56±0.98 | 25.17±1.60 | 56.16±1.92 | 55.60±4.13 | 81.26±1.78 | 7.46±0.68 | 39.92±0.52 | 24.91±0.25 | 20.08±0.07 |
| Tail-GNN [31] | 81.47±0.27 | 66.05±0.65 | 86.64±0.30 | 90.06±0.28 | 93.08±0.19 | — | 53.40±0.71 | 30.44±1.05 | 26.26±0.73 |
| Cold Brew [72] | 76.00±1.85 | 62.88±1.10 | 86.41±0.26 | 85.62±0.07 | 89.20±1.85 | 47.73±0.13 | 58.71±0.46 | 43.04±0.25 | **30.65±0.19** |
| LTE4G [67] | 82.79±0.32 | 65.79±0.52 | 85.07±0.67 | 87.16±1.99 | 92.38±0.17 | — | 55.04±2.99 | 32.43±3.32 | 24.14±0.56 |
| RawlsGCN [19] | 82.11±0.18 | 67.74±0.26 | **86.89±0.16** | 84.76±1.49 | 91.50±0.20 | 11.62±0.13 | 44.42±1.19 | 29.93±0.65 | 27.29±0.20 |
| GraphPatcher [17] | **83.94±0.20** | **69.17±0.22** | 85.72±0.08 | **90.95±0.21** | 93.21±0.06 | 38.49±0.41 | 54.39±0.41 | 36.93±0.11 | 23.10±0.54 |
| **Imbalance Ratio: High** | | | | | | | | | |
| GCN (bb.) [21] | 81.68±0.84 | 62.76±0.69 | 84.70±0.07 | 78.15±3.40 | 90.19±0.35 | 15.93±0.27 | 52.93±1.49 | 29.54±2.72 | 22.83±0.57 |
| DEMO-Net [59] | 82.14±0.32 | 64.16±0.76 | 84.70±0.09 | 87.19±0.50 | 90.51±0.10 | 42.00±1.23 | 54.11±1.30 | 36.37±1.66 | 27.16±0.28 |
| meta-tail2vec [30] | 25.23±4.75 | 17.49±1.12 | 52.61±2.40 | 67.44±2.57 | 72.79±8.67 | 7.82±0.12 | 36.66±0.63 | 25.20±0.30 | 20.03±0.03 |
| Tail-GNN [31] | 83.79±0.49 | 67.15±0.12 | 86.07±0.12 | 90.72±0.50 | **94.62±0.22** | — | 53.12±0.69 | 31.09±0.91 | 27.36±0.70 |
| Cold Brew [72] | 78.25±1.72 | 63.45±1.10 | 86.43±0.09 | 85.73±0.10 | 90.14±0.65 | 48.43±0.16 | 59.66±0.17 | 40.16±0.16 | **30.68±0.18** |
| LTE4G [67] | **84.15±1.36** | 64.63±0.65 | 85.12±0.38 | 89.78±0.59 | 94.24±0.28 | — | 56.25±3.03 | 30.15±4.10 | 24.03±0.53 |
| RawlsGCN [19] | 83.19±0.24 | 67.51±0.41 | **86.60±0.10** | 86.25±0.13 | 91.82±0.24 | 10.65±0.07 | 42.87±0.81 | 28.54±0.79 | 28.14±0.20 |
| GraphPatcher [17] | 83.91±0.11 | **68.89±0.09** | 85.78±0.18 | **91.23±0.05** | 93.57±0.13 | 40.59±0.39 | 56.78±0.25 | 37.11±0.18 | 23.98±0.22 |

Table D.7: **Macro-F1** score (% ± standard deviation) of **node** classification on manipulated **local topology-imbalanced** graph datasets with changing imbalance levels over 10 runs. "—" denotes out of memory or time limit. The best results are shown in **bold** and the runner-ups are underlined.

| Algorithm | Cora 0.81 | CiteSeer 0.74 | PubMed 0.80 | Computers 0.78 | Photo 0.82 | ogbn-arXiv 0.65 | Chameleon 0.23 | Squirrel 0.22 | Actor 0.22 |
|---|---|---|---|---|---|---|---|---|---|
| **Imbalance Ratio: Low** | | | | | | | | | |
| GCN (bb.) [21] | 77.96±1.22 | 63.07±1.72 | 82.88±0.12 | 58.70±2.09 | 82.92±2.33 | 21.64±0.37 | 50.63±1.22 | 29.19±1.40 | 19.05±2.75 |
| DEMO-Net [59] | 79.01±0.52 | 64.74±0.77 | 81.93±0.52 | 65.66±1.07 | 84.73±1.70 | 43.08±1.01 | 55.67±0.53 | 38.58±0.69 | 26.45±0.72 |
| meta-tail2vec [30] | 23.87±1.96 | 11.60±2.38 | 53.82±1.49 | 50.14±3.23 | 71.66±0.31 | 6.68±0.82 | 40.11±0.47 | 22.06±1.32 | 9.48±1.49 |
| Tail-GNN [31] | 77.91±0.37 | 66.32±0.45 | 84.05±0.30 | 75.59±0.45 | 89.56±0.35 | — | 51.09±0.80 | 29.90±0.46 | **26.95±0.65** |
| Cold Brew [72] | 73.42±0.83 | 61.70±0.77 | 85.41±0.04 | 65.71±0.27 | 81.89±0.28 | **48.39±0.13** | **58.50±0.23** | **38.90±0.47** | 26.89±1.99 |
| LTE4G [67] | 79.49±0.55 | 66.21±0.66 | 83.64±0.33 | 69.51±3.48 | 89.78±0.28 | — | 58.48±0.89 | 24.48±3.27 | 23.15±0.67 |
| RawlsGCN [19] | 77.88±0.28 | 68.20±0.29 | 85.43±0.12 | 67.01±2.37 | 85.49±0.34 | 14.56±0.05 | 43.80±1.19 | 29.24±0.50 | 26.80±0.10 |
| GraphPatcher [17] | **81.64±0.78** | **68.28±0.22** | 84.65±0.18 | **78.27±0.69** | **90.48±0.46** | 44.74±0.04 | 55.67±0.98 | 35.11±0.30 | 19.03±0.95 |
| **Imbalance Ratio: Mid** | | | | | | | | | |
| GCN (bb.) [21] | 78.43±0.94 | 62.70±0.77 | 82.81±0.10 | 60.43±2.57 | 84.99±0.76 | 17.40±0.42 | 52.11±0.66 | 27.48±1.74 | 19.37±1.28 |
| DEMO-Net [59] | 78.11±0.54 | 65.23±1.02 | 83.44±0.20 | 67.75±0.73 | 84.14±1.52 | 44.63±0.59 | 55.17±0.98 | 38.93±0.60 | 26.48±0.39 |
| meta-tail2vec [30] | 30.58±1.61 | 19.16±3.28 | 54.86±0.66 | 49.59±1.24 | 76.33±0.82 | 6.62±0.98 | 38.29±0.62 | 24.30±0.24 | 8.70±0.42 |
| Tail-GNN [31] | 77.40±1.05 | 65.77±0.81 | 85.23±0.41 | 73.31±1.40 | 88.73±0.13 | — | 52.53±0.74 | 27.98±0.79 | 25.68±0.61 |
| Cold Brew [72] | 71.85±2.17 | 62.70±1.00 | 85.69±0.07 | 66.03±0.20 | 84.26±0.16 | 49.82±0.18 | **58.91±0.46** | **42.07±0.40** | **30.71±0.20** |
| LTE4G [67] | 80.29±0.45 | 65.76±0.47 | 83.66±0.88 | 68.65±3.35 | 87.57±0.13 | — | 55.32±2.93 | 32.20±3.71 | 22.78±0.50 |
| RawlsGCN [19] | 78.30±0.26 | 67.50±0.18 | 85.49±0.14 | 66.62±2.07 | 87.77±1.45 | 12.56±0.20 | 44.44±1.11 | 29.48±0.49 | 26.32±0.21 |
| GraphPatcher [17] | **80.92±0.42** | **68.95±0.25** | 84.65±0.18 | **74.74±0.73** | **88.89±0.11** | 40.76±0.32 | 54.72±0.35 | 37.06±0.11 | 22.28±0.58 |
| **Imbalance Ratio: High** | | | | | | | | | |
| GCN (bb.) [21] | 77.03±1.19 | 61.89±0.66 | 82.72±0.32 | 58.78±4.61 | 85.73±0.84 | 14.52±0.27 | 52.56±1.25 | 27.43±2.85 | 20.48±2.59 |
| DEMO-Net [59] | 76.85±1.11 | 63.38±0.68 | 82.87±0.32 | 65.92±0.62 | 84.70±0.23 | 43.74±1.02 | 53.73±1.15 | 35.94±1.44 | 27.24±0.28 |
| meta-tail2vec [30] | 21.82±5.45 | 7.56±2.36 | 52.03±3.18 | 54.07±2.19 | 65.22±7.60 | 6.71±0.18 | 34.06±1.22 | 23.00±0.60 | 8.30±0.06 |
| Tail-GNN [31] | 79.66±0.57 | 66.29±0.33 | 84.31±0.35 | 75.04±1.16 | 89.67±0.39 | — | 52.67±0.68 | 28.08±1.18 | 25.85±0.75 |
| Cold Brew [72] | 73.71±2.07 | 62.71±0.76 | 85.72±0.02 | 65.83±0.13 | 83.37±0.18 | 50.43±0.24 | 59.18±0.23 | 39.43±0.41 | 30.32±0.28 |
| LTE4G [67] | 80.57±1.66 | 63.70±0.66 | 83.11±0.54 | 73.38±1.69 | 89.18±0.52 | — | 55.92±2.90 | 28.30±5.09 | 22.95±0.61 |
| RawlsGCN [19] | 80.19±0.20 | 66.41±0.56 | 85.11±0.08 | 69.00±0.95 | 85.79±1.10 | 11.24±0.09 | 41.86±0.35 | 27.86±0.60 | 27.41±0.16 |
| GraphPatcher [17] | 79.35±0.18 | **68.69±0.20** | 84.74±0.16 | **77.10±0.43** | **90.94±0.18** | 42.93±0.29 | 56.05±0.29 | 37.34±0.19 | 23.10±0.41 |

Table D.8: **AUC-ROC** score (% ± standard deviation) of **node** classification on manipulated **local topology-imbalanced** graph datasets with changing imbalance levels over 10 runs. "—" denotes out of memory or time limit. The best results are shown in **bold** and the runner-ups are underlined.

| Algorithm | Cora 0.81 | CiteSeer 0.74 | PubMed 0.80 | Computers 0.78 | Photo 0.82 | ogbn-arXiv 0.65 | Chameleon 0.23 | Squirrel 0.22 | Actor 0.22 |
|---|---|---|---|---|---|---|---|---|---|
| **Imbalance Ratio: Low** | | | | | | | | | |
| GCN (bb.) [21] | 96.34±0.51 | 87.80±1.39 | 94.89±0.06 | 95.61±0.21 | 97.82±0.80 | 87.84±0.06 | 78.76±0.27 | 62.68±0.27 | 53.43±1.42 |
| DEMO-Net [59] | 96.27±0.13 | 88.66±0.30 | 94.22±0.07 | 96.52±0.10 | 98.04±0.29 | 92.89±0.32 | **80.31±0.74** | **68.00±0.69** | 58.96±0.35 |
| meta-tail2vec [30] | 62.83±1.22 | 53.44±2.06 | 71.50±3.86 | 89.26±0.64 | 94.95±0.16 | 62.42±2.26 | 69.63±0.28 | 53.44±0.62 | 50.50±0.35 |
| Tail-GNN [31] | 96.55±0.06 | 89.11±0.56 | 94.28±0.69 | **97.92±0.11** | 98.70±0.13 | — | 76.76±0.44 | 63.91±1.16 | 59.72±0.56 |
| Cold Brew [72] | 93.87±0.50 | 87.38±0.98 | 95.67±0.03 | 97.88±0.05 | 98.71±0.02 | **95.13±0.10** | 78.37±0.48 | 66.72±0.10 | 65.12±1.03 |
| LTE4G [67] | 96.14±0.37 | 88.78±1.05 | 94.52±0.45 | 92.94±2.66 | 98.83±0.05 | — | 78.00±1.53 | 58.28±2.12 | 56.17±0.86 |
| RawlsGCN [19] | 96.84±0.07 | **91.06±0.45** | **96.34±0.03** | 97.47±0.09 | **98.99±0.02** | 79.78±0.12 | 71.94±0.09 | 58.89±0.13 | 59.34±0.16 |
| GraphPatcher [17] | **97.10±0.04** | 90.75±0.15 | 94.98±0.03 | 97.88±0.05 | 98.54±0.05 | 90.90±0.03 | 79.09±0.10 | 66.65±0.30 | 53.80±0.56 |
| **Imbalance Ratio: Mid** | | | | | | | | | |
| GCN (bb.) [21] | 95.88±0.29 | 87.55±0.33 | 95.07±0.06 | 95.70±0.77 | 98.05±0.11 | 86.55±0.29 | 78.60±0.38 | 62.45±0.74 | 51.23±0.46 |
| DEMO-Net [59] | 95.59±0.08 | 88.31±0.26 | 94.78±0.05 | 97.14±0.08 | 97.75±0.13 | 93.29±0.35 | **80.31±0.91** | 67.96±0.32 | 59.54±0.45 |
| meta-tail2vec [30] | 65.88±0.99 | 59.39±0.47 | 73.30±1.00 | 90.45±0.24 | 94.66±0.64 | 64.91±1.30 | 68.34±0.42 | 53.85±0.04 | 49.40±0.20 |
| Tail-GNN [31] | 96.19±0.22 | 89.73±0.39 | 95.17±0.55 | 97.98±0.19 | 98.39±0.27 | — | 76.40±0.64 | 62.60±1.37 | 58.11±0.61 |
| Cold Brew [72] | 93.14±0.87 | 88.06±1.14 | 96.06±0.05 | 98.03±0.02 | 98.40±0.33 | 95.66±0.06 | 77.61±0.13 | **68.96±0.16** | **65.47±0.17** |
| LTE4G [67] | 95.16±0.59 | 89.69±0.44 | 94.41±0.96 | 93.05±2.75 | 97.74±0.33 | — | 77.80±1.83 | 61.90±1.44 | 55.08±0.68 |
| RawlsGCN [19] | 96.65±0.07 | 91.53±0.43 | **96.29±0.04** | 97.53±0.07 | **98.97±0.02** | 79.84±0.22 | 71.42±0.15 | 59.10±0.08 | 59.19±0.14 |
| GraphPatcher [17] | **96.66±0.08** | **91.74±0.09** | 94.63±0.02 | **98.05±0.07** | 98.14±0.02 | 92.04±0.11 | 78.36±0.26 | 66.40±0.12 | 54.64±0.62 |
| **Imbalance Ratio: High** | | | | | | | | | |
| GCN (bb.) [21] | 96.12±0.31 | 87.93±0.88 | 94.96±0.08 | 95.42±0.99 | 87.28±2.77 | 85.44±0.40 | 78.63±0.38 | 63.00±1.36 | 53.72±0.70 |
| DEMO-Net [59] | 95.40±0.19 | 87.64±0.09 | 94.31±0.08 | 97.10±0.25 | 98.34±0.15 | 93.45±0.78 | **79.19±0.89** | 66.28±1.10 | 58.89±0.28 |
| meta-tail2vec [30] | 61.19±1.02 | 49.89±1.49 | 69.63±3.44 | 92.90±0.31 | 91.77±2.16 | 62.67±1.41 | 68.62±0.77 | 55.24±0.23 | 50.04±0.34 |
| Tail-GNN [31] | 96.80±0.05 | 89.96±0.42 | 95.10±0.15 | **97.93±0.10** | 98.60±0.13 | — | 76.81±0.31 | 64.39±0.42 | 59.60±1.06 |
| Cold Brew [72] | 94.04±0.64 | 88.26±0.37 | 95.90±0.03 | 97.82±0.05 | 98.87±0.10 | **95.86±0.06** | 78.97±0.34 | 67.72±0.11 | **65.52±0.16** |
| LTE4G [67] | 96.17±0.35 | 86.40±2.05 | 94.46±0.31 | 95.97±1.48 | 98.72±0.06 | — | 77.34±1.75 | 60.35±2.63 | 54.98±0.84 |
| RawlsGCN [19] | **96.87±0.03** | 90.42±0.13 | **96.16±0.03** | 97.51±0.04 | **99.21±0.02** | 79.47±0.15 | 71.04±0.11 | 58.32±0.12 | 59.38±0.13 |
| GraphPatcher [17] | 96.15±0.05 | **91.55±0.48** | 94.90±0.02 | 97.59±0.05 | 98.75±0.02 | 92.53±0.05 | 78.67±0.15 | 65.82±0.40 | 56.03±0.19 |

### D.1.3 EFFECTIVENESS OF NODE-LEVEL GLOBAL TOPOLOGY-IMBALANCED ALGORITHMS

Table D.9: **Accuracy** score (% ± standard deviation) of **node** classification on manipulated **global topology-imbalanced** graph datasets with changing imbalance levels over 10 runs. "—" denotes out of memory or time limit. The best results are shown in **bold** and the runner-ups are underlined.

| Algorithm | Cora 0.81 | CiteSeer 0.74 | PubMed 0.80 | Computers 0.78 | Photo 0.82 | ogbn-arXiv 0.65 | Chameleon 0.23 | Squirrel 0.22 | Actor 0.22 |
|---|---|---|---|---|---|---|---|---|---|
| **Imbalance Ratio: Low** | | | | | | | | | |
| GCN (bb.) [21] | 81.09±0.76 | 70.39±1.11 | 84.44±0.20 | 76.23±2.16 | 88.16±1.86 | 53.09±0.17 | 37.80±0.59 | 25.15±0.57 | 24.34±0.78 |
| ReNode [7] | 81.94±0.48 | 71.93±0.88 | 83.86±0.13 | 79.42±1.70 | 89.91±0.51 | 52.75±0.14 | 37.67±0.49 | 25.36±0.49 | 24.65±0.28 |
| TAM [49] | 81.48±0.30 | 74.06±0.12 | 84.26±0.08 | 84.17±0.18 | 91.93±0.22 | 54.16±0.05 | 38.78±0.17 | 25.64±0.12 | 24.77±0.33 |
| PASTEL [52] | 82.49±0.34 | 74.38±0.31 | — | 85.08±0.84 | 91.22±0.40 | — | 54.05±1.13 | 34.01±0.57 | 29.50±0.39 |
| TOPOAUC [8] | 81.38±0.80 | 72.31±0.75 | — | 77.23±1.31 | 89.04±0.96 | — | 37.68±0.81 | 23.52±0.78 | 25.88±0.97 |
| HyperIMBA [12] | 80.67±0.64 | 73.46±0.70 | 85.19±0.26 | 85.22±0.64 | 92.75±0.13 | — | 43.48±1.30 | 32.69±0.66 | 27.09±3.15 |
| **Imbalance Ratio: High** | | | | | | | | | |
| GCN (bb.) [21] | 79.10±1.28 | 68.37±1.73 | 83.44±0.16 | 75.02±2.20 | 86.32±1.90 | 51.04±0.18 | 33.90±0.70 | 23.27±0.82 | 22.40±0.68 |
| ReNode [7] | 79.91±1.52 | 69.89±0.73 | 82.97±0.12 | 77.95±1.71 | 87.80±0.52 | 50.68±0.15 | 32.92±0.98 | 23.80±0.59 | 22.39±0.62 |
| TAM [49] | 80.50±0.18 | 73.14±0.13 | 84.07±0.12 | 82.35±0.19 | 89.80±0.23 | 52.09±0.06 | 35.64±0.27 | 24.58±0.09 | 22.55±0.06 |
| PASTEL [52] | 80.91±0.36 | 72.73±0.26 | — | 83.24±0.85 | 89.10±0.41 | — | 47.12±2.82 | 33.15±0.66 | 27.56±1.04 |
| TOPOAUC [8] | 79.27±0.52 | 70.08±0.83 | — | 75.35±1.32 | 87.10±0.98 | — | 33.39±2.09 | 22.86±0.36 | 22.56±0.18 |
| HyperIMBA [12] | 79.81±0.78 | 71.78±0.40 | 84.75±0.30 | 83.43±0.65 | 90.65±0.14 | — | 38.30±2.70 | 29.97±1.79 | 25.30±2.56 |

Table D.10: **Balanced Accuracy** score (% ± standard deviation) of **node** classification on manipulated **global topology-imbalanced** graph datasets with changing imbalance levels over 10 runs. "—" denotes out of memory or time limit. Best results shown in **bold** and the runner-ups are underlined.

| Algorithm | Cora 0.81 | CiteSeer 0.74 | PubMed 0.80 | Computers 0.78 | Photo 0.82 | ogbn-arXiv 0.65 | Chameleon 0.23 | Squirrel 0.22 | Actor 0.22 |
|---|---|---|---|---|---|---|---|---|---|
| **Imbalance Ratio: Low** | | | | | | | | | |
| GCN (bb.) [21] | 82.78±0.52 | 67.45±0.81 | 85.17±0.15 | 85.09±1.63 | 89.28±1.00 | 36.76±0.20 | 38.28±0.50 | 25.14±0.57 | 23.78±0.63 |
| ReNode [7] | 82.89±0.82 | 68.52±1.14 | 84.52±0.11 | 86.93±0.83 | 90.03±0.51 | 36.79±0.18 | 38.02±0.50 | 25.36±0.49 | 24.04±0.39 |
| TAM [49] | 82.49±0.11 | 70.97±0.13 | 84.71±0.06 | 89.91±0.08 | 91.81±0.13 | 39.96±0.05 | 39.16±0.14 | 25.63±0.12 | 24.25±0.19 |
| PASTEL [52] | 83.46±0.32 | 71.17±0.28 | — | 90.41±0.38 | 93.56±0.11 | — | 54.27±1.18 | 34.02±0.57 | 26.89±0.43 |
| TOPOAUC [8] | 83.15±0.26 | 68.07±0.21 | — | 76.90±4.12 | 87.93±3.60 | — | 38.16±0.65 | 23.51±0.78 | 23.96±0.33 |
| HyperIMBA [12] | 82.32±0.41 | 70.75±0.56 | 86.45±0.11 | 91.54±0.30 | 92.36±0.16 | — | 43.39±1.43 | 32.68±0.66 | 28.45±3.43 |
| **Imbalance Ratio: High** | | | | | | | | | |
| GCN (bb.) [21] | 81.99±0.51 | 64.66±0.91 | 84.22±0.13 | 83.42±1.65 | 87.36±1.02 | 34.78±0.21 | 34.75±0.67 | 23.27±0.82 | 22.52±0.42 |
| ReNode [7] | 82.28±0.71 | 66.04±0.52 | 83.85±0.09 | 85.43±0.84 | 88.10±0.52 | 34.75±0.19 | 33.87±0.77 | 23.80±0.59 | 22.68±0.37 |
| TAM [49] | 82.87±0.13 | 69.81±0.118 | 84.59±0.08 | 87.88±0.09 | 89.70±0.14 | 37.92±0.06 | 36.18±0.35 | 24.58±0.09 | 23.15±0.12 |
| PASTEL [52] | 83.36±0.20 | 69.71±0.23 | — | 88.92±0.39 | 91.40±0.12 | — | 47.41±2.27 | 33.15±0.66 | 25.55±0.57 |
| TOPOAUC [8] | 82.28±0.35 | 65.82±1.20 | — | 74.75±4.13 | 86.00±3.65 | — | 34.45±1.56 | 22.85±0.36 | 23.27±0.26 |
| HyperIMBA [12] | 82.54±0.76 | 68.97±0.38 | 85.64±0.12 | 89.74±0.31 | 90.25±0.17 | — | 38.00±3.16 | 29.96±1.79 | 26.77±2.45 |

Table D.11: **Macro-F1** score (% ± standard deviation) of **node** classification on manipulated **global topology-imbalanced** graph datasets with changing imbalance levels over 10 runs. "—" denotes out of memory or time limit. The best results are shown in **bold** and the runner-ups are underlined.

| Algorithm | Cora 0.81 | CiteSeer 0.74 | PubMed 0.80 | Computers 0.78 | Photo 0.82 | ogbn-arXiv 0.65 | Chameleon 0.23 | Squirrel 0.22 | Actor 0.22 |
|---|---|---|---|---|---|---|---|---|---|
| **Imbalance Ratio: Low** | | | | | | | | | |
| GCN (bb.) [21] | 79.45±0.73 | 66.81±1.04 | 83.54±0.19 | 64.78±2.90 | 84.82±1.48 | 33.94±0.21 | 36.69±0.65 | 24.85±0.65 | 23.13±0.55 |
| ReNode [7] | 80.26±0.38 | 68.21±0.81 | 82.98±0.12 | 68.61±1.47 | 86.14±0.66 | 33.88±0.17 | 36.60±0.50 | 24.75±0.61 | 23.33±0.28 |
| TAM [49] | 79.01±0.24 | 70.30±0.13 | 83.22±0.08 | 75.36±0.22 | 88.34±0.22 | 36.75±0.05 | 36.45±0.19 | 23.59±0.27 | 23.44±0.20 |
| PASTEL [52] | 80.79±0.33 | 70.62±0.32 | — | 73.94±1.06 | 88.17±0.41 | — | 53.30±1.30 | 33.54±0.69 | 26.34±0.23 |
| TOPOAUC [8] | 79.75±0.63 | 68.01±1.03 | — | 60.13±3.52 | 84.39±4.10 | — | 36.76±1.01 | 21.50±1.71 | 22.76±0.70 |
| HyperIMBA [12] | 78.85±0.66 | 69.96±0.53 | 84.16±0.26 | 73.55±0.72 | 89.45±0.21 | — | 42.83±1.43 | 30.26±1.87 | 26.41±3.17 |
| **Imbalance Ratio: High** | | | | | | | | | |
| GCN (bb.) [21] | 78.12±1.02 | 64.20±1.32 | 82.55±0.15 | 63.54±2.95 | 82.92±1.50 | 31.85±0.22 | 32.39±1.15 | 22.36±1.62 | 21.66±0.48 |
| ReNode [7] | 78.80±1.23 | 65.50±0.69 | 82.08±0.11 | 66.12±1.48 | 84.20±0.67 | 31.90±0.18 | 30.82±1.71 | 22.77±1.09 | 21.69±0.51 |
| TAM [49] | 79.34±0.23 | 69.13±0.11 | 83.17±0.11 | 73.25±0.23 | 86.25±0.23 | 34.68±0.06 | 33.95±0.38 | 22.52±0.17 | 22.10±0.06 |
| PASTEL [52] | 79.37±0.33 | 68.99±0.25 | — | 72.45±1.07 | 86.25±0.42 | — | 46.59±3.16 | 31.95±1.06 | 25.11±0.67 |
| TOPOAUC [8] | 78.24±0.42 | 65.48±1.05 | — | 58.54±3.54 | 82.50±4.15 | — | 29.95±3.52 | 21.06±1.40 | 22.15±0.18 |
| HyperIMBA [12] | 78.44±0.99 | 68.24±0.30 | 83.83±0.33 | 71.87±0.73 | 87.40±0.22 | — | 37.25±2.98 | 28.82±2.72 | 24.45±2.46 |

Table D.12: **AUC-ROC** score (% ± standard deviation) of **node** classification on manipulated **global topology-imbalanced** graph datasets with changing imbalance levels over 10 runs. "—" denotes out of memory or time limit. The best results are shown in **bold** and the runner-ups are underlined.

| Algorithm | Cora 0.81 | CiteSeer 0.74 | PubMed 0.80 | Computers 0.78 | Photo 0.82 | ogbn-arXiv 0.65 | Chameleon 0.23 | Squirrel 0.22 | Actor 0.22 |
|---|---|---|---|---|---|---|---|---|---|
| **Imbalance Ratio: Low** | | | | | | | | | |
| GCN (bb.) [21] | 94.82±0.72 | 88.43±1.03 | 94.16±0.15 | 94.98±1.18 | 93.61±1.67 | 88.49±0.14 | 66.27±0.87 | 54.98±0.49 | 54.74±0.22 |
| ReNode [7] | 95.16±0.67 | 88.41±0.99 | 94.06±0.40 | 94.91±1.26 | 93.97±1.60 | 88.42±0.16 | 66.08±0.30 | 55.21±0.47 | 54.83±0.20 |
| TAM [49] | 96.75±0.04 | 91.87±0.07 | 94.86±0.01 | 98.17±0.01 | 99.01±0.01 | 92.30±0.02 | 65.00±0.37 | 55.10±0.10 | 55.44±0.24 |
| PASTEL [52] | 97.05±0.07 | 92.90±0.13 | — | 98.45±0.03 | 99.28±0.03 | — | 80.43±0.46 | 63.47±0.47 | 59.43±0.30 |
| TOPOAUC [8] | 97.07±0.20 | 90.67±0.67 | — | 92.28±2.62 | 98.54±0.14 | — | 65.74±0.47 | 52.72±0.34 | 55.21±0.60 |
| HyperIMBA [12] | 96.19±0.19 | 91.50±0.51 | 95.31±0.11 | 98.39±0.06 | 98.71±0.09 | — | 68.48±2.94 | 60.29±1.42 | 59.92±3.31 |
| **Imbalance Ratio: High** | | | | | | | | | |
| GCN (bb.) [21] | 94.97±0.67 | 87.95±1.04 | 93.02±0.17 | 93.05±1.22 | 91.45±1.70 | 86.43±0.15 | 62.31±1.16 | 53.67±0.33 | 53.17±0.29 |
| ReNode [7] | 95.00±0.78 | 87.86±1.00 | 93.23±0.21 | 93.75±1.27 | 92.15±1.62 | 86.42±0.17 | 61.68±1.15 | 54.15±0.49 | 53.02±0.18 |
| TAM [49] | 96.78±0.13 | 92.04±0.12 | 94.88±0.02 | 96.23±0.02 | 97.20±0.02 | 90.25±0.03 | 62.91±0.62 | 54.27±0.06 | 53.49±0.06 |
| PASTEL [52] | 97.31±0.04 | 92.65±0.10 | — | 96.75±0.04 | 97.10±0.04 | — | 76.49±0.97 | 63.78±0.40 | 57.70±0.37 |
| TOPOAUC [8] | 96.54±0.22 | 89.88±0.25 | — | 90.89±2.63 | 96.40±0.15 | — | 60.37±1.82 | 52.77±0.21 | 53.96±0.21 |
| HyperIMBA [12] | 96.57±0.30 | 92.23±0.32 | 94.95±0.13 | 96.85±0.07 | 96.55±0.10 | — | 64.57±1.44 | 58.83±2.05 | 57.20±2.58 |

### D.1.4 Effectiveness of Graph-Level Class-Imbalanced Algorithms

Table D.13: **Accuracy** score (% ± standard deviation) of **graph** classification on manipulated **class-imbalanced** graph datasets with changing imbalance levels over 10 runs. "—" denotes out of memory or time limit. The best results are shown in **bold** and the runner-ups are underlined.

| Algorithm | PTC-MR | FRANKENSTEIN | PROTEINS | D&D | IMDB-B | REDDIT-B | ogbg-molhiv | COLLAB |
|---|---|---|---|---|---|---|---|---|
| $\rho = 1$ (Balanced) | | | | | | | | |
| GIN (bb.) [63] | 50.43±2.69 | 64.27±2.47 | 65.34±2.72 | 64.04±3.79 | 66.05±2.57 | 76.66±4.80 | — | 65.31±3.25 |
| G²GNN [58] | 53.70±3.87 | 63.63±1.16 | 65.50±2.69 | 66.07±2.27 | 61.91±3.77 | 72.34±2.76 | — | 53.82±2.26 |
| TopoImb [71] | 50.91±2.18 | 61.45±3.74 | 55.04±5.13 | 66.57±3.81 | 66.28±1.85 | 73.99±1.18 | — | 65.92±1.36 |
| DataDec [68] | 54.05±4.85 | 66.90±3.36 | 65.24±4.06 | 64.46±1.88 | 64.09±5.75 | 79.29±8.18 | — | 72.24±0.19 |
| ImGKB [54] | 53.48±3.50 | 52.54±6.05 | 69.85±1.95 | 65.45±2.88 | 50.16±0.34 | 50.24±0.29 | — | 39.34±10.88 |
| $\rho = 20$ (Low) | | | | | | | | |
| GIN (bb.) [63] | 47.83±2.95 | 63.38±1.93 | 55.38±3.57 | 51.05±5.07 | 62.31±3.99 | 61.10±4.86 | 60.75±3.79 | 65.01±1.33 |
| G²GNN [58] | 51.88±6.23 | 61.13±1.05 | 63.61±5.03 | 56.29±7.30 | 63.87±4.64 | 69.58±3.59 | 65.00±3.81 | 62.05±3.06 |
| TopoImb [71] | 44.86±3.52 | 49.49±7.14 | 52.12±10.51 | 49.97±7.24 | 59.95±5.19 | 59.67±7.30 | — | 65.88±0.75 |
| DataDec [68] | 55.72±2.88 | 67.99±0.75 | 66.58±1.35 | 63.51±1.62 | 67.92±3.37 | 78.39±5.01 | — | 71.48±1.03 |
| ImGKB [54] | 50.11±5.95 | 40.83±0.02 | 66.60±2.64 | 65.85±3.70 | 47.74±0.29 | 48.57±2.14 | 67.50±2.70 | 51.21±0.10 |
| $\rho = 100$ (High) | | | | | | | | |
| GIN (bb.) [63] | 39.42±1.87 | 56.02±1.43 | 42.50±2.05 | 41.54±6.57 | 53.57±3.21 | 55.56±7.85 | — | 62.00±3.08 |
| G²GNN [58] | 46.52±9.94 | 55.41±3.91 | 52.97±13.44 | 55.38±15.60 | 59.44±6.49 | 63.22±4.67 | — | 62.61±1.14 |
| TopoImb [71] | 39.42±1.24 | 46.45±6.77 | 39.23±4.28 | 39.12±1.62 | 47.75±3.73 | 51.58±4.69 | — | 64.19±1.77 |
| DataDec [68] | 58.69±3.10 | 67.82±1.88 | 61.99±7.15 | 65.77±2.71 | 66.30±6.70 | 77.72±5.12 | — | 71.50±1.15 |
| ImGKB [54] | 44.24±5.65 | 38.34±0.01 | 61.46±10.25 | 59.99±7.57 | 47.08±3.72 | 51.25±5.10 | — | 50.20±0.06 |

Table D.14: **Balanced Accuracy** score (% ± standard deviation) of **graph** classification on manipulated **class-imbalanced** graph datasets with changing imbalance levels over 10 runs. "—" denotes out of memory or time limit. The best results are shown in **bold** and the runner-ups are underlined.

| Algorithm | PTC-MR | FRANKENSTEIN | PROTEINS | D&D | IMDB-B | REDDIT-B | ogbg-molhiv | COLLAB |
|---|---|---|---|---|---|---|---|---|
| $\rho = 1$ (Balanced) | | | | | | | | |
| GIN (bb.) [63] | 51.40±1.64 | 64.33±2.20 | 64.90±1.51 | 63.37±3.26 | 66.05±2.57 | 76.66±4.80 | — | 72.99±1.83 |
| G²GNN [58] | 53.42±3.72 | 63.55±1.22 | 64.39±2.21 | 63.79±2.13 | 61.91±3.77 | 72.34±2.76 | — | 64.17±0.51 |
| TopoImb [71] | 53.99±2.19 | 60.81±2.83 | 59.71±2.03 | 67.19±2.46 | 66.28±1.85 | 73.99±1.18 | — | 51.93±2.88 |
| DataDec [68] | 53.70±4.25 | 65.06±5.28 | 62.39±2.38 | 62.96±1.21 | 64.22±5.73 | 79.80±7.38 | — | 62.31±1.96 |
| ImGKB [54] | 53.34±3.61 | 49.98±0.06 | 68.09±2.49 | 62.33±4.38 | 49.99±0.47 | 50.24±0.29 | — | 33.38±0.10 |
| $\rho = 20$ (Low) | | | | | | | | |
| GIN (bb.) [63] | 51.36±2.25 | 61.90±2.13 | 60.39±2.22 | 57.33±3.16 | 63.54±3.46 | 62.89±4.61 | 60.75±3.79 | 47.03±1.45 |
| G²GNN [58] | 52.75±2.14 | 63.31±1.14 | 66.18±3.06 | 61.09±4.11 | 64.28±5.02 | 70.26±3.41 | 65.00±3.81 | 60.76±2.04 |
| TopoImb [71] | 50.44±3.75 | 56.57±5.30 | 59.88±6.33 | 58.33±4.40 | 61.14±4.56 | 61.34±6.80 | — | 48.95±1.64 |
| DataDec [68] | 53.42±2.33 | 66.10±1.22 | 61.54±2.33 | 61.27±1.04 | 67.95±2.77 | 78.68±4.93 | — | 68.71±1.11 |
| ImGKB [54] | 51.61±3.12 | 50.01±0.02 | 64.72±5.37 | 63.39±1.76 | 50.23±0.28 | 50.07±0.15 | 67.50±2.70 | 33.33±0.00 |
| $\rho = 100$ (High) | | | | | | | | |
| GIN (bb.) [63] | 47.89±2.12 | 54.66±1.53 | 53.30±1.63 | 52.47±4.58 | 57.46±2.77 | 59.54±7.11 | — | 44.67±2.98 |
| G²GNN [58] | 50.76±1.78 | 60.38±1.91 | 51.32±6.16 | 53.17±2.90 | 61.85±5.36 | 65.97±4.07 | — | 54.91±2.09 |
| TopoImb [71] | 49.99±0.93 | 55.82±4.93 | 53.96±2.78 | 52.05±1.31 | 52.82±3.06 | 55.87±4.09 | — | 47.98±1.65 |
| DataDec [68] | 55.22±3.81 | 65.24±2.16 | 60.22±3.13 | 61.60±2.60 | 66.30±4.93 | 77.77±5.49 | — | 68.99±2.04 |
| ImGKB [54] | 51.38±3.53 | 50.00±0.01 | 64.48±5.90 | 59.14±4.10 | 50.07±0.35 | 50.28±0.23 | — | 33.33±0.00 |

Table D.15: **Macro-F1** score (% ± standard deviation) of **graph** classification on manipulated **class-imbalanced** graph datasets with changing imbalance levels over 10 runs. "—" denotes out of memory or time limit. The best results are shown in **bold** and the runner-ups are underlined.

| Algorithm | PTC-MR | FRANKENSTEIN | PROTEINS | D&D | IMDB-B | REDDIT-B | ogbg-molhiv | COLLAB |
|---|---|---|---|---|---|---|---|---|
| $\rho = 1$ (Balanced) | | | | | | | | |
| GIN (bb.) [63] | 48.85±3.33 | **63.86**±2.39 | 63.98±2.09 | 59.87±7.24 | 65.24±3.21 | 76.20±5.00 | — | 60.83±3.61 |
| G²GNN [58] | 52.88±3.70 | 63.28±1.13 | 63.89±2.39 | 63.31±2.42 | 60.99±3.50 | 71.74±3.14 | — | 46.71±2.55 |
| TopoImb [71] | 48.33±3.70 | 58.58±5.21 | 53.13±5.72 | 65.72±3.45 | 65.85±2.29 | 73.41±0.01 | — | 51.32±4.63 |
| DataDec [68] | 52.98±4.65 | 63.64±9.26 | 62.20±3.28 | 62.83±1.20 | 62.42±8.07 | 79.02±8.63 | — | 63.82±1.02 |
| ImGKB [54] | 50.36±5.89 | 34.59±2.65 | 67.98±2.43 | 61.06±7.96 | 35.27±4.74 | 33.97±0.42 | — | 18.66±3.43 |
| $\rho = 20$ (Low) | | | | | | | | |
| GIN (bb.) [63] | 46.63±2.95 | 61.27±2.51 | 55.20±3.70 | 49.89±6.19 | 59.66±5.94 | 56.10±6.75 | 53.49±6.55 | 37.64±1.08 |
| G²GNN [58] | 47.64±7.63 | 61.11±1.06 | 62.89±4.70 | 55.34±8.20 | 61.37±8.64 | 69.03±3.78 | 62.39±6.37 | 55.48±2.78 |
| TopoImb [71] | 40.26±2.06 | 43.64±11.61 | 49.66±12.86 | 47.37±9.53 | 56.78±8.24 | 54.07±11.22 | — | 47.31±0.03 |
| DataDec [68] | 52.31±3.50 | 66.22±1.25 | 61.66±2.33 | 61.12±1.22 | 67.42±3.69 | 77.96±5.66 | — | 68.49±0.60 |
| ImGKB [54] | 44.62±8.86 | 29.05±0.08 | 62.49±8.12 | 62.91±2.34 | 32.70±0.62 | 65.87±3.81 | 65.87±3.81 | 22.58±0.03 |
| $\rho = 100$ (High) | | | | | | | | |
| GIN (bb.) [63] | 35.14±4.01 | 46.81±3.33 | 41.20±2.65 | 38.83±8.36 | 47.51±5.47 | 48.74±11.66 | — | 36.37±0.93 |
| G²GNN [58] | 37.35±7.10 | 54.73±4.33 | 44.16±10.15 | 42.22±11.10 | 56.79±9.27 | 61.15±6.30 | — | 50.73±4.21 |
| TopoImb [71] | 32.16±3.27 | 40.55±10.82 | 34.48±6.61 | 34.24±2.40 | 36.64±6.35 | 41.81±9.90 | — | 45.32±0.02 |
| DataDec [68] | 54.76±4.61 | 64.94±2.97 | 58.47±4.92 | 61.67±2.67 | 64.82±8.14 | 77.39±5.38 | — | 68.87±1.57 |
| ImGKB [54] | 41.35±7.92 | 27.73±0.04 | 59.75±11.28 | 57.16±5.54 | 32.23±1.61 | 34.31±2.68 | — | 22.28±0.02 |

Table D.16: **AUC-ROC** score (% ± standard deviation) of **graph** classification on manipulated **class-imbalanced** graph datasets with changing imbalance levels over 10 runs. "—" denotes out of memory or time limit. The best results are shown in **bold** and the runner-ups are underlined.

| Algorithm | PTC-MR | FRANKENSTEIN | PROTEINS | D&D | IMDB-B | REDDIT-B | ogbg-molhiv | COLLAB |
|---|---|---|---|---|---|---|---|---|
| $\rho = 1$ (Balanced) | | | | | | | | |
| GIN (bb.) [63] | 52.26±2.95 | 69.20±2.35 | 67.91±2.87 | 66.54±5.28 | 73.99±2.41 | **87.61**±4.35 | — | 86.80±1.38 |
| G²GNN [58] | 54.55±5.56 | 68.03±1.39 | 67.09±3.07 | 70.05±2.41 | 64.87±4.48 | 80.22±4.01 | — | 79.90±0.80 |
| TopoImb [71] | 56.93±4.29 | 69.09±1.17 | 69.33±4.09 | 72.86±1.46 | 74.83±1.01 | 81.49±3.03 | — | 85.12±0.39 |
| DataDec [68] | 54.57±6.02 | 72.65±3.55 | 68.80±4.16 | 68.18±1.32 | 72.25±5.65 | 81.26±13.78 | — | 87.29±0.32 |
| ImGKB [54] | 54.09±5.28 | 54.78±2.66 | 73.47±2.00 | 66.54±2.08 | 50.60±1.23 | 74.36±5.01 | — | 50.29±0.86 |
| $\rho = 20$ (Low) | | | | | | | | |
| GIN (bb.) [63] | 51.28±4.16 | 68.74±2.22 | 59.21±3.70 | 54.13±6.72 | 74.38±2.48 | 84.08±4.85 | 76.76±2.18 | 86.05±1.11 |
| G²GNN [58] | 51.90±3.85 | 68.70±1.13 | 69.59±3.54 | 64.49±5.43 | 69.02±3.60 | 77.97±4.62 | 72.07±2.27 | 78.15±2.04 |
| TopoImb [71] | 50.82±3.94 | 67.96±2.26 | 66.77±5.44 | 67.76±4.85 | 72.61±2.10 | 83.30±3.79 | — | 86.31±1.42 |
| DataDec [68] | 53.49±1.94 | 73.98±0.59 | 69.32±1.81 | 67.05±1.47 | 75.40±3.23 | 81.00±8.17 | — | 85.68±2.32 |
| ImGKB [54] | 52.89±4.84 | 53.59±1.12 | 72.57±1.37 | 68.08±2.00 | 51.06±1.08 | 76.25±3.29 | 75.76±2.13 | 50.42±0.97 |
| $\rho = 100$ (High) | | | | | | | | |
| GIN (bb.) [63] | 47.04±3.22 | 64.26±4.79 | 59.00±2.60 | 47.06±5.01 | 65.96±9.53 | 80.58±3.45 | — | 81.04±3.44 |
| G²GNN [58] | 49.32±4.07 | 66.60±1.48 | 52.10±7.10 | 60.76±4.90 | 65.79±7.00 | 72.55±5.64 | — | 75.67±1.51 |
| TopoImb [71] | 48.09±3.94 | 69.48±1.14 | 65.62±3.19 | 61.49±5.30 | 69.12±7.15 | 80.08±5.47 | — | 82.28±2.86 |
| DataDec [68] | 57.66±5.55 | 73.51±1.01 | 64.35±6.86 | 69.11±3.67 | 74.89±4.11 | 82.59±6.05 | — | 87.03±1.21 |
| ImGKB [54] | 52.80±5.19 | 54.03±2.17 | 71.98±2.36 | 63.79±4.04 | 51.04±1.57 | 71.65±6.48 | — | 50.05±0.63 |

Table D.17: **Accuracy** score (% ± standard deviation) of **graph** classification on manipulated **class-imbalanced** graph datasets with changing imbalance levels over 10 runs. "—" denotes out of memory or time limit. The best results are shown in **bold** and the runner-ups are underlined.

| Algorithm | PTC-MR | FRANKENSTEIN | PROTEINS | D&D | IMDB-B | REDDIT-B | COLLAB |
|---|---|---|---|---|---|---|---|
| | | | $\rho = 1$ (Balanced) | | | | |
| GCN (bb.) [21] | 48.62±7.12 | 52.67±6.11 | 65.44±2.73 | 63.97±4.09 | **59.76**±1.78 | 66.65±1.06 | 61.23±3.38 |
| G²GNN [58] | 44.35±4.32 | 52.10±4.27 | **68.82**±2.83 | 61.47±7.70 | 58.35±3.47 | 67.02±2.55 | 49.49±3.18 |
| TopoImb [71] | 50.61±6.39 | 48.03±5.90 | 57.41±5.95 | 56.13±4.61 | 49.52±1.01 | 58.62±6.52 | 58.63±1.93 |
| DataDec [68] | **54.23**±4.16 | **63.46**±4.36 | 67.25±5.25 | 63.10±2.62 | 58.20±4.05 | **69.08**±4.49 | **69.03**±1.89 |
| ImGKB [54] | 52.83±5.45 | 53.97±5.28 | 64.57±9.42 | **64.93**±4.44 | 50.15±0.39 | 50.29±0.19 | 40.44±11.33 |
| | | | $\rho = 20$ (Low) | | | | |
| GCN (bb.) [21] | 43.84±7.03 | 48.59±9.57 | 54.01±2.86 | 58.32±1.51 | 49.06±2.17 | 61.85±3.89 | 53.81±1.88 |
| G²GNN [58] | 47.86±9.03 | 57.28±1.85 | **69.42**±1.80 | 65.65±5.41 | 53.11±3.97 | 66.02±2.08 | 57.72±1.71 |
| TopoImb [71] | 45.90±6.41 | 40.71±0.33 | 38.27±5.28 | 44.63±4.82 | 49.75±4.81 | 54.35±3.13 | 57.72±1.71 |
| DataDec [68] | **55.86**±2.49 | **63.28**±3.54 | 64.23±7.74 | 64.66±2.04 | **57.06**±5.97 | 66.45±5.38 | **71.26**±0.91 |
| ImGKB [54] | 49.49±5.12 | 40.82±0.02 | 68.44±2.58 | **67.44**±3.50 | 47.65±0.23 | 48.58±2.15 | 51.21±0.10 |
| | | | $\rho = 100$ (High) | | | | |
| GCN (bb.) [21] | 40.58±7.61 | 43.03±9.39 | 37.34±3.35 | 42.48±2.61 | 45.09±0.18 | 54.39±5.16 | 50.47±0.38 |
| G²GNN [58] | 41.78±7.61 | 49.50±11.24 | 61.27±6.39 | 44.59±4.72 | 56.40±1.71 | 65.08±3.08 | 58.02±3.29 |
| TopoImb [71] | 40.17±2.21 | 38.50±0.35 | 33.21±0.05 | 36.50±1.44 | 46.18±3.06 | 52.21±3.56 | 61.40±2.44 |
| DataDec [68] | **55.41**±3.37 | **63.82**±6.75 | **66.08**±7.51 | **65.07**±2.40 | **58.58**±5.07 | 65.56±8.61 | 70.48±0.49 |
| ImGKB [54] | 45.98±8.71 | 38.34±0.02 | 65.09±2.81 | 58.21±10.60 | 46.06±2.74 | 51.23±5.09 | 50.20±0.06 |

Table D.18: **Balanced Accuracy** score (% ± standard deviation) of **graph** classification on manipulated **class-imbalanced** graph datasets with changing imbalance levels over 10 runs. "—" denotes out of memory or time limit. The best results are shown in **bold** and the runner-ups are underlined.

| Algorithm | PTC-MR | FRANKENSTEIN | PROTEINS | D&D | IMDB-B | REDDIT-B | COLLAB |
|---|---|---|---|---|---|---|---|
| | | | $\rho = 1$ (Balanced) | | | | |
| GCN (bb.) [21] | 50.07±0.23 | 50.00±0.02 | 64.60±2.26 | 62.63±2.06 | **59.76**±1.78 | 66.65±1.06 | **69.44**±2.08 |
| G²GNN [58] | 50.02±0.06 | 51.82±1.81 | **67.39**±1.76 | 60.38±5.83 | 58.35±3.47 | 67.02±2.55 | 61.96±0.90 |
| TopoImb [71] | 53.71±2.88 | 51.13±1.64 | 61.24±5.25 | 60.56±2.30 | 49.71±0.34 | 58.68±6.48 | 44.78±3.02 |
| DataDec [68] | **54.36**±3.61 | **62.09**±3.25 | 63.92±4.14 | 61.41±2.42 | 58.11±3.96 | **69.08**±4.38 | 62.12±4.15 |
| ImGKB [54] | 53.03±3.56 | 49.98±0.06 | 65.28±5.36 | **62.84**±2.56 | 50.15±0.39 | 50.29±0.19 | 33.35±0.01 |
| | | | $\rho = 20$ (Low) | | | | |
| GCN (bb.) [21] | 49.69±0.68 | 50.73±2.05 | 59.49±1.56 | 60.53±1.43 | 51.35±1.88 | 63.51±3.57 | 37.97±2.15 |
| G²GNN [58] | 49.73±0.87 | 53.76±2.16 | **68.31**±1.72 | **65.61**±2.52 | 53.84±3.33 | 66.54±2.09 | 61.19±1.44 |
| TopoImb [71] | 51.60±3.14 | 49.98±0.02 | 51.14±2.23 | 54.87±2.95 | 48.94±4.07 | 56.51±2.84 | 46.73±0.87 |
| DataDec [68] | **54.50**±1.74 | **59.94**±3.73 | 61.80±4.57 | 62.01±1.75 | **56.87**±5.73 | 67.02±5.12 | **67.53**±2.03 |
| ImGKB [54] | 51.70±2.33 | 49.99±0.02 | 67.46±1.22 | 63.70±3.66 | 50.14±0.22 | 50.08±0.16 | 33.34±0.01 |
| | | | $\rho = 100$ (High) | | | | |
| GCN (bb.) [21] | 50.36±1.14 | 50.30±0.61 | 51.78±1.94 | 53.22±2.13 | 50.08±0.16 | 58.47±4.64 | 33.99±0.32 |
| G²GNN [58] | 49.94±1.47 | 50.31±0.97 | **65.24**±4.47 | 54.99±3.11 | **58.87**±1.18 | 65.09±3.02 | 53.69±3.41 |
| TopoImb [71] | 50.78±0.68 | 49.99±0.02 | 50.01±0.03 | 49.47±1.20 | 51.02±2.25 | 56.40±2.67 | 46.42±3.04 |
| DataDec [68] | **53.28**±2.65 | **58.87**±3.26 | 63.07±5.35 | **60.83**±2.61 | 58.11±4.26 | **66.61**±6.89 | **67.66**±1.56 |
| ImGKB [54] | 50.19±3.07 | 49.99±0.02 | 64.52±4.98 | 60.22±4.56 | 50.06±0.31 | 50.26±0.22 | 33.33±0.01 |

Table D.19: **Macro-F1** score (% ± standard deviation) of **graph** classification on manipulated **class-imbalanced** graph datasets with changing imbalance levels over 10 runs. "—" denotes out of memory or time limit. The best results are shown in **bold** and the runner-ups are underlined.

| Algorithm | PTC-MR | FRANKENSTEIN | PROTEINS | D&D | IMDB-B | REDDIT-B | COLLAB |
|---|---|---|---|---|---|---|---|
| | | | $\rho = 1$ (Balanced) | | | | |
| GCN (bb.) [21] | 32.82±3.29 | 34.45±2.70 | 64.08±2.43 | 62.02±2.84 | **59.32**±2.48 | **67.58**±1.06 | 56.55±4.31 |
| G²GNN [58] | 31.29±2.68 | 45.92±8.09 | **67.11**±2.26 | 56.99±10.85 | 56.41±7.64 | 65.63±3.26 | 41.15±4.18 |
| TopoImb [71] | 45.01±8.36 | 37.18±7.96 | 56.62±6.11 | 55.05±4.87 | 33.04±0.43 | 50.17±12.11 | 40.73±3.28 |
| DataDec [68] | **53.24**±3.23 | **61.26**±4.55 | 64.05±4.48 | 61.35±2.55 | 54.26±7.60 | 67.37±6.20 | **62.14**±3.48 |
| ImGKB [54] | 48.53±8.31 | 35.15±2.37 | 62.33±11.90 | **62.34**±3.64 | 33.84±0.60 | 33.97±0.41 | 18.99±3.59 |
| | | | $\rho = 20$ (Low) | | | | |
| GCN (bb.) [21] | 31.49±5.35 | 34.14±7.63 | 53.81±2.94 | 58.06±1.46 | 36.70±6.23 | 57.56±6.17 | 42.17±5.55 |
| G²GNN [58] | 33.06±4.46 | 51.94±5.20 | **67.58**±1.71 | **64.18**±4.21 | 45.81±10.04 | **65.51**±2.39 | 47.61±4.89 |
| TopoImb [71] | 39.66±9.73 | 28.90±0.15 | 30.36±7.82 | 40.50±7.73 | 37.79±5.88 | 45.50±5.89 | 41.59±1.32 |
| DataDec [68] | **54.16**±1.97 | **57.35**±6.49 | 60.87±6.00 | 61.94±1.91 | **50.18**±11.79 | 64.62±6.82 | **67.94**±1.44 |
| ImGKB [54] | 45.80±7.16 | 29.04±0.10 | 66.63±1.90 | 63.41±4.10 | 32.52±0.48 | 32.82±1.24 | 22.58±0.03 |
| | | | $\rho = 100$ (High) | | | | |
| GCN (bb.) [21] | 29.74±6.39 | 31.00±6.60 | 32.66±5.33 | 40.59±3.18 | 31.21±0.36 | 47.46±8.81 | 35.27±2.66 |
| G²GNN [58] | 32.13±7.08 | 34.53±7.98 | 60.58±5.98 | 42.79±6.25 | **54.41**±2.63 | 64.77±3.25 | 45.03±5.63 |
| TopoImb [71] | 32.03±3.31 | 27.74±0.17 | 24.89±0.07 | 31.98±1.84 | 33.79±6.25 | 44.46±6.18 | 42.64±3.76 |
| DataDec [68] | **52.48**±3.00 | **56.32**±7.38 | **62.23**±5.92 | **60.89**±2.60 | 53.87±8.76 | 62.46±11.54 | **67.68**±0.43 |
| ImGKB [54] | 39.08±8.03 | 27.76±0.09 | 61.19±7.33 | 55.46±11.43 | 31.73±0.22 | 34.27±2.64 | 22.28±0.02 |

Table D.20: **AUC-ROC** score (% ± standard deviation) of **graph** classification on manipulated **class-imbalanced** graph datasets with changing imbalance levels over 10 runs. "—" denotes out of memory or time limit. The best results are shown in **bold** and the runner-ups are underlined.

| Algorithm | PTC-MR | FRANKENSTEIN | PROTEINS | D&D | IMDB-B | REDDIT-B | COLLAB |
|---|---|---|---|---|---|---|---|
| | | | $\rho = 1$ (Balanced) | | | | |
| GCN (bb.) [21] | 49.33±7.50 | 47.99±6.53 | 67.82±1.87 | 67.24±2.12 | **63.97**±2.24 | 75.18±0.08 | **82.29**±1.74 |
| G²GNN [58] | 45.43±5.01 | 53.96±1.98 | 70.27±2.46 | 68.38±3.80 | 61.49±6.19 | 74.01±3.96 | 76.39±1.05 |
| TopoImb [71] | 56.98±4.51 | 57.57±1.02 | 69.62±6.65 | **70.29**±1.96 | 48.80±10.38 | **77.48**±1.84 | 74.54±1.33 |
| DataDec [68] | 55.99±4.78 | **69.51**±3.35 | 69.52±7.69 | 66.85±3.20 | 62.53±3.42 | 66.49±9.00 | 77.38±4.14 |
| ImGKB [54] | 53.61±5.87 | 54.93±1.84 | **72.75**±1.17 | 67.82±1.50 | 51.83±1.56 | 72.98±5.95 | 49.92±0.89 |
| | | | $\rho = 20$ (Low) | | | | |
| GCN (bb.) [21] | 45.84±4.13 | 61.32±4.03 | 67.36±1.94 | 65.26±2.20 | 62.50±1.26 | 75.19±1.08 | 81.28±1.49 |
| G²GNN [58] | 48.93±5.12 | 54.45±3.61 | 71.85±1.86 | 71.03±1.85 | 57.06±4.74 | 73.96±2.66 | 76.39±1.05 |
| TopoImb [71] | 55.07±4.49 | 57.41±1.16 | 72.43±5.51 | 72.19±3.37 | 50.00±10.14 | **77.58**±0.54 | 77.08±0.50 |
| DataDec [68] | **56.27**±2.77 | **68.73**±2.50 | 67.18±7.60 | 68.13±2.14 | **64.08**±3.48 | 65.24±11.34 | **87.01**±0.67 |
| ImGKB [54] | 52.91±3.96 | 54.52±2.20 | 71.14±3.19 | 68.34±2.86 | 51.08±0.71 | 76.70±3.52 | 50.16±0.94 |
| | | | $\rho = 100$ (High) | | | | |
| GCN (bb.) [21] | 44.65±4.74 | 46.38±5.76 | 65.59±1.10 | 64.67±1.98 | 56.60±8.91 | 76.80±2.42 | 77.77±2.43 |
| G²GNN [58] | 46.82±4.30 | 49.91±1.64 | 68.51±3.63 | 67.30±2.70 | 62.46±2.16 | 72.42±3.79 | 70.81±3.61 |
| TopoImb [71] | 53.31±5.12 | 56.71±0.63 | **70.74**±4.14 | 57.78±6.66 | 52.42±10.28 | **77.07**±0.24 | 75.96±0.51 |
| DataDec [68] | **55.65**±3.63 | **68.13**±2.76 | 69.26±8.47 | **67.68**±2.86 | **63.17**±4.96 | 64.92±15.31 | **87.21**±0.35 |
| ImGKB [54] | 51.01±4.09 | 53.74±0.93 | 70.71±1.27 | 65.42±3.01 | 51.05±1.00 | 71.01±5.58 | 50.01±0.58 |

### D.1.5 EFFECTIVENESS OF GRAPH-LEVEL TOPOLOGY-IMBALANCED ALGORITHMS

Table D.21: **Accuracy** score (% ± standard deviation) of **graph** classification on manipulated **topology-imbalanced** graph datasets with changing imbalance levels over 10 runs. "—" denotes out of memory or time limit. The best results are shown in **bold**.

| Algorithm | PTC-MR | FRANKENSTEIN | PROTEINS | D&D | IMDB-B | REDDIT-B | ogbg-molhiv | COLLAB |
|---|---|---|---|---|---|---|---|---|
| **Imbalance Ratio: Low** | | | | | | | | |
| GIN (bb.) [63] | **52.17**±4.36 | 54.96±1.00 | 63.58±1.76 | 65.01±0.69 | **66.38**±4.27 | 67.41±2.32 | 59.35±2.58 | **64.34**±3.55 |
| SOLT-GNN [32] | 47.54±4.33 | **59.98**±1.11 | **65.56**±4.83 | 63.78±1.06 | 64.20±5.46 | 49.57±6.78 | — | 61.79±2.09 |
| TopoImb [71] | 49.71±1.98 | 53.13±0.45 | 61.19±4.61 | **65.16**±1.76 | 66.00±2.41 | **69.47**±3.70 | — | 62.71±2.42 |
| **Imbalance Ratio: Mid** | | | | | | | | |
| GIN (bb.) [63] | 51.38±6.78 | 54.82±2.26 | 62.14±2.43 | 61.46±2.43 | 65.08±5.78 | 68.32±1.77 | 57.67±3.12 | 65.84±3.12 |
| SOLT-GNN [32] | **53.04**±3.91 | **68.71**±1.60 | **71.95**±2.36 | 63.33±1.86 | **69.38**±1.23 | **73.51**±1.14 | — | **69.69**±2.45 |
| TopoImb [71] | 51.59±4.30 | 54.52±0.87 | 64.03±4.43 | **65.99**±1.25 | 68.10±0.87 | 71.54±0.75 | — | 68.68±1.34 |
| **Imbalance Ratio: High** | | | | | | | | |
| GIN (bb.) [63] | 48.41±7.07 | 53.99±7.96 | 58.00±4.19 | 60.68±6.89 | 62.60±3.82 | 67.41±2.23 | 56.69±2.87 | 67.05±2.46 |
| SOLT-GNN [32] | 51.74±5.25 | **67.88**±2.37 | **72.04**±2.18 | **64.97**±3.24 | 65.03±4.12 | 60.24±2.11 | — | 67.12±3.28 |
| TopoImb [71] | **51.96**±1.16 | 56.32±0.61 | 54.89±13.58 | 64.16±2.96 | **66.75**±0.91 | **69.14**±4.83 | — | **67.52**±0.77 |

Table D.22: **Balanced Accuracy** score (% ± standard deviation) of **graph** classification on manipulated **topology-imbalanced** graph datasets with changing imbalance levels over 10 runs. "—" denotes out of memory or time limit. The best results are shown in **bold**.

| Algorithm | PTC-MR | FRANKENSTEIN | PROTEINS | D&D | IMDB-B | REDDIT-B | ogbg-molhiv | COLLAB |
|---|---|---|---|---|---|---|---|---|
| **Imbalance Ratio: Low** | | | | | | | | |
| GIN (bb.) [63] | **51.08**±2.07 | 55.22±1.39 | 59.76±1.78 | 63.46±0.92 | **66.38**±4.27 | 67.41±2.32 | 58.63±2.23 | 64.73±5.42 |
| SOLT-GNN [32] | 45.47±2.10 | **63.58**±0.82 | **66.60**±3.25 | 62.58±0.61 | 64.20±5.46 | 49.57±6.78 | — | 70.23±1.56 |
| TopoImb [71] | 50.07±2.77 | 54.44±0.41 | 63.87±1.68 | **67.24**±1.53 | 66.00±2.41 | **69.47**±3.70 | — | **71.56**±1.68 |
| **Imbalance Ratio: Mid** | | | | | | | | |
| GIN (bb.) [63] | 49.83±1.30 | 54.43±2.04 | 56.84±2.54 | 62.22±0.98 | 65.08±5.78 | 68.32±1.77 | 56.85±3.10 | 74.40±1.72 |
| SOLT-GNN [32] | 50.06±0.75 | **69.36**±0.87 | **70.88**±1.67 | 59.61±3.40 | **69.38**±1.23 | **73.51**±1.14 | — | 75.86±1.12 |
| TopoImb [71] | **50.88**±2.36 | 54.60±1.03 | 59.45±4.27 | **65.25**±2.43 | 68.10±0.87 | 71.54±0.75 | — | **76.54**±0.54 |
| **Imbalance Ratio: High** | | | | | | | | |
| GIN (bb.) [63] | 50.20±0.69 | 53.65±3.89 | 55.82±1.76 | 61.43±2.36 | 62.60±3.82 | 67.41±2.23 | 56.22±2.37 | 74.99±0.95 |
| SOLT-GNN [32] | 48.02±1.77 | **67.98**±2.19 | **71.07**±2.22 | 60.75±6.03 | 65.03±4.12 | 60.24±2.11 | — | 73.58±2.12 |
| TopoImb [71] | **52.07**±0.96 | 56.84±0.41 | 55.47±5.17 | **66.48**±1.89 | **66.75**±0.91 | **69.14**±4.83 | — | **76.10**±0.32 |

Table D.23: **Macro-F1** score (% ± standard deviation) of **graph** classification on manipulated **topology-imbalanced** graph datasets with changing imbalance levels over 10 runs. "—" denotes out of memory or time limit. The best results are shown in **bold**.

| Algorithm | PTC-MR | FRANKENSTEIN | PROTEINS | D&D | IMDB-B | REDDIT-B | ogbg-molhiv | COLLAB |
|---|---|---|---|---|---|---|---|---|
| **Imbalance Ratio: Low** | | | | | | | | |
| GIN (bb.) [63] | 48.26±3.11 | 54.63±0.98 | 59.67±1.87 | 63.39±0.84 | 65.67±5.40 | 66.14±1.90 | 58.13±4.10 | 58.45±1.80 |
| SOLT-GNN [32] | 43.63±1.91 | **58.54**±1.50 | **64.60**±4.66 | 62.34±0.66 | 63.66±5.94 | 49.17±6.76 | — | 59.76±1.81 |
| TopoImb [71] | **48.87**±2.90 | 53.07±0.49 | 59.94±4.32 | **64.97**±1.65 | **65.71**±2.62 | **69.15**±3.83 | — | **60.91**±2.15 |
| **Imbalance Ratio: Mid** | | | | | | | | |
| GIN (bb.) [63] | 38.58±5.97 | 50.50±5.43 | 56.46±2.90 | 60.89±1.84 | 63.28±9.06 | 67.20±2.32 | 55.05±3.86 | 63.79±2.55 |
| SOLT-GNN [32] | 43.20±5.36 | **68.52**±1.49 | **70.58**±2.03 | 58.67±4.91 | **68.73**±1.97 | **73.24**±1.40 | — | **66.96**±1.86 |
| TopoImb [71] | **49.17**±0.95 | 54.25±0.91 | 59.03±4.35 | **64.65**±1.76 | 67.98±0.95 | 71.52±0.75 | — | 66.58±1.09 |
| **Imbalance Ratio: High** | | | | | | | | |
| GIN (bb.) [63] | 34.56±6.32 | 43.71±10.57 | 53.48±2.03 | 57.98±5.51 | 59.75±6.69 | 66.20±2.77 | 54.38±4.67 | 64.92±2.18 |
| SOLT-GNN [32] | 40.70±3.27 | **67.54**±2.28 | **70.70**±2.20 | 58.50±10.48 | 64.53±4.68 | 54.80±3.23 | — | 64.68±2.81 |
| TopoImb [71] | **51.65**±1.07 | 56.18±0.53 | 44.79±14.19 | **63.97**±2.78 | **66.67**±0.91 | **68.41**±5.34 | — | **65.65**±0.63 |

Table D.24: **AUC-ROC** score (% ± standard deviation) of **graph** classification on manipulated **topology-imbalanced** graph datasets with changing imbalance levels over 10 runs. "—" denotes out of memory or time limit. The best results are shown in **bold**.

| Algorithm | PTC-MR | FRANKENSTEIN | PROTEINS | D&D | IMDB-B | REDDIT-B | ogbg-molhiv | COLLAB |
|---|---|---|---|---|---|---|---|---|
| | | | **Imbalance Ratio: Low** | | | | | |
| GIN (bb.) [63] | **51.36**±2.71 | 57.38±1.46 | 60.17±2.84 | 66.09±0.71 | 71.34±4.89 | **78.51**±9.44 | 63.71±3.82 | 77.46±2.90 |
| SOLT-GNN [32] | 45.70±3.17 | **72.90**±0.24 | 70.79±7.53 | 68.55±0.93 | 71.29±6.08 | 48.18±7.14 | — | **82.66**±1.17 |
| TopoImb [71] | 49.33±1.90 | 55.80±0.30 | **71.26**±3.18 | **73.92**±3.23 | **72.37**±2.68 | 65.86±4.55 | — | 82.06±1.77 |
| | | | **Imbalance Ratio: Mid** | | | | | |
| GIN (bb.) [63] | 48.84±2.20 | 54.01±8.26 | 53.62±7.67 | 67.45±0.86 | 68.30±6.16 | **78.44**±2.37 | 61.33±4.06 | 86.00±1.23 |
| SOLT-GNN [32] | 50.40±2.91 | **75.83**±0.29 | **76.11**±2.55 | 68.57±2.72 | **76.53**±1.52 | 76.14±4.50 | — | 87.51±0.66 |
| TopoImb [71] | **52.09**±1.56 | 55.67±1.09 | 63.96±8.30 | **72.48**±2.37 | 73.33±2.00 | 74.96±1.51 | — | **88.27**±0.10 |
| | | | **Imbalance Ratio: High** | | | | | |
| GIN (bb.) [63] | 49.85±1.83 | 56.40±9.37 | 52.96±7.75 | 70.66±0.82 | 69.92±3.25 | **75.86**±5.42 | 58.96±4.75 | 85.97±0.28 |
| SOLT-GNN [32] | 47.82±3.75 | **73.62**±1.58 | **76.43**±2.46 | 73.57±5.28 | 70.85±5.29 | 43.84±5.55 | — | 85.61±1.65 |
| TopoImb [71] | **52.26**±1.24 | 58.29±0.54 | 61.66±10.19 | **73.84**±1.92 | **72.55**±1.25 | 69.04±9.48 | — | **87.96**±0.71 |

Table D.25: **Accuracy** score (% ± standard deviation) of **graph** classification on manipulated **topology-imbalanced** graph datasets with changing imbalance levels over 10 runs. "—" denotes out of memory or time limit. The best results are shown in **bold**.

| Algorithm | PTC-MR | FRANKENSTEIN | PROTEINS | D&D | IMDB-B | REDDIT-B | COLLAB |
|---|---|---|---|---|---|---|---|
| | | | **Imbalance Ratio: Low** | | | | |
| GCN (bb.) [21] | **51.59**±7.07 | 50.53±0.87 | **65.33**±1.85 | 64.82±0.75 | **69.15**±1.44 | 53.44±1.78 | **64.29**±1.42 |
| SOLT-GNN [32] | 51.45±3.13 | **59.25**±0.91 | 65.24±4.77 | **65.69**±2.36 | 68.23±2.48 | 37.58±3.04 | 63.82±2.64 |
| TopoImb [71] | 46.74±1.57 | 54.36±0.48 | 59.73±5.13 | 56.18±2.10 | 68.20±0.70 | **56.94**±3.23 | 60.47±0.91 |
| | | | **Imbalance Ratio: Mid** | | | | |
| GCN (bb.) [21] | 50.00±5.87 | 50.10±2.55 | 64.32±2.81 | 63.99±2.16 | 67.95±2.82 | 67.96±0.89 | **67.64**±2.04 |
| SOLT-GNN [32] | **56.23**±0.84 | **66.42**±1.37 | **68.91**±1.89 | 62.78±1.33 | **70.25**±1.15 | 61.95±5.55 | 66.94±3.67 |
| TopoImb [71] | 54.13±4.62 | 55.38±0.90 | 49.72±13.46 | **64.73**±7.09 | 68.75±0.76 | **69.12**±0.52 | 66.48±1.03 |
| | | | **Imbalance Ratio: High** | | | | |
| GCN (bb.) [21] | 49.93±5.90 | 51.12±1.01 | 58.02±5.02 | 60.98±6.71 | 64.88±2.02 | 66.38±0.46 | **68.99**±1.36 |
| SOLT-GNN [32] | **54.78**±6.03 | **68.26**±0.28 | **67.38**±2.89 | 63.21±3.40 | **69.80**±2.07 | 67.05±2.54 | 65.57±5.59 |
| TopoImb [71] | 51.81±1.26 | 55.20±0.59 | 54.31±13.19 | **69.66**±1.92 | 66.60±0.91 | **69.09**±1.00 | 67.74±0.63 |

Table D.26: **Balanced Accuracy** score (% ± standard deviation) of **graph** classification on manipulated **topology-imbalanced** graph datasets with changing imbalance levels over 10 runs. "—" denotes out of memory or time limit. The best results are shown in **bold**.

| Algorithm | PTC-MR | FRANKENSTEIN | PROTEINS | D&D | IMDB-B | REDDIT-B | COLLAB |
|---|---|---|---|---|---|---|---|
| | | | **Imbalance Ratio: Low** | | | | |
| GCN (bb.) [21] | **50.47**±0.75 | 52.46±0.50 | 61.92±2.38 | 63.13±0.43 | **69.15**±1.44 | 53.44±1.78 | **70.14**±1.68 |
| SOLT-GNN [32] | 48.24±1.20 | **63.10**±0.74 | 61.76±3.57 | **63.81**±1.86 | 68.23±2.48 | 37.58±3.04 | 69.49±0.19 |
| TopoImb [71] | 50.43±0.87 | 55.53±0.61 | **62.18**±2.05 | 57.64±0.94 | 68.20±0.70 | **56.94**±3.23 | 69.51±0.91 |
| | | | **Imbalance Ratio: Mid** | | | | |
| GCN (bb.) [21] | 49.74±2.52 | 50.83±0.66 | 61.89±1.10 | 58.27±4.65 | 67.95±2.82 | 67.96±0.89 | **76.18**±0.89 |
| SOLT-GNN [32] | 51.13±0.79 | **66.62**±1.36 | **67.58**±2.10 | 59.12±2.47 | **70.25**±1.15 | 61.95±5.55 | 74.88±1.59 |
| TopoImb [71] | **54.34**±3.22 | 54.62±1.00 | 54.47±6.12 | **64.63**±3.42 | 68.75±0.76 | **69.12**±0.52 | 74.87±0.87 |
| | | | **Imbalance Ratio: High** | | | | |
| GCN (bb.) [21] | 49.96±0.54 | 52.43±1.04 | 52.66±2.17 | 57.58±4.32 | 64.88±2.02 | 66.38±0.46 | 66.80±1.08 |
| SOLT-GNN [32] | 51.11±1.15 | **68.06**±0.41 | **66.34**±2.08 | 60.94±1.68 | **69.80**±2.07 | 67.05±2.54 | 74.16±2.63 |
| TopoImb [71] | **52.12**±0.78 | 55.50±1.13 | 55.34±5.76 | **66.63**±4.54 | 66.60±0.91 | **69.09**±1.00 | **75.02**±0.79 |

Table D.27: **Macro-F1** score (% ± standard deviation) of **graph** classification on manipulated **topology-imbalanced** graph datasets with changing imbalance levels over 10 runs. "—" denotes out of memory or time limit. The best results are shown in **bold**.

| Algorithm | PTC-MR | FRANKENSTEIN | PROTEINS | D&D | IMDB-B | REDDIT-B | COLLAB |
|---|---|---|---|---|---|---|---|
| **Imbalance Ratio: Low** | | | | | | | |
| GCN (bb.) [21] | 36.72±5.58 | 50.16±1.26 | **62.08**±2.40 | 63.09±0.46 | **69.02**±1.52 | 49.68±8.19 | **61.44**±1.38 |
| SOLT-GNN [32] | **45.00**±3.14 | **57.49**±1.18 | 61.41±4.07 | **63.82**±2.01 | 67.88±2.36 | 35.00±2.77 | 61.07±1.80 |
| TopoImb [71] | 44.72±2.61 | 54.31±0.49 | 58.67±6.03 | 55.77±1.91 | 67.78±1.00 | **56.09**±3.68 | 58.79±1.01 |
| **Imbalance Ratio: Mid** | | | | | | | |
| GCN (bb.) [21] | 40.94±5.67 | 48.11±3.33 | 61.59±1.64 | 54.22±8.96 | 67.80±2.84 | 67.18±1.39 | **65.45**±1.75 |
| SOLT-GNN [32] | 45.63±4.84 | **66.16**±1.30 | **67.16**±2.03 | 58.14±3.23 | **70.05**±1.19 | 57.57±8.59 | 64.70±3.08 |
| TopoImb [71] | **51.99**±3.88 | 54.34±0.88 | 40.81±15.22 | **62.10**±6.59 | 68.64±0.77 | **69.10**±0.49 | 64.49±0.93 |
| **Imbalance Ratio: High** | | | | | | | |
| GCN (bb.) [21] | 39.32±7.82 | 50.84±1.33 | 51.29±1.10 | 52.46±8.79 | 63.65±3.15 | 65.91±0.31 | **66.80**±1.08 |
| SOLT-GNN [32] | 39.57±6.16 | **67.89**±0.33 | **65.88**±2.46 | 60.32±2.31 | **69.54**±2.16 | 65.40±3.77 | 63.70±4.83 |
| TopoImb [71] | **51.43**±0.98 | 54.99±0.81 | 44.51±14.79 | **65.66**±5.28 | 66.54±0.89 | **68.76**±1.17 | 65.56±0.52 |

Table D.28: **AUC-ROC** score (% ± standard deviation) of **graph** classification on manipulated **topology-imbalanced** graph datasets with changing imbalance levels over 10 runs. "—" denotes out of memory or time limit. The best results are shown in **bold**.

| Algorithm | PTC-MR | FRANKENSTEIN | PROTEINS | D&D | IMDB-B | REDDIT-B | COLLAB |
|---|---|---|---|---|---|---|---|
| **Imbalance Ratio: Low** | | | | | | | |
| GCN (bb.) [21] | 50.81±2.98 | 53.52±0.69 | 62.80±5.19 | 64.25±0.94 | **74.69**±1.04 | 50.48±13.45 | 81.81±0.88 |
| SOLT-GNN [32] | 46.36±0.64 | **73.17**±0.36 | 59.26±6.01 | **65.61**±1.44 | 74.16±1.36 | 32.50±5.56 | 80.89±1.27 |
| TopoImb [71] | **52.94**±0.50 | 56.71±0.57 | **67.71**±4.70 | 60.66±1.62 | 74.55±1.80 | **63.78**±2.65 | **82.67**±0.42 |
| **Imbalance Ratio: Mid** | | | | | | | |
| GCN (bb.) [21] | 50.03±4.80 | 51.66±0.64 | 62.02±1.93 | 61.45±4.93 | 74.23±3.12 | 74.51±2.91 | **87.46**±0.79 |
| SOLT-GNN [32] | 52.54±0.92 | **72.48**±1.32 | **71.33**±2.35 | 59.75±4.06 | **78.21**±2.21 | 66.93±6.74 | 86.46±0.45 |
| TopoImb [71] | **57.29**±2.45 | 56.98±1.58 | 65.74±3.16 | **74.47**±2.44 | 74.30±0.66 | **77.25**±0.41 | 86.74±0.85 |
| **Imbalance Ratio: High** | | | | | | | |
| GCN (bb.) [21] | 51.12±1.94 | 52.85±2.56 | 47.22±5.45 | 65.50±6.38 | 68.31±3.07 | 73.41±0.82 | 87.56±0.44 |
| SOLT-GNN [32] | 49.47±4.36 | **73.27**±0.46 | **69.84**±3.01 | 64.79±0.57 | **78.04**±1.14 | 72.92±1.77 | 86.77±1.65 |
| TopoImb [71] | **53.65**±2.72 | 57.66±0.95 | 63.26±12.24 | **74.37**±3.19 | 71.84±1.07 | **75.52**±1.17 | **88.42**±0.13 |

## D.2 Additional Results for Algorithm Robustness (**RQ2**)

### D.2.1 Robustness of Node-Level Class-Imbalanced Algorithms

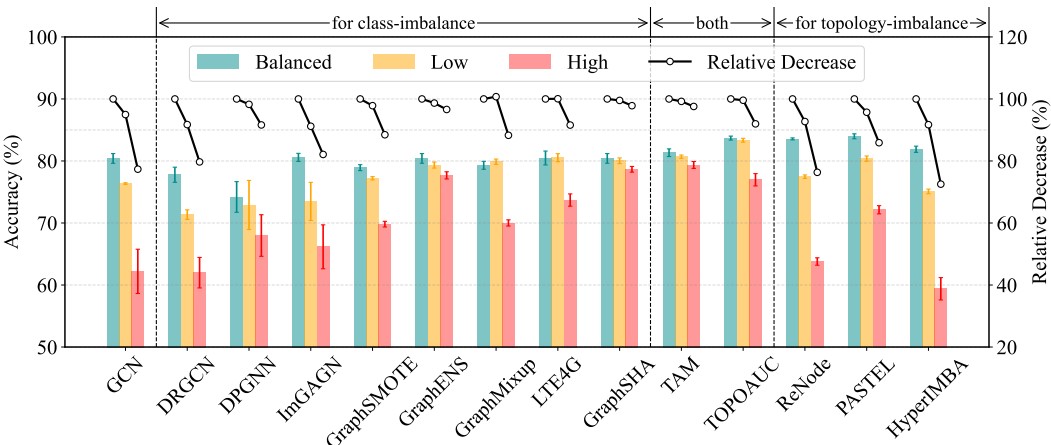

Figure D.1: Robustness analysis of the **node**-level algorithms under different **class-imbalance** degrees on **Cora** (homophilic). Results are reported with the algorithm performance (**Accuracy**) and its relative decrease (%) compared to the class-balanced data split (the green bar).

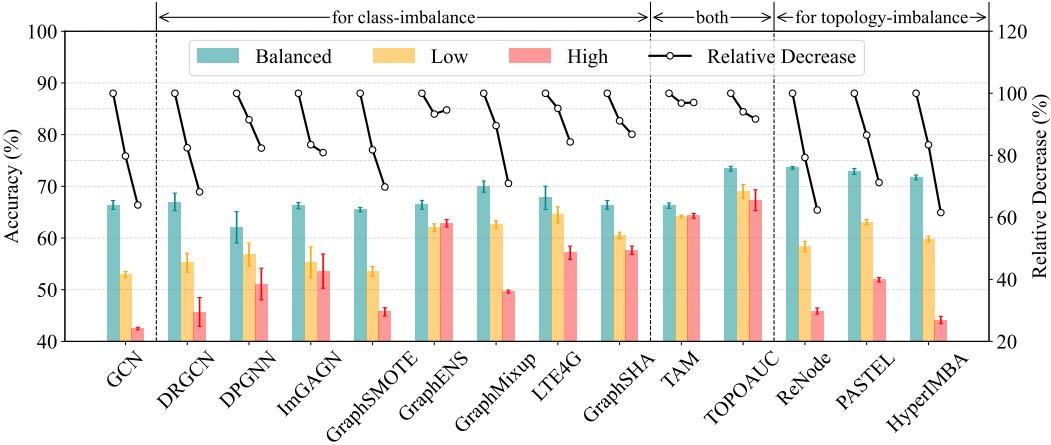

Figure D.2: Robustness analysis of the **node**-level algorithms under different **class-imbalance** degrees on **CiteSeer** (homophilic). Results are reported with the algorithm performance (**Accuracy**) and its relative decrease (%) compared to the class-balanced data split (the green bar).

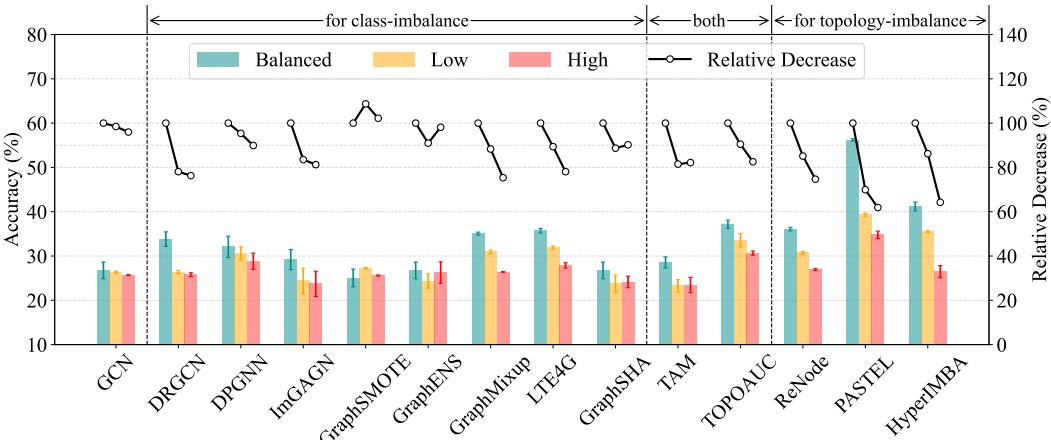

Figure D.3: Robustness analysis of the **node**-level algorithms under different **class-imbalance** degrees on **Chameleon** (heterophilic). Results are reported with the algorithm performance (**Accuracy**) and its relative decrease (%) compared to the class-balanced data split (the green bar).

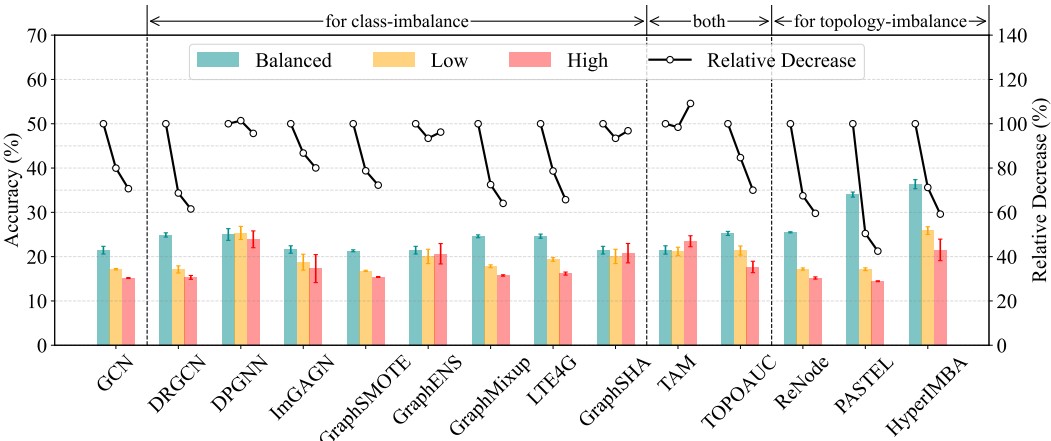

Figure D.4: Robustness analysis of the **node**-level algorithms under different **class-imbalance** degrees on **Squirrel** (heterophilic). Results are reported with the algorithm performance (**Accuracy**) and its relative decrease (%) compared to the class-balanced data split (the green bar).

### D.2.2 ROBUSTNESS OF NODE-LEVEL LOCAL TOPOLOGY-IMBALANCED ALGORITHMS

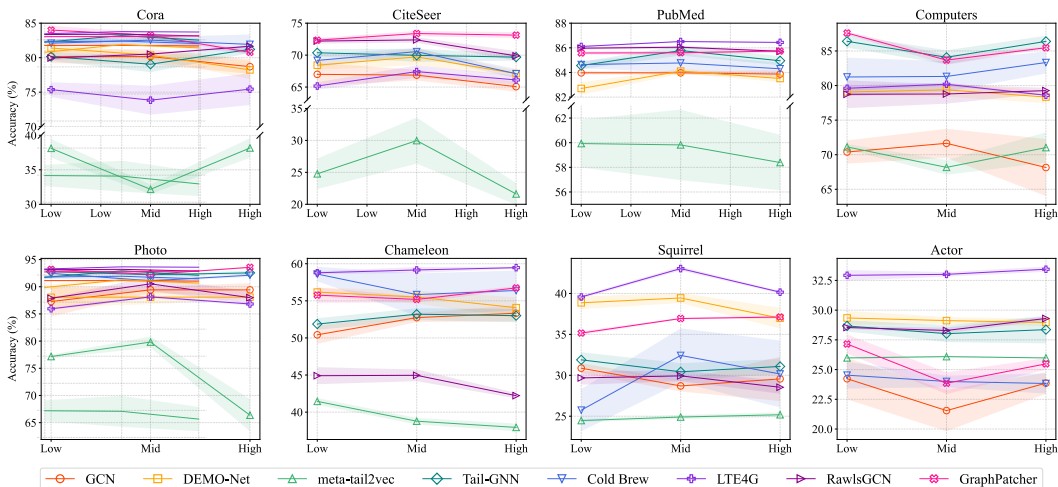

Figure D.5: Robustness analysis of the **node**-level algorithms under different **local topology-imbalance** degrees (Low, Mid, and High). Results are reported with the algorithm performance (**Accuracy**) with the standard deviation error area.

### D.2.3 ROBUSTNESS OF NODE-LEVEL GLOBAL TOPOLOGY-IMBALANCED ALGORITHMS

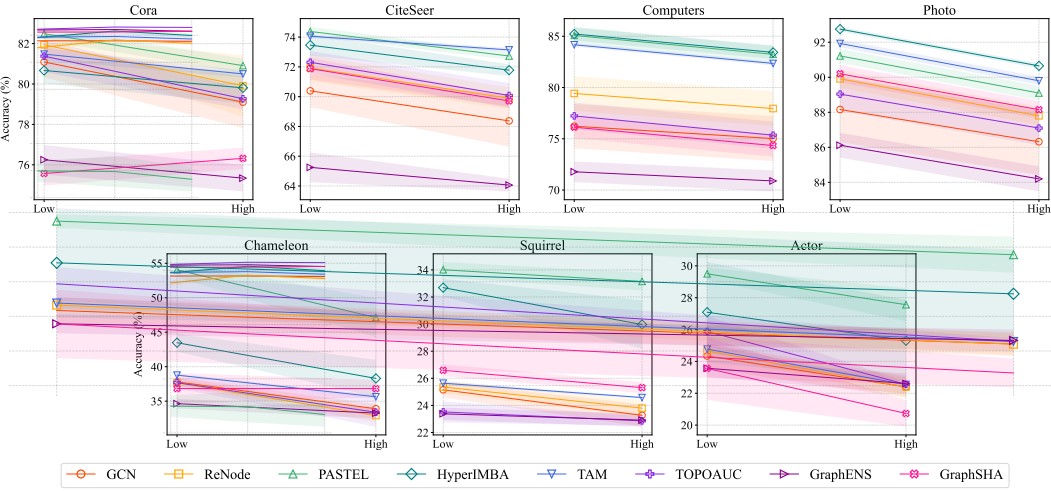

Figure D.6: Robustness analysis of the **node**-level algorithms under different **global topology-imbalance** degrees (Low, Mid, and High). Results are reported with the algorithm performance (**Accuracy**) with the standard deviation error area.

### D.2.4 ROBUSTNESS OF GRAPH-LEVEL CLASS-IMBALANCED ALGORITHMS

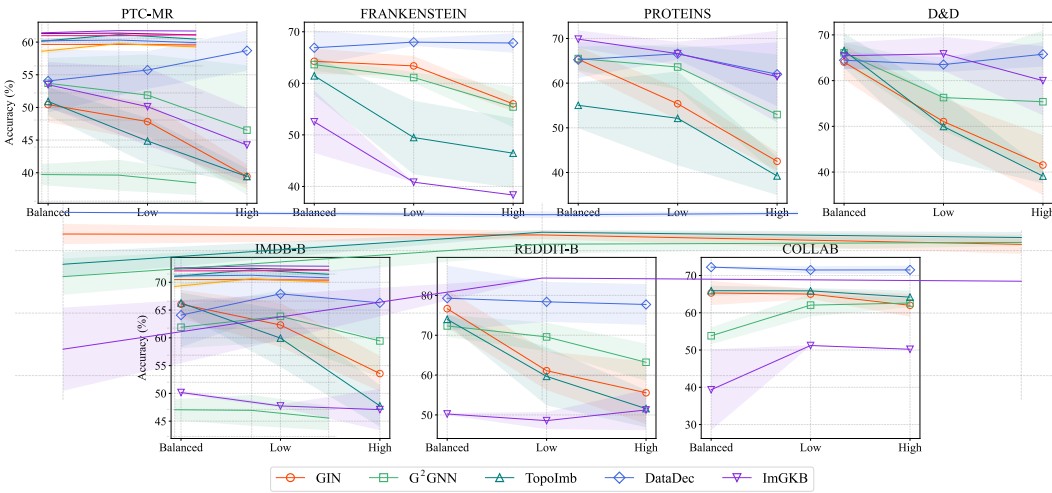

Figure D.7: Robustness analysis of the **graph**-level algorithms under different **class-imbalance** degrees (Low, Mid, and High). Results are reported with the algorithm performance (**Accuracy**) with the standard deviation error area.

### D.2.5 ROBUSTNESS OF GRAPH-LEVEL TOPOLOGY-IMBALANCED ALGORITHMS

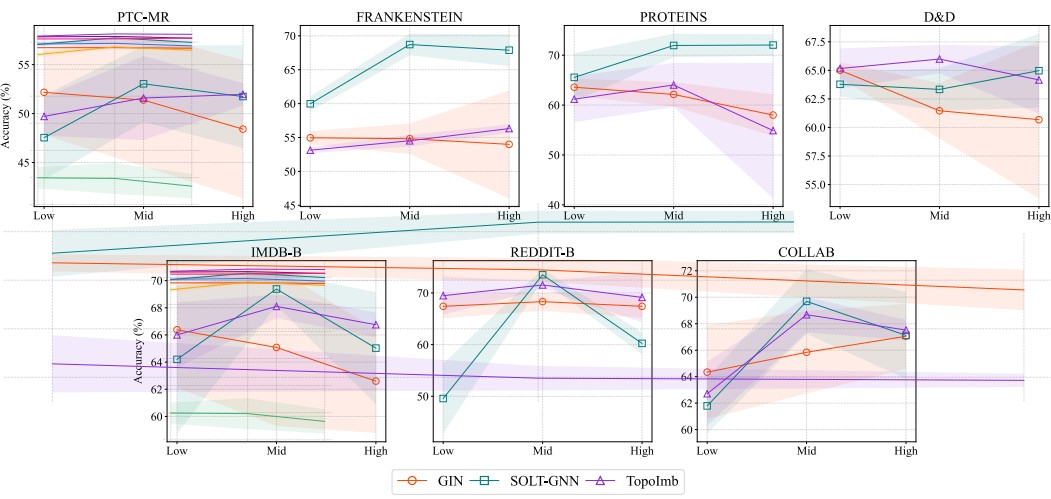

Figure D.8: Robustness analysis of the **graph**-level algorithms under different **topology-imbalance** degrees (Low, Mid, and High). Results are reported with the algorithm performance (**Accuracy**) with the standard deviation error area.

## D.3 ADDITIONAL RESULTS FOR VISUALIZATIONS (**RQ3**)

### D.3.1 VISUALIZATIONS OF NODE-LEVEL CLASS-IMBALANCED ALGORITHMS

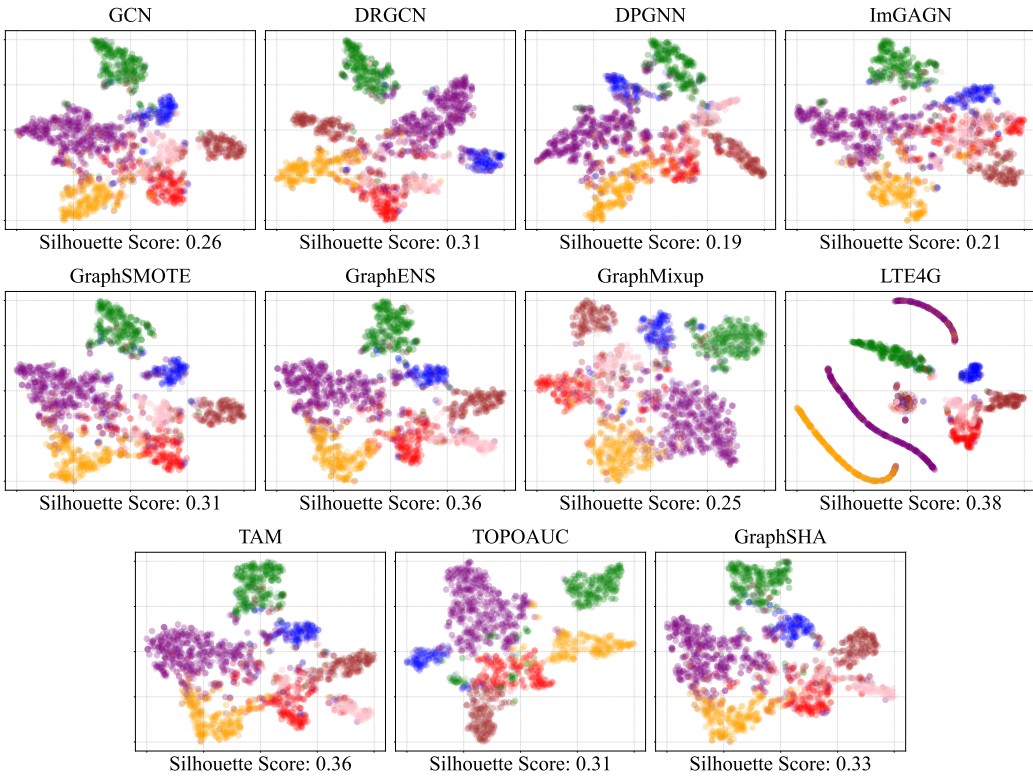

Figure D.9: Visualizations of the embedding for **node**-level **class**-imbalanced algorithms.

### D.3.2 VISUALIZATIONS OF NODE-LEVEL LOCAL TOPOLOGY-IMBALANCED ALGORITHMS

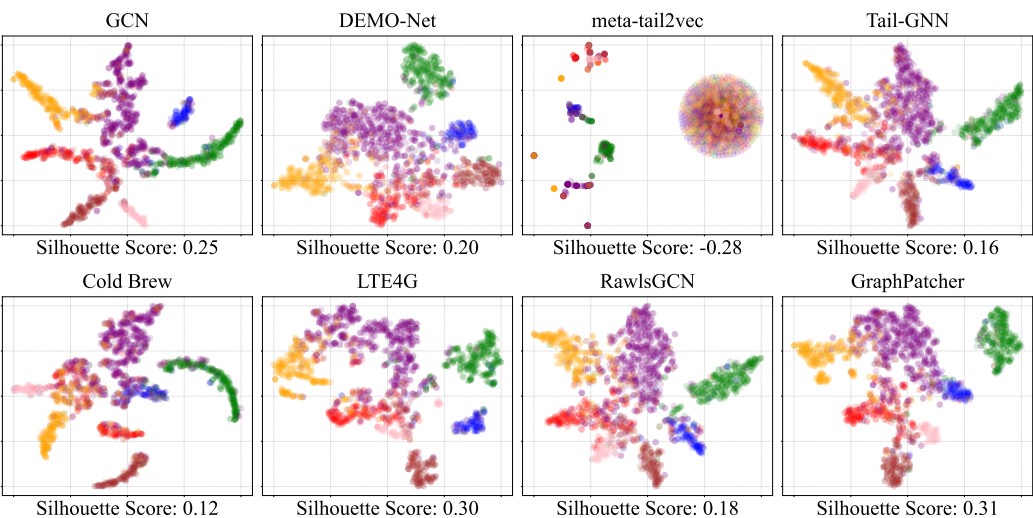

Figure D.10: Visualizations of the embedding for **node**-level **local topology**-imbalanced algorithms.

### D.3.3    VISUALIZATIONS OF NODE-LEVEL GLOBAL TOPOLOGY-IMBALANCED ALGORITHMS

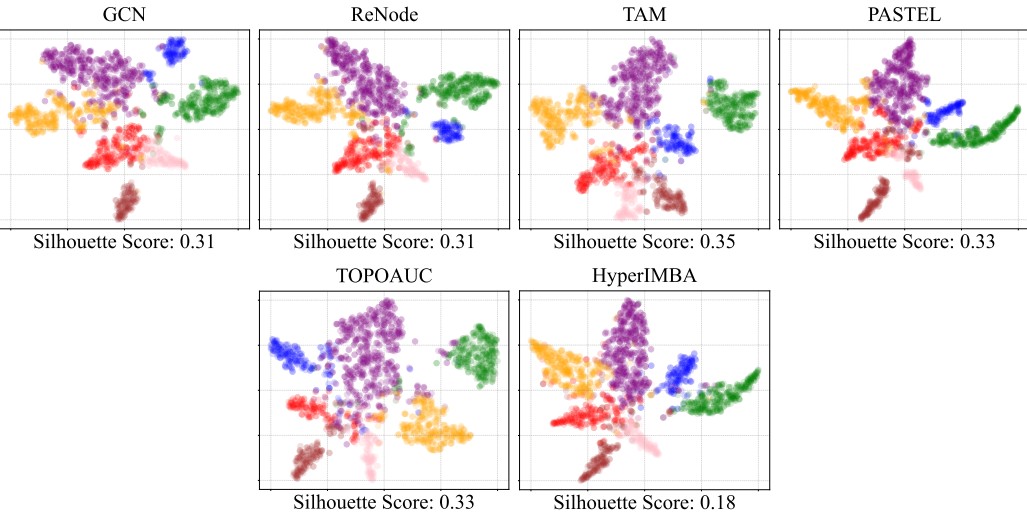

Figure D.11: Visualizations of the embedding for **node**-level **global topology**-imbalanced algorithms.

### D.3.4    VISUALIZATIONS OF GRAPH-LEVEL CLASS-IMBALANCED ALGORITHMS

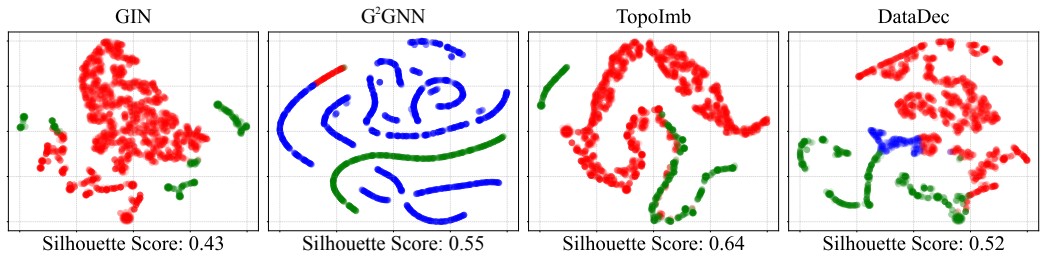

Figure D.12: Visualizations of the embedding for **graph**-level **class**-imbalanced algorithms.

### D.3.5    VISUALIZATIONS OF GRAPH-LEVEL TOPOLOGY-IMBALANCED ALGORITHMS

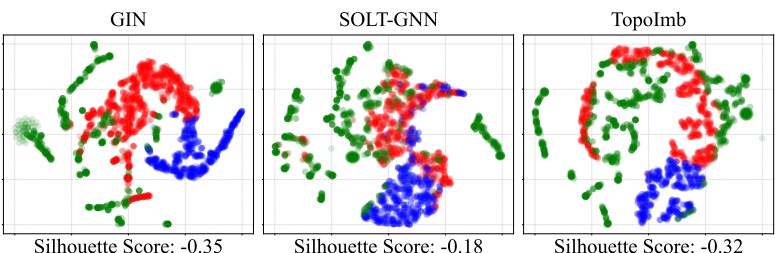

Figure D.13: Visualizations of the embedding for **global**-level **topology**-imbalanced algorithms.

## D.4 Additional Results for Efficiency Analysis (**RQ4**)

### D.4.1 Efficiency Analysis of Node-Level Class-Imbalanced Algorithms

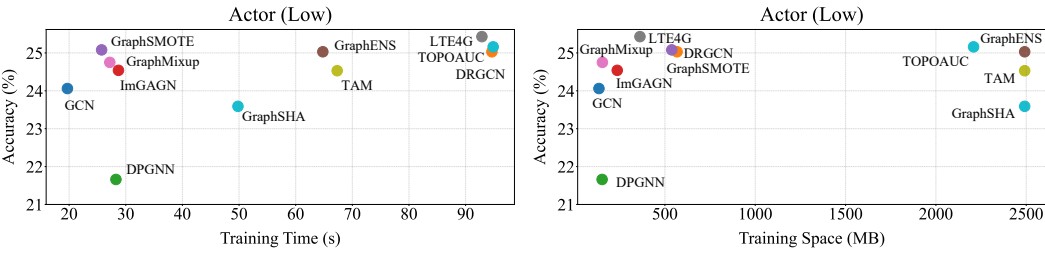

Figure D.14: Time and space analysis of **node**-level **class**-imbalanced IGL algorithms on Actor.

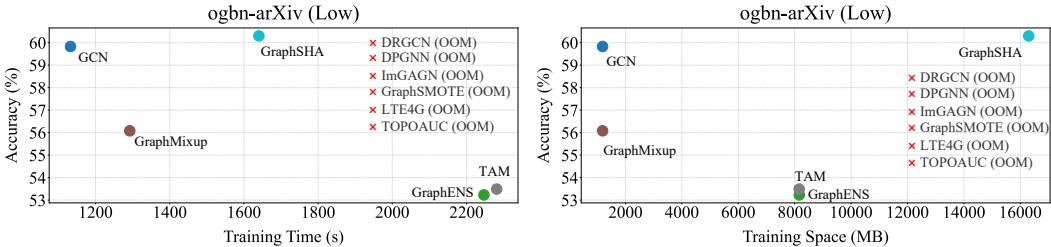

Figure D.15: Time and space analysis of **node**-level **class**-imbalanced IGL algorithms on ogbn-arXiv.

### D.4.2 Efficiency Analysis of Node-Level Local Topology-Imbalanced Algorithms

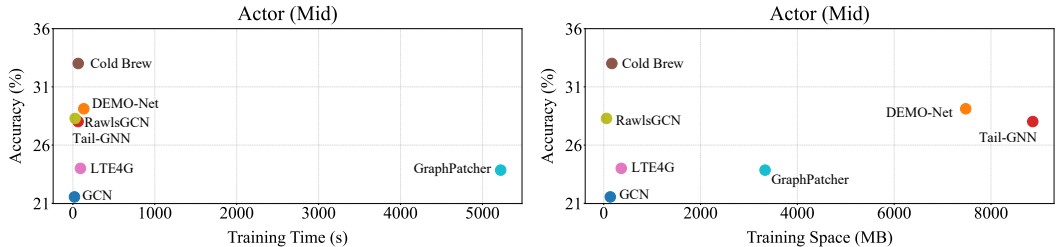

Figure D.16: Time and space analysis of **node**-level **local topology**-imbalanced IGL algorithms on Actor.

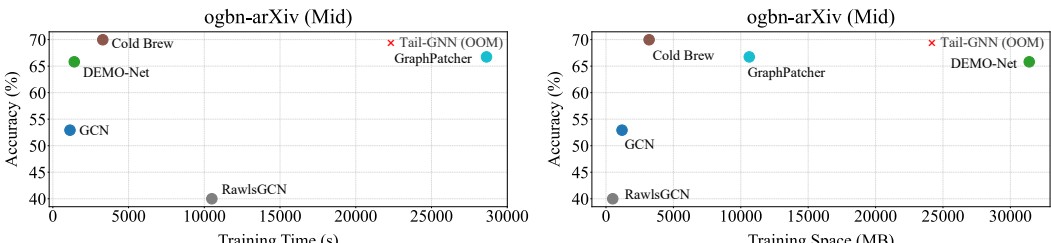

Figure D.17: Time and space analysis of **node**-level **local topology**-imbalanced IGL algorithms on ogbn-arXiv.

### D.4.3 EFFICIENCY ANALYSIS OF NODE-LEVEL GLOBAL TOPOLOGY-IMBALANCED ALGORITHMS

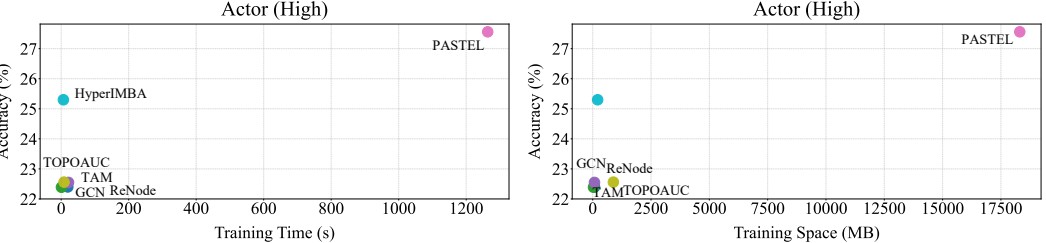

Figure D.18: Time and space analysis of **node**-level **global topology**-imbalanced IGL algorithms on Actor.

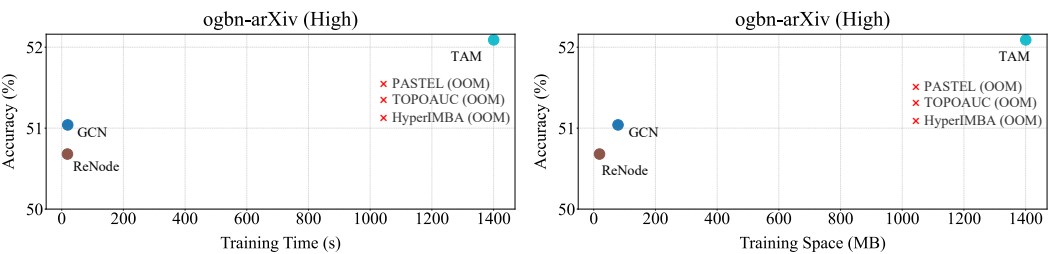

Figure D.19: Time and space analysis of **node**-level **global topology**-imbalanced IGL algorithms on ogbn-arXiv.

### D.4.4 EFFICIENCY ANALYSIS OF GRAPH-LEVEL CLASS-IMBALANCED ALGORITHMS

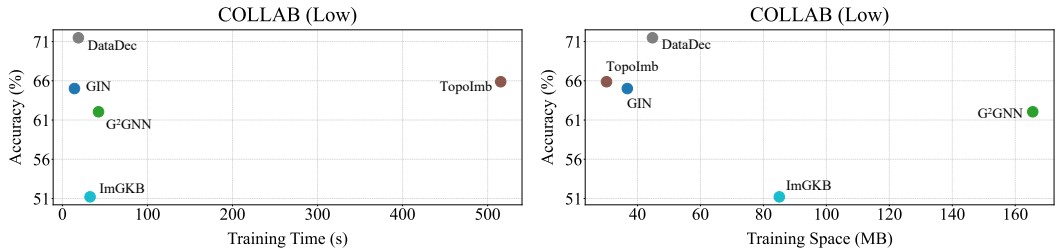

Figure D.20: Time and space analysis of **graph**-level **class**-imbalanced IGL algorithms on COLLAB.

### D.4.5 EFFICIENCY ANALYSIS OF GRAPH-LEVEL TOPOLOGY-IMBALANCED ALGORITHMS

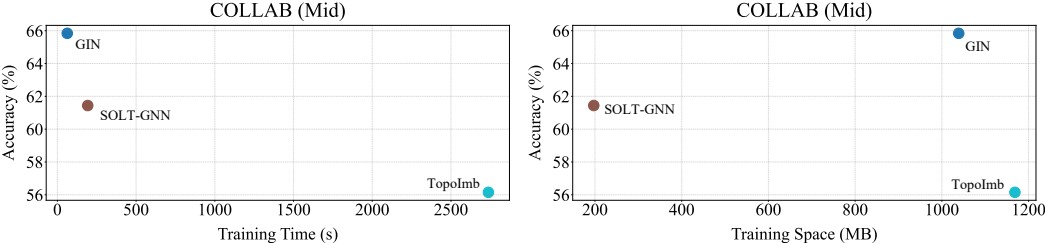

Figure D.21: Time and space analysis of **graph**-level **topology**-imbalanced IGL algorithms on COLLAB.

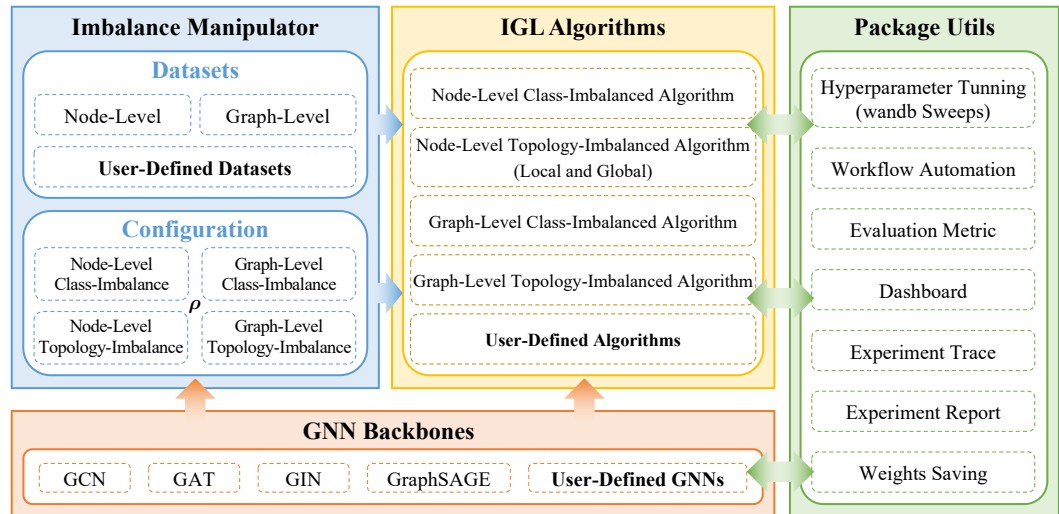

Figure E.1: The package structure of `IGL-Bench`, which mainly consists of four modules.

# E    PACKAGE AND REPRODUCIBILITY

**Package.** We established and released a comprehensive **I**mbalanced **G**raph **L**earning **Bench**mark (`IGL-Bench`) package, which serves as the first open-sourced[5] benchmark for graph-specific imbalanced learning to the best of our knowledge. `IGL-Bench` encompasses **24** state-of-the-art IGL algorithms and **17** diverse graph datasets covering node-level and graph-level tasks, addressing *class-* and *topology-imbalance* issues, while also adopting consistent data processing and splitting approaches for fair comparisons over multiple metrics with different focus.

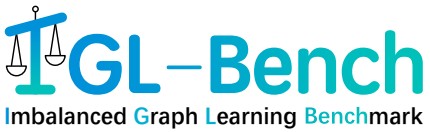

Figure E.2: The `IGL-Bench` package.

As shown in Figure E.1, the `IGL-Bench` package is mainly composed of four modules. ❶ The Imbalance Manipulator module performs different types of imbalance manipulations for the imbalance ratio on the built-in 17 node-level datasets, graph-level datasets, or user-defined datasets according to user configurations. ❷ The IGL Algorithms module has 24 state-of-the-art algorithms built-in and also supports calling user-defined IGL algorithms. ❸ The GNN Backbones module supports a variety of mainstream GNNs and also allows for user-defined GNNs. ❹ The Package Utils module offers a variety of utility tools, enhancing the usability and benchmarking efficiency of the package.

**Documentation and Uses.** We have made a concerted effort to provide users with comprehensive documentation to ensure the seamless use of the package. Additionally, we have included necessary comments to enhance code readability. We supply the required configuration files to reproduce the experimental results, which also serve as examples of how to use the package effectively.

**License.** Our package (codes and datasets) is licensed under the MIT License. This license permits users to freely use, copy, modify, merge, publish, distribute, sublicense, and sell copies of the software, provided that the original copyright notice and permission notice are included in all copies or substantial portions of the software. The MIT License is widely accepted for its simplicity and permissive terms, ensuring ease of use and contribution to the codes and datasets. We bear all responsibility in case of violation of rights, *etc*, and confirmation of the data license.

**Code Maintenance.** We are committed to continuously updating our code and actively addressing users' issues and feedback. Additionally, we warmly welcome community contributions to enhance our library and benchmark algorithms. Nonetheless, we will enforce strict version control measures to ensure reproducibility throughout the maintenance process.

---

[5]https://github.com/RingBDStack/IGL-Bench.

# F  FURTHER DISCUSSIONS

## F.1  RELATED WORKS

Benchmarking is widely used in reviews and standardized evaluations of a particular field, providing unique insights (Sun et al., 2024). To the best of our knowledge, there exists no established benchmark specifically dedicated to evaluating imbalanced learning on graphs. Our `IGL-Bench` represents the foundational effort in this domain, encompassing both node-level and graph-level challenges related to class- and topology-imbalance. This section compares and contextualizes our contributions within the broader landscape of imbalanced graph learning. We position our work in relation to notable surveys in the field.

Liu et al. (2023b) comprehensively reviews the landscape of imbalanced learning on graphs, outlining key terminologies and taxonomies related to problem types and solution strategies. It establishes a foundational understanding crucial for addressing skewed data distributions in graph-based tasks.

Focused on the challenges of GNNs in practical applications, Ju et al. (2024b) addresses imbalance in data distribution and the robustness against noise, privacy concerns, and out-of-distribution scenarios. It highlights solutions that enhance the reliability of GNNs in real-world settings.

Ma et al. (2023) specifically explores class-imbalanced learning on graphs, emphasizing the integration of graph representation learning with imbalanced learning techniques. It provides a taxonomy of existing works and outlines future directions in the evolving field of graph class-imbalanced learning.

In contrast to these surveys, our `IGL-Bench` offers a practical benchmarking package tailored explicitly for imbalanced graph learning. By systematically evaluating the performance of algorithms across various imbalance types, `IGL-Bench` provides a standardized package for assessing the efficacy and robustness of existing and future methods in this emerging field. While existing surveys establish the theoretical underpinnings and methodological approaches in imbalanced learning on graphs, `IGL-Bench` offers a concrete tool for empirical validation and comparison. This practical focus enables researchers and practitioners to not only understand the theoretical aspects but also to apply and benchmark algorithms effectively across diverse real-world graph datasets.

In summary, our work fills a critical gap by introducing `IGL-Bench` as the first benchmarking suite tailored for imbalanced graph learning, thereby advancing the state-of-the-art in the field and fostering deeper insights into the challenges and opportunities of imbalanced graph data analysis.

## F.2  LIMITATIONS

`IGL-Bench` has some limitations that we aim to address in future work.

❶ We hope to include a broader range of datasets to evaluate algorithms in different scenarios. Our current datasets are predominantly homogeneous graphs, which do not fully capture the diversity and complexity of real-world networks. Many IGL methods struggle with complex graph types, such as heterogeneous graphs with multiple types of nodes and edges. Including such datasets would provide a more robust evaluation of these algorithms and highlight their strengths and weaknesses.

❷ We hope to implement more IGL algorithms for various tasks, such as few-shot classification, dynamic graph learning, and anomaly detection, *etc*. Our current benchmark is limited to a specific set of tasks, which might not reflect the full potential and versatility of IGL methods. By expanding the range of tasks, we can gain a deeper understanding of the progress in the field and provide insights into how different algorithms perform across diverse applications.

❸ Due to resource constraints and the availability of implementations, we could not include some of the latest state-of-the-art IGL algorithms in our benchmark. This might impact the comprehensiveness of our evaluation, as some promising methods are not represented. We aim to address this by continuously updating our package and incorporating these algorithms as they become available.

❹ Our current evaluation framework primarily focuses on the performance metrics of the algorithms. However, practical aspects such as scalability, computational efficiency, and memory usage are also crucial for real-world applications. We plan to include these factors in future evaluations to provide a more holistic view of each algorithm's practicality and efficiency.

We will continuously update our repository to keep track of the latest advances in the field. We are also open to any suggestions and contributions that will improve the usability and effectiveness of our benchmark, ensuring it remains a valuable resource for the IGL research community.

### F.3 DATASET PRIVACY AND ETHICS

We ensured all datasets were sourced from publicly available repositories with explicit research permissions. For user-generated or social platform data, we rely on terms including research consent. We anonymized Personally Identifiable Information (PII) and screened for offensive content, though complete risk elimination remains challenging. Users are urged to use datasets responsibly and be mindful of ethical implications.

In terms of negative social impact, we believe our work does not pose a potentially significant negative societal impact to the best of our knowledge. Our research is primarily focused on benchmarking graph learning algorithms in the context of imbalanced data. However, we remain mindful of ethical considerations and will continue to monitor any broader implications as our work progresses.

