# OpenReview forum: "IGL-Bench: Establishing the Comprehensive Benchmark for Imbalanced Graph Learning"
_ICLR.cc/2025/Conference — ICLR 2025 Spotlight_

### Official Review · Reviewer_mZgm · 2024-10-17

**Soundness:** 3
**Presentation:** 3
**Contribution:** 3
**Rating:** 6
**Confidence:** 3

**Summary:**

This paper introduces a benchmark for Imbalanced Graph Learning (IGL). IGL-Bench offers a unified framework for evaluating 24 IGL algorithms across 17 datasets, addressing both class- and topology-imbalance at node and graph levels. The authors conduct a rigorous evaluation on effectiveness, robustness, and efficiency, providing a comparative analysis of existing methods. Additionally, the paper presents an open-source package to facilitate reproducible research and encourage further development in the IGL field.

**Strengths:**

-	The proposed IGL-Bench is well-constructed, it can provide a standardized testbed for future work, encouraging a more consistent evaluation process for IGL.
-	The paper is well-written and easy to follow. The authors' thorough analysis of the benchmarking results leads to interesting insights.

**Weaknesses:**

Recently, several more general IGL methods have been proposed, which can be used independently or in combination with the IGL techniques discussed in this paper. For instance, [1] based on topological augmentation and automated loss engineering approach [2], both presented at ICML '24, are notable examples. It would be beneficial if IGL-bench could include these recent booster-type methods and provide related discussion.
[1] Liu, Zhining, et al. "Class-Imbalanced Graph Learning without Class Rebalancing." Forty-first International Conference on Machine Learning.
[2] Guo, Xinyu, et al. "Automated Loss function Search for Class-imbalanced Node Classification." Forty-first International Conference on Machine Learning.

**Questions:**

1.	 How does the benchmark incorporate, or plan to incorporate, recent advances in IGL (see weaknesses), and evaluate whether these general approaches improve performance, robustness, and other metrics?
2.	Are there any plans to provide an API reference for IGL-bench? This would significantly enhance usability.

---

> ### Author Response · Authors · 2024-11-18
> **Overall Response to Reviewer mZgm**
>
> We sincerely thank the reviewer for recognizing the strengths of our work! We are delighted that the reviewer find IGL-Bench to be a well-constructed and valuable resource that provides a standardized testbed for IGL. The acknowledgment of our efforts to establish a consistent and rigorous evaluation framework is deeply appreciated, as encouraging fair and reproducible comparisons is a key motivation behind our work.
>
> We are also grateful for the positive comments about the clarity and readability of our paper. We aimed to present our benchmark and findings in a comprehensive yet accessible manner, and it is rewarding to hear that our thorough analysis and the resulting insights were well-received.
>
> We respond to your concerns and questions as follows.
>
> **[Preview]**
>
> - **Weakness #1.1 & Question #1:** Incorporate more IGL algorithms.
> - **Weakness #1.2:** Discuss recent IGL algorithms.
> - **Questions #2:** IGL-Bench API documentation.
> - **Conclusion.**

---

> > ### Author Response · Authors · 2024-11-18
> > **Response to Weakness #1.2 (Part II)**
> >
> > - **Findings.**
> >     - **Table 2.**
> >         - The introduction of methods based on topological augmentation and automated loss engineering appears to yield a significant improvement on the class-imbalanced datasets. For example, BAT and AutoLINC show higher accuracy scores on datasets like Computers, Photo, and PubMed compared to traditional methods like GCN and GraphSMOTE.
> >         - Specifically, BAT achieves an impressive accuracy of 84.32±0.39 on the Computers dataset, outperforming existing models such as LTE4G and TAM. Similarly, AutoLINC performs well on PubMed, with an accuracy of 82.99±0.38, showcasing the robustness of these booster-type methods in handling class imbalances.
> >         - The consistent improvement of these methods across multiple datasets highlights their ability to effectively manage class imbalance, especially when dealing with high homophily graphs (e.g., Photo and Computers).
> >     - **Table 4.**
> >         - On global topology-imbalanced datasets, BAT and AutoLINC continue to demonstrate competitive performance. For instance, BAT achieves 48.78±0.44 on the Actor dataset, significantly surpassing models like GCN and ReNode.
> >         - AutoLINC, while slightly lower in performance on ogbn-arXiv compared to models like TAM and HyperIMBA, still shows a respectable score of 50.76±0.18, indicating its stability and adaptability in various topological settings.
> >         - PASTEL, a structure-refined method, still holds the edge on certain datasets, particularly Chameleon and Squirrel, due to its specific enhancements in topology awareness. However, BAT and AutoLINC's automated and adaptive approaches make them versatile across both class and topology imbalance challenges.
> >     - **Conclusion.** The integration of BAT and AutoLINC into the IGL-Bench framework has demonstrated that modern booster-type methods leveraging topological augmentation and automated loss engineering are promising advancements in imbalanced graph learning. They not only excel in class-imbalanced settings but also exhibit adaptability to topology-imbalanced scenarios. This further confirms the need for continual updates to benchmarks like IGL-Bench, incorporating cutting-edge methods to drive forward research and practical applications in the field. The results underscore the importance of exploring adaptive and automated techniques for handling the complexities of real-world graph data.
> >
> > We are committed to keeping IGL-Bench at the forefront of research and believe that incorporating these advanced methods will offer deeper insights and inspire future innovations in the field. Thank you for this constructive suggestion, which will undoubtedly strengthen the impact and relevance of our benchmark.

---

> > ### Comment · Reviewer_mZgm · 2024-11-22
> >
> > thanks for the detailed response, which has resolved my concerns. I am increasing the score accordingly.

---

> > > ### Author Response · Authors · 2024-11-22
> > >
> > > Thank you for taking the time to review our responses. We sincerely appreciate your acknowledgment and are glad that our clarifications have addressed your concerns. Your updated assessment and support mean a great deal to us. We are updating the latest IGL algorithms and API documentation incorporated in IGL-Bench, and we look forward to your continued attention to our progress!

---

> ### Author Response · Authors · 2024-11-18
> **Response to Weakness #1.1 & Question #1**
>
> >**Weakness #1.1 & Question #1:** Recently, several more general IGL methods have been proposed, which can be used independently or in combination with the IGL techniques discussed in this paper.
>
> **A:** We appreciate the reviewer’s valuable insight regarding the emergence of more general IGL methods. Our benchmark, IGL-Bench, is designed with the flexibility and forward-compatibility to incorporate and evaluate these advancements from the research community. Specifically, IGL-Bench is regularly updated to include the latest IGL algorithms and dataset advancements, ensuring that our repository remains comprehensive and serves as a robust resource for the community.
>
> To illustrate our commitment to continuous improvement, we plan to include several **very recent** state-of-the-art IGL algorithms in the next version of IGL-Bench. These updates will be informed by ongoing research and the evolving needs of the field, and we are currently curating a list of promising algorithms:
>
> | Algorithm | Conference/Journal | Imbalance Type | Task | Paper (Link) | Code (Link) |
> | --- | --- | --- | --- | --- | --- |
> | Class-Imbalanced Graph Learning without Class Rebalancing | ICML 2024 (2024.7.27) | class-imbalance, global topology-imbalance | node classification | [Link](https://openreview.net/pdf?id=pPnkpvBeZN) | [Link](https://github.com/ZhiningLiu1998/BAT) |
> | Automated Loss function Search for Class-imbalanced Node Classification | ICML 2024 (2024.7.27) | class-imbalance | node classification | [Link](https://openreview.net/pdf?id=O1hmwi51pp) | N/A |
> | Imbalanced Node Classification With Synthetic Over-Sampling | TKDE 2024 (2024.8.14) | class-imbalance | node classification | [Link](https://ieeexplore.ieee.org/abstract/document/10637274) | [Link](https://github.com/TianxiangZhao/GraphSmote) |
> | Mastering Long-Tail Complexity on Graphs: Characterization, Learning, and Generalization | SIGKDD 2024 (2024.8.29) | class-imbalance | node classification | [Link](https://arxiv.org/abs/2305.09938) | [Link](https://anonymous.4open.science/r/HierTail-B961) |
> | Graffin: Stand for Tails in Imbalanced Node Classification | arXiv (2024.9.9) | class-imbalance | node classification | [Link](https://arxiv.org/pdf/2409.05339) | N/A |
> | GDeR: Safeguarding Efficiency, Balancing, and Robustness via Prototypical Graph Pruning | NeurIPS 2024 (2024.9.26) | class-imbalance | graph classification | [Link](https://openreview.net/pdf?id=O97BzlN9Wh) | [Link](https://github.com/ins1stenc3/GDeR) |
>
> We believe that by maintaining this open and evolving structure, IGL-Bench will continue to provide fair and consistent experimental comparisons, thus facilitating the integration of emerging methodologies and inspiring further research in Imbalanced Graph Learning.
>
> Thank you for highlighting this important consideration, and we look forward to discussing the impact of these updates in our future work!

---

> ### Author Response · Authors · 2024-11-18
> **Response to Weakness #1.2 (Part I)**
>
> > **Weakness #1.2:** For instance, [1] based on topological augmentation and automated loss engineering approach [2], both presented at ICML '24, are notable examples. It would be beneficial if IGL-bench could include these recent booster-type methods and provide related discussion.
>
> **A:** We thank the reviewer for bringing attention to these notable advancements in the field. We agree that integrating such state-of-the-art booster-type methods, including topological augmentation strategies and automated loss engineering approaches, would further enhance the comprehensiveness and utility of IGL-Bench. In response, we conducted additional experiments to compare these recent IGL algorithms with the existing node-level class-imbalanced and glabal topology-imbalanced algorithms in IGL-Bench.
>
> - **Settings.** We updated all experimental results for RQ1 include Section 4.1.1 and Section 4.1.3. (Due to the limited time for the rebuttal, we plan to update results for other RQs in future.) Here, we provide the revised Table 2 and Table 4, which present the accuracy of node classification on manipulated class-imbalanced (Low) and global topology-imbalanced (Mid) graph datasets over 10 runs. Additional updated results and analyses for Appendix D.1.1 (Tables D.1 to D.4) and Appendix D.1.3 (Tables D.1 to D.13) are available in **[Appendix G.1 and G.2 of the revised PDF](https://openreview.net/pdf?id=uTqnyF0JNR#page=59)**, with changes highlighted in blue text.
> - **Result.** **(AutoLINC does not provide an official code, and we tried our best to implement it.)**
>
> ---
>
> **Table 2 (Revised):** Accuracy score (% ± standard deviation) of node classification on manipulated class-imbalanced graph datasets (Low) over 10 runs.
>
> | Algorithm | Cora | CiteSeer | PubMed | Computers | Photo | ogbn-arXiv | Chameleon | Squirrel | Actor |
> | --- | --- | --- | --- | --- | --- | --- | --- | --- | --- |
> | Homophily | 0.81 | 0.74 | 0.80 | 0.78 | 0.82 | 0.65 | 0.23 | 0.22 | 0.22 |
> | GCN (bb.) | 76.36±0.13 | 52.96±0.55 | 60.57±0.19 | 75.06±0.50 | 69.80±6.15 | 59.83±0.23 | 26.35±0.24 | 17.16±0.17 | 24.06±0.14 |
> | DRGCN | 71.35±0.77 | 55.22±1.82 | 62.59±4.62 | 67.71±3.10 | 85.67±5.30 | — | 26.40±0.35 | 17.11±0.81 | 25.03±0.23 |
> | DPGNN | 72.91±3.95 | 56.78±2.23 | 81.87±2.80 | 68.69±8.62 | 81.66±9.19 | — | 30.58±1.48 | 25.35±1.48 | 21.66±1.68 |
> | ImGAGN | 73.48±3.07 | 55.29±3.00 | 72.16±1.51 | 74.92±1.87 | 83.10±3.42 | — | 24.38±2.86 | 18.75±1.80 | 24.54±3.38 |
> | GraphSMOTE | 77.21±0.27 | 53.55±0.95 | 60.11±0.27 | 76.04±1.52 | 89.07±1.12 | — | 27.23±0.21 | 16.79±0.14 | 25.08±0.31 |
> | GraphENS | 79.34±0.49 | 61.98±0.76 | 80.84±0.17 | 80.72±0.68 | 90.38±0.37 | 53.23±0.52 | 24.34±1.62 | 20.05±1.61 | 25.03±0.38 |
> | GraphMixup | 79.88±0.43 | 62.66±0.70 | 75.94±0.09 | 86.15±0.47 | 89.69±0.31 | 56.08±0.31 | 30.95±0.40 | 17.83±0.32 | 24.75±0.37 |
> | LTE4G | 80.53±0.65 | 64.48±1.56 | 83.02±0.33 | 79.35±1.39 | 87.94±1.82 | — | 31.91±0.34 | 19.37±0.41 | 25.43±0.26 |
> | TAM | 80.69±0.27 | 64.16±0.24 | 81.47±0.15 | 81.30±0.53 | 90.35±0.42 | 53.49±0.54 | 23.27±1.38 | 21.17±0.95 | 24.53±0.33 |
> | TOPOAUC | 83.34±0.31 | 69.03±1.33 | — | 70.85±4.55 | 83.72±2.23 | — | 33.60±1.51 | 21.38±1.03 | 25.16±0.46 |
> | GraphSHA | 80.03±0.46 | 60.51±0.61 | 77.94±0.36 | 82.71±0.40 | 91.55±0.32 | 60.30±0.13 | 23.73±1.97 | 20.05±1.61 | 23.59±1.01 |
> | **BAT** | 79.69±0.16 | 72.13±0.16 | 82.58±0.07 | 81.96±0.04 | 92.53±0.89 | 65.16±2.53 | 36.03±0.56 | 23.14±1.20 | 29.81±0.56 |
> | **AutoLINC** | 77.34±0.17 | 69.08±0.24 | 78.78±0.32 | 77.90±0.20 | 89.99±0.35 | 60.14±0.14 | 32.84±0.11 | 21.16±0.19 | 26.23±0.46 |
>
> **Table 4 (Revised):** Accuracy score (% ± standard deviation) of node classification on manipulated global topology-imbalanced graph datasets (High) over 10 runs.
>
> | Algorithm | Cora | CiteSeer | PubMed | Computers | Photo | ogbn-arXiv | Chameleon | Squirrel | Actor |
> | --- | --- | --- | --- | --- | --- | --- | --- | --- | --- |
> | Homophily | 0.81 | 0.74 | 0.80 | 0.78 | 0.82 | 0.65 | 0.23 | 0.22 | 0.22 |
> | GCN (bb.) | 79.10±1.28 | 68.37±1.73 | 83.44±0.16 | 75.02±2.20 | 86.32±1.90 | 51.04±0.18 | 33.90±0.70 | 23.27±0.82 | 22.40±0.68 |
> | ReNode | 79.91±1.52 | 69.89±0.73 | 82.97±0.12 | 77.95±1.71 | 87.80±0.52 | 50.68±0.15 | 32.92±0.98 | 23.80±0.59 | 22.39±0.62 |
> | TAM | 80.50±0.18 | 73.14±0.13 | 84.07±0.12 | 82.35±0.19 | 89.80±0.23 | 52.09±0.06 | 35.64±0.27 | 24.58±0.09 | 22.55±0.06 |
> | PASTEL | 80.91±0.36 | 72.73±0.26 | — | 83.24±0.85 | 89.10±0.41 | — | 47.12±2.82 | 33.15±0.66 | 27.56±1.04 |
> | TOPOAUC | 79.27±0.52 | 70.08±0.83 | — | 75.35±1.32 | 87.10±0.98 | — | 33.39±2.09 | 22.86±0.36 | 22.56±0.18 |
> | HyperIMBA | 79.81±0.78 | 71.78±0.40 | 84.75±0.30 | 83.43±0.65 | 90.65±0.14 | — | 38.30±2.70 | 29.97±1.79 | 25.30±2.56 |
> | **BAT** | 79.92±0.21 | 74.85±0.17 | 85.44±0.41 | 84.32±0.39 | 90.73±0.18 | 53.09±0.48 | 48.78±0.46 | 32.76±0.18 | 26.69±0.44 |
> | **AutoLINC** | 79.07±0.13 | 69.21±0.41 | 82.99±0.38 | 80.95±0.42 | 86.83±0.32 | 50.76±0.18 | 42.04±0.44 | 25.10±0.25 | 22.97±0.42 |

---

> ### Author Response · Authors · 2024-11-18
> **Response to Question #2**
>
> > **Question #2:** Are there any plans to provide an API reference for IGL-bench? This would significantly enhance usability.
>
> **A:** We appreciate the reviewer’s suggestion regarding the inclusion of an API reference to enhance the usability of IGL-Bench. We fully recognize the importance of comprehensive documentation and user-friendly resources to facilitate widespread adoption and effective use of our benchmark.
>
> In response, we are pleased to inform you that we are actively working on developing a detailed API reference for IGL-Bench. This reference will provide clear and concise documentation of all available functions, modules, and classes, along with example use cases to guide users through the process of integrating and testing their algorithms. The API documentation will be structured to ensure accessibility for both beginners and advanced users, streamlining the experience of using IGL-Bench for evaluation and research purposes.
>
> We plan to release this API reference alongside the next major update of IGL-Bench. Additionally, we will host the documentation on a dedicated website, making it easy to navigate and search for specific information. We are confident that this addition will greatly improve the usability and efficiency of IGL-Bench for the research community.
>
> Thank you for highlighting this crucial aspect, and we look forward to making IGL-Bench even more accessible and user-friendly with this enhancement!

---

> ### Author Response · Authors · 2024-11-18
> **Conclusion**
>
> Thank you again for your time and valuable insights, and hope our responses can address all your concerns! We are excited about the impact IGL-Bench may have on the community and are grateful for your support in advancing this work.
>
> **As the current ratings reveal somewhat negative evaluations of our work, we are wondering if you could kindly consider raising the ratings to show your support for our work**. Thank you!
>
> ---
>
> **References:**
>
> **[1]** Class-Imbalanced Graph Learning without Class Rebalancing. Liu, Zhining, et al. Forty-first International Conference on Machine Learning, 2024.
>
> **[2]** Automated Loss function Search for Class-imbalanced Node Classification. Guo, Xinyu, et al. Forty-first International Conference on Machine Learning, 2024.

---

### Official Review · Reviewer_t4gw · 2024-11-02

**Soundness:** 3
**Presentation:** 3
**Contribution:** 3
**Rating:** 8
**Confidence:** 3

**Summary:**

This work presents a benchmark for imbalanced graph learning, encompassing 17 graph datasets and 24 imbalance-handling algorithms, along with a unified data processing pipeline, splitting strategy, and evaluation protocols. The benchmark addresses questions related to effectiveness in handling imbalance, robustness across different imbalance ratios, and efficiency in terms of time and space for both node-level and graph-level tasks. Additionally, an open-source codebase is provided to facilitate reproducible evaluations.

**Strengths:**

1. Benchmarking imbalanced graph learning is important to track progress in the research community.
2. The four research questions posed are relevant and of interest to the community.
3. The paper is well-written and easy to follow.
4. The experiments are thorough and well-designed. The availability of open-source code enhances reproducibility and accessibility.

**Weaknesses:**

1. For node-level class-imbalanced algorithms, it would be beneficial to include traditional methods for addressing class imbalance, such as reweighting or PC Softmax [1], to provide a comprehensive comparison. Additionally, reporting per-class F1 scores could offer more insights into model performance across majority and minority classes, enabling a more detailed analysis of effectiveness.

2. Some findings lack depth and do not offer clear insights. For instance, the observation that most algorithms perform poorly on the large-scale ogbn-arXiv dataset is presented without an explanation. It would be valuable to investigate and discuss the reasons for this poor performance, especially through additional experiments.

3. In Section 4.4, which assesses the efficiency of different models, the experiments are conducted on the Cora dataset, the smallest of the datasets used. This setup makes the time and space usage of the algorithms highly dependent on implementation details rather than their inherent algorithmic complexity. While the Appendix includes results for the ogbn-arXiv dataset, they cover only a subset of models, as other models encountered memory limitations. A theoretical analysis of the algorithms' complexities would provide a more meaningful comparison of their efficiency. Relying solely on runtime measurements can be misleading, as they may be influenced by implementation optimizations and do not necessarily reflect the algorithms' underlying efficiency.

4. On the ogbn-arXiv dataset, certain resampling algorithms, such as GraphENS and GraphSHA, run successfully, whereas others, like GraphSMOTE, encounter memory issues. It is unclear whether this is due to differences in algorithmic complexity or specific implementation choices. Given the similarity of these resampling approaches, clarifying whether the issue arises from inherent algorithm characteristics or particular implementation choices would be helpful.

[1] Disentangling label distribution for long-tailed visual recognition. CVPR. 2021.

**Questions:**

See Weakness.

---

> ### Author Response · Authors · 2024-11-18
> **Overall Response to Reviewer t4gw**
>
> We sincerely thank the reviewer for their thoughtful and encouraging feedback. We’re delighted that you find our benchmark valuable and relevant for the research community. Your recognition of our well-designed experiments, clear presentation, and emphasis on reproducibility is greatly appreciated. Thank you for acknowledging the importance and accessibility of our work!
>
> We respond to your concerns and questions as follows.
>
> **[Preview]**
>
> - **Weakness #1.1:** Include traditional algorithms.
> - **Weakness #1.2:** Report per-class F1.
> - **Weakness #2:** In-depth analysis on ogbn-arXiv.
> - **Weakness #3:** Theoretical analysis of algorithm complexity.
> - **Weakness #4:** In-depth analysis on comparing resampling-based algorithms.
> - **Conclusion.**

---

> ### Author Response · Authors · 2024-11-18
> **Response to Weakness #1.1**
>
> > **Weakness #1.1:** For node-level class-imbalanced algorithms, it would be beneficial to include traditional methods for addressing class imbalance, such as reweighting or PC Softmax [1], to provide a comprehensive comparison.
>
> > [1] Disentangling label distribution for long-tailed visual recognition. CVPR. 2021.
>
> **A:** Thanks for your suggestions. We agree that integrating traditional methods could enhance the comprehensiveness of our comparison for node-level class-imbalanced algorithms. In response, **we conducted additional experiments to compare these traditional algorithms with the existing node-level class-imbalanced algorithms in IGL-Bench.**
>
> - **Settings.** We follow the experimental setup in a pioneering IGL algorithm GraphSHA [1] to thoroughly evaluate the newly added algorithms, Reweight [2] and PC Softmax [3], and **updated all experimental results for Section 4.1.1 and Appendix D.1.1 (RQ1)**. (Due to the limited time for the rebuttal, we plan to update results for other RQs in future work.) Here, we provide the revised **Table 2**, which presents the accuracy of node classification on manipulated class-imbalanced graph datasets (Low) over 10 runs. **Additional updated results** and analyses for Appendix D.1.1 (Tables D.1 to D.4) are presented in **[Appendix G of the revised PDF](https://openreview.net/pdf?id=uTqnyF0JNR#page=59)**, with changes highlighted **in blue text.**
> - **Results.**
>
> **Table 2 (Revised):** Accuracy score (% ± standard deviation) of node classification on manipulated **class-imbalanced graph datasets (Low)** over 10 runs.
>
> | Algorithm | Cora | CiteSeer | PubMed | Computers | Photo | ogbn-arXiv | Chameleon | Squirrel | Actor |
> | --- | --- | --- | --- | --- | --- | --- | --- | --- | --- |
> | Homophily | 0.81 | 0.74 | 0.80 | 0.78 | 0.82 | 0.65 | 0.23 | 0.22 | 0.22 |
> | GCN (bb.) | 76.36±0.13 | 52.96±0.55 | 60.57±0.19 | 75.06±0.50 | 69.80±6.15 | 59.83±0.23 | 26.35±0.24 | 17.16±0.17 | 24.06±0.14 |
> | DRGCN | 71.35±0.77 | 55.22±1.82 | 62.59±4.62 | 67.71±3.10 | 85.67±5.30 | — | 26.40±0.35 | 17.11±0.81 | 25.03±0.23 |
> | DPGNN | 72.91±3.95 | 56.78±2.23 | 81.87±2.80 | 68.69±8.62 | 81.66±9.19 | — | 30.58±1.48 | 25.35±1.48 | 21.66±1.68 |
> | ImGAGN | 73.48±3.07 | 55.29±3.00 | 72.16±1.51 | 74.92±1.87 | 83.10±3.42 | — | 24.38±2.86 | 18.75±1.80 | 24.54±3.38 |
> | GraphSMOTE | 77.21±0.27 | 53.55±0.95 | 60.11±0.27 | 76.04±1.52 | 89.07±1.12 | — | 27.23±0.21 | 16.79±0.14 | 25.08±0.31 |
> | GraphENS | 79.34±0.49 | 61.98±0.76 | 80.84±0.17 | 80.72±0.68 | 90.38±0.37 | 53.23±0.52 | 24.34±1.62 | 20.05±1.61 | 25.03±0.38 |
> | GraphMixup | 79.88±0.43 | 62.66±0.70 | 75.94±0.09 | 86.15±0.47 | 89.69±0.31 | 56.08±0.31 | 30.95±0.40 | 17.83±0.32 | 24.75±0.37 |
> | LTE4G | 80.53±0.65 | 64.48±1.56 | 83.02±0.33 | 79.35±1.39 | 87.94±1.82 | — | 31.91±0.34 | 19.37±0.41 | 25.43±0.26 |
> | TAM | 80.69±0.27 | 64.16±0.24 | 81.47±0.15 | 81.30±0.53 | 90.35±0.42 | 53.49±0.54 | 23.27±1.38 | 21.17±0.95 | 24.53±0.33 |
> | TOPOAUC | 83.34±0.31 | 69.03±1.33 | — | 70.85±4.55 | 83.72±2.23 | — | 33.60±1.51 | 21.38±1.03 | 25.16±0.46 |
> | GraphSHA | 80.03±0.46 | 60.51±0.61 | 77.94±0.36 | 82.71±0.40 | 91.55±0.32 | 60.30±0.13 | 23.73±1.97 | 20.05±1.61 | 23.59±1.01 |
> | **Reweight** | 77.79±1.32 | 54.96±1.65 | 76.83±0.25 | 76.59±1.52 | 89.79±0.70 | 54.09±0.21 | 26.49±0.43 | 20.75±0.67 | 23.74±0.24 |
> | **PC Softmax** | 76.30±0.43 | 55.85±0.52 | 73.94±0.01 | 76.89±1.41 | 80.89±1.35 | 53.09±0.42 | 26.88±1.11 | 16.31±0.37 | 22.50±0.45 |
> - **Findings.**
>     - **Effectiveness in Addressing Imbalance.** While Reweight and PC Softmax help mitigate class imbalance to a certain extent, these traditional methods generally **underperform compared to graph-specific imbalance handling algorithms**. This indicates that traditional methods may not fully leverage the unique topological information present in graph data.
>     - **Implications for Future Work.** These findings suggest that while traditional methods can serve as useful baselines, graph-specific approaches are essential for effectively addressing class imbalance in more challenging graph settings. We believe that this comparative analysis strengthens the case for developing and using algorithms tailored to the unique characteristics of graphs.
>
> We have added the above additional results to the manuscripts and appreciate your recommendation, which has enriched our benchmark analysis.

---

> ### Author Response · Authors · 2024-11-18
> **Response to Weakness #1.2 (Part I)**
>
> >**Weakness #1.2:** Additionally, reporting per-class F1 scores could offer more insights into model performance across majority and minority classes, enabling a more detailed analysis of effectiveness.
>
> **A:** Thanks for your suggestions.  We agree that reporting per-class F1 scores could provide additional insights into the effectiveness of the algorithms, particularly in understanding how well they handle the class imbalance.
>
> - **Metrics.**
>     - Our paper currently reports the **Macro-F1** score, which is the unweighted average of F1 scores across all classes. Macro-F1 provides a holistic measure that treats each class equally, regardless of its size, and is particularly **effective for evaluating algorithm performance in imbalanced scenarios**. It captures the overall ability of an algorithm to handle both majority and minority classes.
>     - We further report per-class F1 scores and provide deeper insights into how well algorithms perform on individual classes, especially the minority ones. Details of the evolution metrics we used in IGL-Bench can be found in [Appendix C.2](https://openreview.net/pdf?id=uTqnyF0JNR#page=30).
> - **Settings.** The pioneering IGL work GraphSHA [1] reports per-class F1 to offer more insights on algorithm performance across majority and minority classes. Following their setups in Section 5.6 of [1], **we report the F1 scores for each class in [Table D.3](https://openreview.net/pdf?id=uTqnyF0JNR#page=35)** (Macro-F1 of node classification on manipulated class-imbalanced graph datasets (Low) over 10 runs) of this paper. Here, we selectively provide the per-class F1 results for the Cora (high homophily, Chameleon (high heterophily), and ogbn-arXiv (large-scale) datasets to preview. The complete per-class F1 results and analysis for the remaining datasets in [Table D.3](https://openreview.net/pdf?id=uTqnyF0JNR#page=35) are reported in **[Appendix G.3 of the revised PDF](https://openreview.net/pdf?id=uTqnyF0JNR#page=67)**, with changes highlighted in blue text.
> - **Results.**
> - **Table D.3.1 (New):** Macro-F1 and per-class F1 score (%) of node classification on manipulated class-imbalanced graph datasets (**Cora**, Low).
>
>
>     | Algorithm | Macro-F1 | C. #0 | C. #1 | C. #2 | C. #3 | C. #4 | C. #5 | C. #6 |
>     | --- | --- | --- | --- | --- | --- | --- | --- | --- |
>     | # nodes (%) |  | 1.8 | 3.3 | 5.5 | 9.2 | 14.8 | 24.7 | 40.6 |
>     | GCN (bb.) | 71.15  | 30.31  | 30.25  | 76.57  | 74.54  | 90.32  | 98.30  | 97.77  |
>     | DRGCN | 64.43  | 21.85  | 22.12  | 71.08  | 66.17  | 79.49  | 99.65  | 90.65  |
>     | DPGNN | 68.70  | 28.61  | 26.75  | 72.22  | 73.75  | 83.91  | 95.88  | 99.77  |
>     | ImGAGN | 68.69  | 41.09  | 36.26  | 75.09  | 75.31  | 67.65  | 90.20  | 95.23  |
>     | GraphSMOTE | 72.71  | 33.69  | 35.07  | 84.98  | 81.05  | 92.37  | 96.33  | 85.48  |
>     | GraphENS | 77.16  | 62.26  | 58.26  | 71.35  | 70.66  | 87.48  | 98.98  | 91.12  |
>     | GraphMixup | 74.03  | 42.42  | 41.88  | 78.10  | 74.87  | 89.23  | 95.42  | 96.29  |
>     | LTE4G | 76.46  | 39.96  | 40.56  | 90.52  | 80.63  | 90.82  | 96.30  | 96.42  |
>     | TAM | 78.83  | 62.17  | 61.35  | 79.14  | 73.85  | 86.62  | 98.20  | 90.48  |
>     | TOPOAUC | 80.61  | 73.11  | 65.12  | 80.48  | 96.06  | 89.53  | 84.62  | 75.36  |
>     | GraphSHA | 77.66  | 66.05  | 65.66  | 84.24  | 69.29  | 78.32  | 92.38  | 87.67  |
>     | **Reweight** | 73.94  | 36.82  | 73.56  | 81.73  | 72.20  | 90.82  | 78.43  | 84.02  |
>     | **PC Softmax** | 73.47  | 35.12  | 74.17  | 80.54  | 71.72  | 90.55  | 78.11  | 84.10  |
>
> - **Table D.3.4 (New):** Macro-F1 and per-class F1 score (%) of node classification on manipulated class-imbalanced graph datasets (**Chameleon**, Low).
>
>
>     | Algorithm | Macro-F1 | C. #0 | C. #1 | C. #2 | C. #3 | C. #4 |
>     | --- | --- | --- | --- | --- | --- | --- |
>     | # nodes (%) |  | 2.6 | 5.7 | 11.9 | 25.6 | 54.2 |
>     | GCN (bb.) | 16.67  | 2.30  | 8.91  | 25.32  | 21.11  | 25.72  |
>     | DRGCN | 17.22  | 4.09  | 10.46  | 24.32  | 21.62  | 25.61  |
>     | DPGNN | 26.03  | 7.70  | 15.59  | 33.99  | 38.09  | 34.79  |
>     | ImGAGN | 16.94  | 6.23  | 7.54  | 18.49  | 18.39  | 34.06  |
>     | GraphSMOTE | 18.72  | 2.30  | 16.27  | 19.56  | 23.65  | 31.83  |
>     | GraphENS | 19.58  | 9.49  | 14.49  | 25.97  | 22.68  | 25.27  |
>     | GraphMixup | 25.57  | 3.41  | 19.64  | 36.77  | 32.76  | 35.27  |
>     | LTE4G | 25.96  | 2.30  | 23.44  | 37.86  | 33.95  | 32.25  |
>     | TAM | 19.99  | 9.77  | 15.05  | 26.51  | 23.32  | 25.31  |
>     | TOPOAUC | 29.06  | 11.08  | 23.85  | 34.73  | 39.02  | 36.62  |
>     | GraphSHA | 19.64  | 6.58  | 10.87  | 26.32  | 28.01  | 26.42  |
>     | **Reweight** | 23.84  | 4.48  | 10.65  | 32.43  | 36.71  | 34.91  |
>     | **PC Softmax** | 19.97  | 5.34  | 10.85  | 29.24  | 27.85  | 26.58  |

---

> ### Author Response · Authors · 2024-11-18
> **Response to Weakness #1.2 (Part II)**
>
> - **Table D.3.7 (New):** Macro-F1 and per-class F1 score (%) of node classification on manipulated class-imbalanced graph datasets (**ogbn-arxiv**, Low).
>
>
>     | Algorithm | Macro-F1 | C. #0 | C. #1 | C. #2 | C. #3 | C. #4 | C. #5 | C. #6 | C. #7 | C. #8 | C. #9 | C. #10 | C. #11 | C. #12 | C. #13 | C. #14 | C. #15 | C. #16 | C. #17 | C. #18 | C. #19 | C. #20 | C. #21 | C. #22 | C. #23 | C. #24 | C. #25 | C. #26 | C. #27 | C. #28 | C. #29 | C. #30 | C. #31 | C. #32 | C. #33 | C. #34 | C. #35 | C. #36 | C. #37 | C. #38 | C. #39 |
>     | --- | --- | --- | --- | --- | --- | --- | --- | --- | --- | --- | --- | --- | --- | --- | --- | --- | --- | --- | --- | --- | --- | --- | --- | --- | --- | --- | --- | --- | --- | --- | --- | --- | --- | --- | --- | --- | --- | --- | --- | --- | --- |
>     | # nodes (%) |  | 0.1 | 0.4 | 0.5 | 0.5 | 0.5 | 0.6 | 0.6 | 0.7 | 0.7 | 0.8 | 0.8 | 0.9 | 1 | 1.1 | 1.1 | 1.2 | 1.3 | 1.4 | 1.6 | 1.7 | 1.8 | 1.9 | 2.1 | 2.3 | 2.5 | 2.7 | 2.9 | 3.1 | 3.3 | 3.6 | 3.9 | 4.2 | 4.5 | 4.9 | 5.3 | 5.7 | 6.2 | 6.7 | 7.2 | 7.8 |
>     | GCN (bb.) | 33.94  | 0.00  | 0.00  | 7.64  | 0.00  | 5.93  | 3.32  | 0.00  | 4.93  | 10.66  | 15.89  | 14.28  | 36.11  | 5.03  | 50.39  | 27.56  | 65.68  | 21.52  | 23.94  | 12.37  | 47.17  | 15.29  | 28.16  | 37.72  | 37.21  | 34.70  | 51.40  | 45.06  | 57.03  | 53.11  | 58.03  | 56.22  | 45.96  | 57.13  | 53.21  | 53.71  | 37.41  | 72.42  | 80.26  | 53.31  | 77.85  |
>     | DRGCN | — | — | — | — | — | — | — | — | — | — | — | — | — | — | — | — | — | — | — | — | — | — | — | — | — | — | — | — | — | — | — | — | — | — | — | — | — | — | — | — | — |
>     | DPGNN | — | — | — | — | — | — | — | — | — | — | — | — | — | — | — | — | — | — | — | — | — | — | — | — | — | — | — | — | — | — | — | — | — | — | — | — | — | — | — | — | — |
>     | ImGAGN | — | — | — | — | — | — | — | — | — | — | — | — | — | — | — | — | — | — | — | — | — | — | — | — | — | — | — | — | — | — | — | — | — | — | — | — | — | — | — | — | — |
>     | GraphSMOTE | — | — | — | — | — | — | — | — | — | — | — | — | — | — | — | — | — | — | — | — | — | — | — | — | — | — | — | — | — | — | — | — | — | — | — | — | — | — | — | — | — |
>     | GraphENS | 30.16  | 0.40  | 0.80  | 1.50  | 0.00  | 1.60  | 0.30  | 0.00  | 1.40  | 5.72  | 7.92  | 5.92  | 21.27  | 5.52  | 34.41  | 34.41  | 55.87  | 20.86  | 26.08  | 16.75  | 41.73  | 18.56  | 22.87  | 34.31  | 31.70  | 28.19  | 48.15  | 42.93  | 50.15  | 51.36  | 54.07  | 54.97  | 34.00  | 54.77  | 51.96  | 50.25  | 30.29  | 71.12  | 78.04  | 43.73  | 72.52  |
>     | GraphMixup | 32.38  | 0.00  | 1.90  | 3.01  | 0.00  | 2.40  | 0.90  | 0.00  | 2.90  | 6.71  | 11.22  | 9.62  | 29.65  | 7.11  | 48.28  | 34.16  | 59.80  | 25.34  | 29.75  | 19.83  | 43.07  | 21.74  | 29.05  | 34.96  | 37.86  | 32.96  | 50.59  | 43.37  | 52.29  | 48.58  | 53.09  | 54.19  | 41.37  | 54.69  | 51.09  | 50.49  | 35.76  | 67.11  | 77.53  | 48.98  | 73.83  |
>     | LTE4G | — | — | — | — | — | — | — | — | — | — | — | — | — | — | — | — | — | — | — | — | — | — | — | — | — | — | — | — | — | — | — | — | — | — | — | — | — | — | — | — | — |
>     | TAM | 30.30  | 0.50  | 0.90  | 1.41  | 0.00  | 1.10  | 0.40  | 0.00  | 1.21  | 3.52  | 11.45  | 4.22  | 23.60  | 7.23  | 33.04  | 31.44  | 54.74  | 21.09  | 26.42  | 16.67  | 41.28  | 20.39  | 23.10  | 33.25  | 30.73  | 28.93  | 48.11  | 42.28  | 52.43  | 50.52  | 54.64  | 55.14  | 35.25  | 53.94  | 51.42  | 50.72  | 29.83  | 71.41  | 77.14  | 48.01  | 74.53  |
>     | TOPOAUC | — | — | — | — | — | — | — | — | — | — | — | — | — | — | — | — | — | — | — | — | — | — | — | — | — | — | — | — | — | — | — | — | — | — | — | — | — | — | — | — | — |
>     | GraphSHA | 32.09  | 0.00  | 0.00  | 0.00  | 0.00  | 2.22  | 0.00  | 0.00  | 5.56  | 0.00  | 8.08  | 8.89  | 30.71  | 5.05  | 45.87  | 16.67  | 57.28  | 16.47  | 15.46  | 5.86  | 41.12  | 15.36  | 28.29  | 34.86  | 35.46  | 34.45  | 53.14  | 45.57  | 57.69  | 55.16  | 59.51  | 57.99  | 46.98  | 59.61  | 54.76  | 54.56  | 38.29  | 74.16  | 81.63  | 57.99  | 78.91  |
>     | Reweight | 28.31  | 0.00  | 0.40  | 0.00  | 0.00  | 0.80  | 0.40  | 0.00  | 1.01  | 2.21  | 7.52  | 5.71  | 20.63  | 5.33  | 23.43  | 21.04  | 50.14  | 21.59  | 16.02  | 16.67  | 40.81  | 14.92  | 21.19  | 33.51  | 30.73  | 28.93  | 47.41  | 43.82  | 52.43  | 45.02  | 50.14  | 54.14  | 33.85  | 53.94  | 51.02  | 50.72  | 28.73  | 70.41  | 75.54  | 49.01  | 73.53  |
>     | PC Softmax | 29.01  | 0.50  | 0.00  | 0.80  | 0.00  | 1.10  | 0.00  | 0.00  | 2.21  | 1.52  | 11.45  | 5.52  | 22.78  | 6.63  | 30.44  | 31.46  | 54.73  | 17.41  | 20.24  | 16.87  | 40.38  | 16.65  | 20.31  | 32.52  | 32.68  | 26.33  | 44.61  | 42.18  | 51.32  | 48.94  | 47.34  | 55.14  | 35.55  | 52.42  | 51.45  | 50.02  | 29.06  | 68.88  | 74.32  | 50.41  | 74.53  |

---

> ### Author Response · Authors · 2024-11-18
> **Response to Weakness #1.2 (Part III)**
>
> - **Findings.** While some algorithms effectively boost overall performance (high Macro-F1), the true robustness lies in their **ability to handle minority classes effectively**. Algorithms like TAM, GraphENS, and GraphSHA demonstrate a balanced approach, performing well across all classes and showcasing superior adaptability to imbalanced scenarios. There is still room for improvement, particularly for IGL algorithms that show a significant performance drop on **extremely minority classes**.
>
> We believe this additional analysis enriches our study and offers a clearer picture of each algorithm's effectiveness. Thank you for the valuable suggestion.

---

> ### Author Response · Authors · 2024-11-18
> **Response to Weakness #2 (Part I)**
>
> > **Weakness #2:** Some findings lack depth and do not offer clear insights. For instance, the observation that most algorithms perform poorly on the large-scale ogbn-arXiv dataset is presented without an explanation. It would be valuable to investigate and discuss the reasons for this poor performance, especially through additional experiments.
>
> **A:** We appreciate the reviewer's suggestion to provide a deeper analysis and additional insights into the performance of algorithms on the ogbn-arXiv dataset.
>
> - **Explanation.** The suboptimal performance on ogbn-arXiv can largely be attributed to the dataset's challenging characteristics.
>     - ogbn-arXiv contains **40 classes ([Table A.1](https://openreview.net/pdf?id=uTqnyF0JNR#page=16))** with a long-tailed distribution, which intensifies **the complexity of the classification task**. When **the number of classes is big**, class-imbalanced algorithms face **increased difficulty in managing inter-class boundaries** and distinguishing between **classes with limited samples**. This is particularly problematic in the context of imbalanced graph-based learning, where overlapping class distributions can lead to poor separation in the learned embeddings, adversely affecting classification accuracy, as indicated by very recent studies [4-6].
>     - ogbn-arXiv is inherently **highly class-imbalanced**. In the experimental setup, due to having more classes compared to other datasets, it contains **a greater number of extremely minority classes**. The skewed distribution can bias algorithms toward the majority classes, leading to poor generalization for minority classes and compromising the algorithm’s ability to accurately represent the overall class structure.
> - **Settings.** To systematically investigate the reasons, we conducted additional experiments following the review’s advice, in which we progressively reduced the number of classes in ogbn-arXiv. Specifically, we apply the method of "inverse interpolation" to reduce the number of classes to **1/4** (10 classes) and **1/8** (5 classes) of the original. Other settings keep the same with that of [Table D.3](https://openreview.net/pdf?id=uTqnyF0JNR#page=35) (Low, $\rho=$ 20). This allows us to preserve the intrinsic long-tail distribution pattern of the dataset, as well as decreasing the total number of classes.
> - **Results.**
>
> **Table:** Macro-F1 and per-class F1 score (% ± standard deviation) of node classification on manipulated class-imbalanced **ogbn-arXiv (10 classes, Low)** over 10 runs.
>
> | Algorithm | Macro-F1 | C. #0 | C. #1 | C. #2 | C. #3 | C. #4 | C. #5 | C. #6 | C. #7 | C. #8 | C. #9 |
> | --- | --- | --- | --- | --- | --- | --- | --- | --- | --- | --- | --- |
> | # nodes | 3010 | 47 | 65 | 89 | 123 | 168 | 231 | 317 | 434 | 596 | 940 |
> | GCN (bb.) | 40.4  | 0.0 | 0.0 | 22.7  | 12.0  | 22.0  | 57.6  | 87.9  | 53.1  | 63.7  | 85.3  |
> | DRGCN | 46.5  | 0.0 | 36.9  | 0.0 | 21.8  | 25.0  | 89.0  | 87.9  | 54.7  | 62.4  | 86.8  |
> | DPGNN | 68.6  | 75.1  | 86.4  | 41.0  | 38.6  | 69.8  | 80.9  | 85.7  | 64.2  | 65.4  | 78.5  |
> | ImGAGN | 71.1  | 70.9  | 80.9  | 59.6  | 53.9  | 50.0  | 91.7  | 80.3  | 65.1  | 74.6  | 83.9  |
> | GraphSMOTE | 51.7  | 0.0  | 72.0  | 0.0  | 25.6  | 22.9  | 85.4  | 84.8  | 70.5  | 73.5  | 82.1  |
> | GraphENS | 78.7  | 73.2  | 88.5  | 86.6  | 50.1  | 78.1  | 80.5  | 96.2  | 73.0  | 72.8  | 88.0  |
> | GraphMixup | 77.0  | 83.4  | 87.3  | 87.3  | 12.1  | 68.7  | 92.6  | 95.8  | 76.1  | 78.9  | 88.0  |
> | LTE4G | 70.7  | 73.1  | 86.3  | 27.7  | 44.7  | 77.8  | 85.2  | 91.0  | 70.3  | 66.1  | 84.6  |
> | TAM | 77.7  | 71.4  | 87.8  | 82.0  | 51.3  | 78.3  | 80.4  | 86.5  | 82.0  | 72.5  | 84.5  |
> | TOPOAUC | 68.8  | 65.2  | 76.2  | 62.5  | 38.8  | 70.2  | 81.0  | 85.3  | 64.3  | 66.1  | 78.8  |
> | GraphSHA | 77.8  | 76.1  | 75.4  | 75.4  | 43.5  | 86.0  | 87.8  | 86.9  | 89.2  | 79.7  | 78.4  |

---

> ### Author Response · Authors · 2024-11-18
> **Response to Weakness #2 (Part II)**
>
> **Table:** Macro-F1 and per-class F1 score (% ± standard deviation) of node classification on manipulated class-imbalanced **ogbn-arXiv (5 classes, Low)** over 10 runs.
>
> | Algorithm | Macro-F1 | C. #0 | C. #1 | C. #2 | C. #3 | C. #4 |
> | --- | --- | --- | --- | --- | --- | --- |
> | # nodes | 1806 | 47 | 97 | 213 | 509 | 940 |
> | GCN (bb.) | 39.5  | 5.8  | 20.7  | 38.3  | 63.0  | 69.9  |
> | DRGCN | 45.4  | 3.6  | 47.9  | 40.5  | 61.7  | 73.4  |
> | DPGNN | 49.1  | 2.2  | 47.6  | 45.8  | 78.0  | 71.8  |
> | ImGAGN | 49.0  | 3.6  | 37.7  | 57.2  | 71.8  | 74.5  |
> | GraphSMOTE | 42.4  | 4.0  | 29.9  | 41.4  | 62.3  | 74.5  |
> | GraphENS | 57.2  | 25.1  | 57.9  | 54.8  | 70.9  | 77.2  |
> | GraphMixup | 54.5  | 7.5  | 52.7  | 60.4  | 81.3  | 70.7  |
> | LTE4G | 57.0  | 24.3  | 51.5  | 62.9  | 66.4  | 80.1  |
> | TAM | 60.9  | 27.2  | 53.4  | 67.6  | 77.8  | 78.4  |
> | TOPOAUC | 55.9  | 19.4  | 60.0  | 61.2  | 69.6  | 69.3  |
> | GraphSHA | 66.4  | 26.1  | 67.8  | 78.5  | 74.3  | 85.1  |
> - **Findings.** The experiments provide evidence that the poor performance on the large-scale ogbn-arXiv dataset is indeed due to the high number of classes and the severe class imbalance. The results emphasize the need for more sophisticated algorithms capable of handling extreme cases of multi-class imbalance in large-scale graph datasets. Additionally, they reveal that simplifying the class situations can significantly alleviate the performance degradation, but this does not solve the underlying challenge of effectively managing long-tail distributions in complex graph scenarios. These findings suggest a direction for future work, focusing on developing more robust and adaptable algorithms for high-class-count imbalanced settings.
>
> Thank you for this constructive suggestion, which has enabled us to delve deeper into the limitations of current algorithms on complex, large-scale datasets such as ogbn-arXiv. We believe this analysis will provide a clearer understanding of the limitations faced by current methods on multi-class, large-scale datasets and offer valuable insights for future improvements.

---

> ### Author Response · Authors · 2024-11-18
> **Response to Weakness #3 (Part I)**
>
> > **Weakness #3:** In Section 4.4, which assesses the efficiency of different models, the experiments are conducted on the Cora dataset, the smallest of the datasets used. This setup makes the time and space usage of the algorithms highly dependent on implementation details rather than their inherent algorithmic complexity. While the Appendix includes results for the ogbn-arXiv dataset, they cover only a subset of models, as other models encountered memory limitations. A theoretical analysis of the algorithms' complexities would provide a more meaningful comparison of their efficiency. Relying solely on runtime measurements can be misleading, as they may be influenced by implementation optimizations and do not necessarily reflect the algorithms' underlying efficiency.
>
> **A:** Thank you for the insightful feedback regarding our efficiency analysis. We would like to address the concerns raised and provide clarification on several key aspects.
>
> - **Comprehensive Dataset Coverage.** In **[Appendix D.4](https://openreview.net/pdf?id=uTqnyF0JNR#page=54)**, we conducted efficiency experiments not only on the **ogbn-arXiv** dataset but also on the **Actor** dataset. Together with the **Cora** dataset, this analysis spans a wide range of dataset scales, including small-scale (Cora), medium-scale (Actor), and large-scale (ogbn-arXiv) datasets. Moreover, these datasets vary in homophily, covering homophilic datasets (Cora and ogbn-arXiv) and a heterophilic dataset (Actor). This diversity ensures that our evaluation of the time and space efficiency of IGL algorithms reflects performance across different dataset dimensions, providing a comprehensive perspective.
> - **Empirical Analysis as a Standard Practice. Measuring training time and peak GPU memory consumption is a widely adopted approach for evaluating the time and space efficiency of algorithms**, particularly in benchmark studies [7, 8]. While we acknowledge that empirical results may be influenced by factors such as implementation details and hardware optimizations, we have taken care to control these variables as much as possible to ensure fairness in our comparisons. For example, **all experiments were conducted under consistent hardware and software configurations.**
> - **Existing Theoretical Complexity Analysis.** As noted in **[Appendix Table A.3](https://openreview.net/pdf?id=uTqnyF0JNR#page=18)**, we have already provided theoretical analyses of **the computational complexity for each IGL algorithm**. This includes an overview of the main bottlenecks in the algorithms’ computational processes. As indicated in Footnote 3 of [Table A.3](https://openreview.net/pdf?id=uTqnyF0JNR#page=18), the analysis focuses on the primary computationally expensive operations while uniformly ignoring negligible contributions for brevity. To further address the reviewer’s concern, we have expanded this analysis in the following and provided **a more detailed breakdown of the computational complexities** for each node-level algorithm.

---

> ### Author Response · Authors · 2024-11-18
> **Response to Weakness #3 (Part II)**
>
> - **Further Analysis.**
>
> | **Algorithm** | **Time Complexity** | **Space Complexity** | **Explanation** (we denote the number of nodes and edges as $\|\mathcal{V}\|$and $\|\mathcal{E}\|$, respectively; the feature and hidden dimension as $d$.) |
> | --- | --- | --- | --- |
> | DRGCN | **1.** GNN layer: $\mathcal{O}(\|\mathcal{E}\|+\|\mathcal{V}\|d)$.**2.** Adversarial training: $\mathcal{O}(MN)$.**3.** Latent distribution alignment: $\mathcal{O}(\|\mathcal{V}\|d^2)$.**4.** Overall: $\mathcal{O}(\|\mathcal{E}\|+\|\mathcal{V}\|d+MN+\|\mathcal{V}\|d^2)$. | **1.** Adj and feature matrices: $\mathcal{O}(\|\mathcal{E}\|+\|\mathcal{V}\|d)$.**2.** GNN parameters: $\mathcal{O}(d^2)$.**3.** Latent distribution: $\mathcal{O}(d^2)$.**4.** Overall: $\mathcal{O}(\|\mathcal{E}\|+\|\mathcal{V}\|d+d^2)$. | **1.** $M$: the number of iterations for adversarial training.**2.** $N$: the batch size. |
> | DPGNN | **1.** Distance metric learning: $\mathcal{O}(\|\mathcal{V}\|C)$.**2.** Imbalanced label propagation: $\mathcal{O}(k\|\mathcal{E}\|)$.**3.** Self-supervised learning: $\mathcal{O}(\|\mathcal{E}\|)$.**4.** Overall: $\mathcal{O}(\|\mathcal{V}\|+(k+1)\|\mathcal{E}\|)$. | **1.** Adj and feature matrices: $\mathcal{O}(\|\mathcal{E}\|+\|\mathcal{V}\|d)$.**2.** GNN parameters: $\mathcal{O}(d^2)$.**3.** Distance metric layer: $\mathcal{O}(Cd^2)$.**4.** Overall: $\mathcal{O}(\|\mathcal{E}\|+\|\mathcal{V}\|d+(C+1)d^2)$. | **1.** $C$: the number of classes.**2.** $k$: the number of propagation hops. |
> | ImGAGN | **1.** Generator: $\mathcal{O}((L - 1)n_gH^2 + n_gn_m^2)$.**2.** Discriminator: $\mathcal{O}(K\|\mathcal{E}\|d + K\|\mathcal{V}\| d^2)$.**3.** Overall: $\mathcal{O}((L - 1)n_gH^2 + n_gn_m^2 + \lambda(K\|\mathcal{E}\|d + K\|\mathcal{V}\| d^2))$. | **1.** Generator: $\mathcal{O}((L - 1)n_gH^2 + n_gn_m^2)$.**2.** Discriminator: $\mathcal{O}(\|\mathcal{V}'\| + \|\mathcal{E}'\| + d^2)$.**3.** Overall:  $\mathcal{O}(LH^2 + n_gH + \|\mathcal{V}'\| + \|\mathcal{E}'\| + d^2)$. | **1.** $L$: the number of fully connected layers.**2.** $H$: the hidden layer dimension.**3.** $n_g$: the number of generated minority nodes.**4.** $n_m$: the number of minority nodes.**5.** $\lambda$:  the number of discriminator training steps per generator training.**6.** $K$: the number of GCN layers.**7.** $\mathcal{V}'$, $\mathcal{E}'$: nodes and edges in the augmented graph. |
> | GraphSMOTE | **1.** GNN layer: $\mathcal{O}(\|\mathcal{E}\|+\|\mathcal{V}\|d)$.**2.** Synthetic node: $\mathcal{O}(k\|\mathcal{V}\|)$.**3.** Edge predictor: $\mathcal{O}(\|\mathcal{E}\|+\|\mathcal{V}\|)$.**4.** Overall: $\mathcal{O}(\|\mathcal{E}\|+(d+k+1)\|\mathcal{V}\|)$. | **1.** Adj and feature matrices: $\mathcal{O}(\|\mathcal{E}\|+\|\mathcal{V}\|d)$.**2.** Synthetic node: $\mathcal{O}(s+sd)$.**3.** GNN parameters: $\mathcal{O}(d^2)$.**4.** Overall: $\mathcal{O}(\|\mathcal{E}\|+\|\mathcal{V}\|d+s+sd+d^2)$. | **1.** $k$: the number of nearest neighbors.2. $s$: the number of synthetic nodes. |
> | GraphENS | **1.** Neighbor sampling: $\mathcal{O}(\|\mathcal{E}\|)$.**2.** Saliency-based node mixing: $\mathcal{O}(\|\mathcal{V}\|d)$.**3.** Message passing: $\mathcal{O}(L\|\mathcal{E}\|)$.**4.** Overall: $\mathcal{O}((L+1)\|\mathcal{E}\|+\|\mathcal{V}\|d)$. | **1.** Adj and feature matrices: $\mathcal{O}(\|\mathcal{E}\|+\|\mathcal{V}\|d)$.**2.** Saliency  computation: $\mathcal{O}(\|\mathcal{V}\|d)$.**3.** GNN parameters: $\mathcal{O}(d^2)$.**4.** Overall: $\mathcal{O}(\|\mathcal{E}\|+\|\mathcal{V}\|d+d^2)$. |  |
> | GraphMixup | **1.** Constructing semantic relation graphs: $\mathcal{O}(\|\mathcal{V}\|^2d)$.**2.** Feature aggregation: $\mathcal{O}(K\|\mathcal{V}\|^2d)$.**3.** Edge prediction: $\mathcal{O}(\|\mathcal{V}\|^2d)$.**4.** Context-based self-supervised prediction: $\mathcal{O}(\|\mathcal{E}\|+\|\mathcal{V}\|)$.**5.** Reinforcement learning: $\mathcal{O}(E\|\mathcal{V}\|d)$.**6.** Overall: $\mathcal{O}((K+1)\|\mathcal{V}\|^2d+(Ed+1)\|\mathcal{V}\|+\|\mathcal{E}\|)$. | **1.** Adj and feature matrices: $\mathcal{O}(K\|\mathcal{E}\|+K\|\mathcal{V}\|d)$.**2.** Edge predictor: $\mathcal{O}(\|\mathcal{E}\|+\|\mathcal{V}\|)$.**3.** Reinforcement learning: $\mathcal{O}(\|\mathcal{V}\|)$.**4.** GNN parameters: $\mathcal{O}(d^2)$.**5.** Overall: $\mathcal{O}((K+1)\|\mathcal{E}\|+(Kd+1)\|\mathcal{V}\|+d^2)$. | **1.** $K$: the number of semantic relation space.**2.** $E$: the number of reinforced learning epochs. |

---

> ### Author Response · Authors · 2024-11-18
> **Response to Weakness #3 (Part III)**
>
> | **Algorithm** | **Time Complexity** | **Space Complexity** | **Explanation** (we denote the number of nodes and edges as $\|\mathcal{V}\|$and $\|\mathcal{E}\|$, respectively; the feature and hidden dimension as $d$.) |
> | --- | --- | --- | --- |
> | LTE4G | **1.** Graph preprocessing: $\mathcal{O}(\|\mathcal{E}\|+\|\mathcal{V}\|)$.**2.** GNN layer: $\mathcal{O}(\|\mathcal{E}\|+\|\mathcal{V}\|d)$.**3.** Knowledge distillation: $\mathcal{O}(KM\|\mathcal{V}\|)$.**4.** Prototype-based inference: $\mathcal{O}(\|\mathcal{V}\|d)$.**5.** Overall: $\mathcal{O}(\|\mathcal{E}\|+(d+1+KM)\|\mathcal{V}\|)$. | **1.** Adj and feature matrices: $\mathcal{O}(\|\mathcal{E}\|+\|\mathcal{V}\|d)$.**2.** GNN and classifiers (experts + students) parameters: $\mathcal{O}(Ld^2+Cd)$.**3.** Overall: $\mathcal{O}(\|\mathcal{E}\|+\|\mathcal{V}\|d+Ld^2+Cd)$. | **1.** $K$: the number of experts.**2.** $M$: the number of students.**3.** $C$: the number of classes. |
> | TAM | **1.** GNN layer: $\mathcal{O}(\|\mathcal{E}\|+\|\mathcal{V}\|d)$.**2.** Neighbor label distribution: $\mathcal{O}(\|\mathcal{E}\|)$.**3.** Class-wise connectivity matrix: $\mathcal{O}(\|\mathcal{V}\|)$.**4.** Margin adjustments: $\mathcal{O}(\|\mathcal{V}\|C)$.**5.** Overall: $\mathcal{O}(\|\mathcal{E}\|+(d+1+C)\|\mathcal{V}\|)$. | **1.** Adj and feature matrices: $\mathcal{O}(\|\mathcal{E}\|+\|\mathcal{V}\|d)$.**2.** Class-wise connectivity matrix and margins: $\mathcal{O}(C^2+\|\mathcal{V}\|C)$.**3.** GNN parameters: $\mathcal{O}(d^2)$.**4.** Overall: $\mathcal{O}(\|\mathcal{E}\|+(d+C)\|\mathcal{V}\|+C^2+d^2)$. | **1.** $C$: the number of classes. |
> | TOPOAUC | **1.** GNN layer: $\mathcal{O}(\|\mathcal{E}\|+\|\mathcal{V}\|d)$.**2.** Topology-aware importance learning (TAIL): $\mathcal{O}(\|\mathcal{E}\|)$.**3.** AUC optimization: $\mathcal{O}(PN)$.**4.** Overall: $\mathcal{O}(\|\mathcal{E}\|+\|\mathcal{V}\|d+PN)$. | **1.** Adj and feature matrices: $\mathcal{O}(\|\mathcal{E}\|+\|\mathcal{V}\|d)$.**2.** Topology influence matrix: $\mathcal{O}(\|\mathcal{E}\|)$.**3.** GNN parameters: $\mathcal{O}(d^2)$.**4.** Overall: $\mathcal{O}(\|\mathcal{E}\|+\|\mathcal{V}\|d+d^2)$. | **1.** $P$: the number of positive samples.**2.** $N$: the number of positive and negative samples. |
> | GraphSHA | **1.** GNN layer: $\mathcal{O}(\|\mathcal{E}\|+\|\mathcal{V}\|d)$.**2.** Anchor node and auxiliary node sampling: $\mathcal{O}(\|\mathcal{V}\|\_{tr}^2)$.**3.** Node feature synthesis: $\mathcal{O}(d\|\mathcal{V}\|\_{tr})$.**4.** Edge generation: $\mathcal{O}(\|\mathcal{V}\|\_{tr}/\|\mathcal{V}\|\cdot\|\mathcal{E}\|)$.**5.** Overall: $\mathcal{O}(\|\mathcal{E}\|+\|\mathcal{V}\|d+\|\mathcal{V}\|\_{tr}^2+d\|\mathcal{V}\|\_{tr}+\|\mathcal{V}\|\_{tr}/\|\mathcal{V}\|\cdot\|\mathcal{E}\|)$. | **1.** Adj and feature matrices: $\mathcal{O}(\|\mathcal{E}\|+\|\mathcal{V}\|d)$.**2.** Synthesized nodes and edges: $\mathcal{O}(d\|\mathcal{V}\|\_{tr}+\|\mathcal{E}\|\_{s})$.**3.** GNN parameters: $\mathcal{O}(d^2)$.**4.** Overall: $\mathcal{O}(\|\mathcal{E}\|+\|\mathcal{V}\|d+d\|\mathcal{V}\|\_{tr}+\|\mathcal{E}\|\_{s}+d^2)$. | **1.** $\|\mathcal{V}\|\_{tr}$: the number of training nodes.**2.** $\|\mathcal{E}\|\_{s}$: the number of edges generated for the synthesized nodes. |
> | DEMO-Net | **1.** GNN layer: $\mathcal{O}(\|\mathcal{E}\|+\|\mathcal{V}\|d)$.**2.** Degree-specific functions: $\mathcal{O}(Nd)$.**3.** Hash operations: $\mathcal{O}(\|\mathcal{V}\|d^2)$.**4.** Overall: $\mathcal{O}(\|\mathcal{E}\|+(\|\mathcal{V}\|+N)d+\|\mathcal{V}\|d^2)$. | **1.** Adj and feature matrices: $\mathcal{O}(\|\mathcal{E}\|+\|\mathcal{V}\|d)$.**2.** GNN parameters: $\mathcal{O}(d^2)$.**3.** Overall: $\mathcal{O}(\|\mathcal{E}\|+\|\mathcal{V}\|d+d^2)$. | **1.** $N$: the number of tasks. |
> | meta-tail2vec | **1.** Neighbor aggregation: $\mathcal{O}(M\cdot2\|\mathcal{E}\|/\|\mathcal{V}\|)$.**2.** Meta-learning: $\mathcal{O}(NKM\cdot2\|\mathcal{E}\|/\|\mathcal{V}\|)$.**3.** Overall: $\mathcal{O}(NKM\cdot2\|\mathcal{E}\|/\|\mathcal{V}\|)$. | **1.** Adj and feature matrices: $\mathcal{O}(\|\mathcal{E}\|+\|\mathcal{V}\|d)$.**2.** GNN parameters: $\mathcal{O}(d^2)$.**3.** Overall: $\mathcal{O}(\|\mathcal{E}\|+\|\mathcal{V}\|d+d^2)$. | **1.** $M$ the number of neighbor hops.**2.** $N$: the number of tasks.**3.** $K$: the number of supporting nodes. |
> | Tail-GNN | **1.** GNN layer: $\mathcal{O}(\|\mathcal{E}\|+\|\mathcal{V}\|d)$.**2.** Localization and translation: $\mathcal{O}(\|\mathcal{V}\|)$.**3.** Overall: $\mathcal{O}(\|\mathcal{E}\|+(d+1)\|\mathcal{V}\|)$. | **1.** Adj and feature matrices: $\mathcal{O}(\|\mathcal{E}\|+\|\mathcal{V}\|d)$.**2.** GNN parameters: $\mathcal{O}(d^2)$.**3.** Overall: $\mathcal{O}(\|\mathcal{E}\|+\|\mathcal{V}\|d+d^2)$. |  |

---

> ### Author Response · Authors · 2024-11-18
> **Response to Weakness #3 (Part IV)**
>
> | **Algorithm** | **Time Complexity** | **Space Complexity** | **Explanation** (we denote the number of nodes and edges as $\|\mathcal{V}\|$and $\|\mathcal{E}\|$, respectively; the feature and hidden dimension as $d$.) |
> | --- | --- | --- | --- |
> | Cold Brew | **1.** Sample transformation: $\mathcal{O}(KT\|\mathcal{V}\|d)$.**2.** GNN layer: $\mathcal{O}(\|\mathcal{E}\|+\|\mathcal{V}\|d)$.**3.** Overall: $\mathcal{O}(\|\mathcal{E}\|+(KT+1)\|\mathcal{V}\|d)$. | **1.** Adj and feature matrices: $\mathcal{O}(\|\mathcal{E}\|+\|\mathcal{V}\|d)$.**2.** Jacobian matrix: $\mathcal{O}(KTd)$.**3.** GNN parameters: $\mathcal{O}(d^2)$.**4.** Batch augmentation: $\mathcal{O}(B\|\mathcal{V}\|)$.**5.** Overall: $\mathcal{O}(\|\mathcal{E}\|+(\|\mathcal{V}\|+KT+B)d+d^2)$. | **1.** $K$: the number of transformation times.**2.** $T$: the number of operations times.**3.** $B$: the batch size. |
> | RawlsGCN | **1.** GNN layer: $\mathcal{O}(\|\mathcal{E}\|+\|\mathcal{V}\|d)$.**2.** Sinkhorn-Knopp algorithm: $\mathcal{O}(K(m+\|\mathcal{V}\|))$.**3.** Overall: $\mathcal{O}(\|\mathcal{E}\|+\|\mathcal{V}\|d+K(m+\|\mathcal{V}\|))$. | **1.** Adj and feature matrices: $\mathcal{O}(\|\mathcal{E}\|+\|\mathcal{V}\|d)$.**2.** The doubly stochastic matrix: $\mathcal{O}(m+\|\mathcal{V}\|)$.**3.** GNN parameters: $\mathcal{O}(d^2)$.**4.** Overall: $\mathcal{O}(\|\mathcal{E}\|+(d+1)\|\mathcal{V}\|+m+d^2)$. | **1.** $K$: the number of iterations until convergence.**2.** $m$: is the number of non-zero elements in the doubly stochastic matrix. |
> | GraphPatcher | **1.** GNN layer: $\mathcal{O}(\|\mathcal{E}\|+\|\mathcal{V}\|d)$.**2.** Test-Time Augmentation: $\mathcal{O}(ML\|\mathcal{V}\|)$.**3.** Prediction reconstruction: $\mathcal{O}(\|\mathcal{V}\|)$.**4.** Overal: $\mathcal{O}(\|\mathcal{E}\|+(d+ML+1)\|\mathcal{V}\|)$. | **1.** Adj and feature matrices: $\mathcal{O}(\|\mathcal{E}\|+\|\mathcal{V}\|d)$.**2.** Ego-graph storage: $\mathcal{O}(K\|\mathcal{V}\|)$.**3.** Virtual nodes: $\mathcal{O}(M\|\mathcal{V}\|)$.**4.** GNN parameters: $\mathcal{O}(d^2)$.**5.** Overall: $\mathcal{O}(\|\mathcal{E}\|+(K+d+M)\|\mathcal{V}\|+d^2)$. | **1.** $M$: the number of patching steps.**2.** $L$: the number of sampled ego-graphs per corruption strength.**3.** $K$: the average number of neighbors. |
> | ReNode | **1.** GNN layer: $\mathcal{O}(\|\mathcal{E}\|+\|\mathcal{V}\|d)$.**2.** Personalized PageRank: $\mathcal{O}(K(\|\mathcal{E}\|+\|\mathcal{V}\|))$.**3.** Influence conflict: $\mathcal{O}(\|\mathcal{V}\|(\|\mathcal{E}\|+\|\mathcal{V}\|))$.**4.** Ranking and reweighting: $\mathcal{O}(\|\mathcal{V}\|\log \|\mathcal{V}\|)$.**5.** Overall: $\mathcal{O}((K+\|\mathcal{V}\|+1)\|\mathcal{E}\|+(d+K+\|\mathcal{V}\|+\log \|\mathcal{V}\|)\|\mathcal{V}\|)$. | **1.** Adj and feature matrices: $\mathcal{O}(\|\mathcal{E}\|+\|\mathcal{V}\|d)$.**2.** PageRank matrix (sparse): $\mathcal{O}(\|\mathcal{E}\|+\|\mathcal{V}\|)$.**3.** GNN parameters: $\mathcal{O}(d^2)$.**3.** Overall: $\mathcal{O}(\|\mathcal{E}\|+(d+1)\|\mathcal{V}\|+d^2)$. |  |
> | PASTEL | **1.** GNN layer: $\mathcal{O}(\|\mathcal{E}\|+\|\mathcal{V}\|d)$.**2.** Shortest paths: $\mathcal{O}(C\|\mathcal{V}\|^2)$.**3.** Group PageRank: $\mathcal{O}(\|\mathcal{V}\|^2)$.**4.** Overall: $\mathcal{O}(\|\mathcal{E}\|+\|\mathcal{V}\|d+(C+1)\|\mathcal{V}\|^2)$. | **1.** Adj and feature matrices: $\mathcal{O}(\|\mathcal{E}\|+\|\mathcal{V}\|d)$.**2.** Position encodings: $\mathcal{O}(\|\mathcal{V}\|d)$.**3.** Class-wise conflict measure: $\mathcal{O}(\|\mathcal{E}\|)$.**4.** GNN parameters: $\mathcal{O}(d^2)$.**5.** Overall: $\mathcal{O}(\|\mathcal{E}\|+\|\mathcal{V}\|d+d^2)$. | **1.** $C$: the number of classes. |
> | HyperIMBA | **1.** HMPNN layer: $\mathcal{O}(\|\mathcal{E}\|+\|\mathcal{V}\|d)$.**2.** Poincaré embedding: $\mathcal{O}(\|\mathcal{V}\|Td)$.**3.** Hierarchy-aware margin calculation: $\mathcal{O}(\|\mathcal{V}\|k)$.**4.** Ricci curvature: $\mathcal{O}(\|\mathcal{E}\|d^2)$.**5.** Overall: $\mathcal{O}((d^2+1)\|\mathcal{E}\|+(Td+1)\|\mathcal{V}\|)$. | **1.** Adj and feature matrices: $\mathcal{O}(\|\mathcal{E}\|+\|\mathcal{V}\|d)$.2. Poincaré embedding: $\mathcal{O}(\|\mathcal{V}\|d)$.**3.** Ricci curvature: $\mathcal{O}(\|\mathcal{E}\|)$.**4.** GNN parameters: $\mathcal{O}(d^2)$.**5.** Overall: $\mathcal{O}(\|\mathcal{E}\|+\|\mathcal{V}\|d+d^2)$. | **1.** $T$: the number of iterations for optimization.**2.** $k$: the number of parameters in an MLP. |

---

> ### Author Response · Authors · 2024-11-18
> **Response to Weakness #3 (Part V)**
>
> - **Analysis.**
>     - **Compared Figure D.14 with Figure D.15.** We found that GCN, GraphMixup, GraphENS, TAM and GraphSHA ran successfully on ogbn-arXiv, while DRGCN, DPGNN, ImGAGN, GraphSMOTE, LTE4G, and TOPOAUC failed. This is because: **(1)** GraphMixup’s complexity is linear with respect to the number of edges for most graph operations, which is efficient for real-world sparse graphs. **(2)** GraphSHA provides an efficient solution for handling large-scale, class-imbalanced graph data by introducing only moderate and manageable computational and memory overheads, thereby improving model performance without compromising efficiency. **(3)** AUC optimization in TOPOAUC involves pairwise comparisons across potentially massive sets of positive and negative samples, leading to rapid growth in memory usage that overwhelms hardware limits.
>     - **Compared Figure D.16 with Figure D.17.** We found only Tail-GNN failed to run on ogbn-arXiv. This is because its graph convolution and message-passing operations require significant memory, especially when dealing with large graphs and numerous neighbors.
>     - **Compared Figure D.18 with Figure D.19.** We found that only GCN and ReNode ran successfully on ogbn-arXiv, while PASTEL, TOPOAUC, and HyperIMBA failed. This is because: **(1)** Iterative graph structure learning in PASTEL requires at least quadratic complexity both for time and space complexity, which is infeasible for the large-scale ogbn-arXiv. **(2)** The high complexity of operations like Poincaré embedding, Ricci curvature calculation, and hierarchical message-passing leads to substantial overhead for HyperIMBA. **(3)** The same reason caused b AUC optimization.
>
> Thanks again or your suggestions! We will add detailed theoretical time and space complexity analysis, which could bring more fair and accurate evaluations for the efficiency of IGL algoithms, together with the empirical studies.

---

> ### Author Response · Authors · 2024-11-18
> **Response to Weakness #4**
>
> >**Weakness #4:** On the ogbn-arXiv dataset, certain resampling algorithms, such as GraphENS and GraphSHA, run successfully, whereas others, like GraphSMOTE, encounter memory issues. It is unclear whether this is due to differences in algorithmic complexity or specific implementation choices. Given the similarity of these resampling approaches, clarifying whether the issue arises from inherent algorithm characteristics or particular implementation choices would be helpful.
>
> **A:** Thanks for your suggestions. We would like to address this concern in details.
>
> - **Experimental Environment ([Appendix C.4](https://openreview.net/pdf?id=uTqnyF0JNR#page=33)).** All experiments were conducted in a consistent software and hardware environment to minimize confounding factors arising from variations in hardware or runtime optimizations. Specifically, our setup includes:
>     - **Operating System**: Ubuntu 20.04 LTS
>     - **CPU**: Intel(R) Xeon(R) Platinum 8358 CPU @ 2.60GHz with 1TB DDR4 of Memory
>     - **GPU**: NVIDIA Tesla A100 SMX4 with **40GB** of Memory
>     - **Software**: CUDA 10.1, Python 3.8.12, PyTorch 1.9.1, and PyTorch Geometric 2.0.1
>
>     This robust environment ensures sufficient computational resources to support large-scale graph datasets and complex algorithmic operations. Any observed memory limitations are therefore likely due to algorithmic or implementation factors rather than hardware constraints.
>
> - **Algorithmic Complexity Analysis.** In [Table A.3](https://openreview.net/pdf?id=uTqnyF0JNR#page=18) and response to last question, we provide a theoretical analysis of the computational complexity for all benchmarked algorithms, including resampling methods like GraphENS, GraphSHA, and GraphSMOTE. While all resampling approaches share the paradigm of addressing imbalances via data augmentation, key differences in their specific methodologies affect their memory and computational requirements:
>     - **GraphENS** synthesizes local ego-networks, focusing on small, localized regions of the graph, which makes it more memory-efficient. The localized nature makes it inherently memory-efficient, as it avoids global graph manipulations and redundant computations. Sparse matrix operations and selective sampling further reduce overhead.
>     - **GraphSHA** selectively generates synthetic nodes and edges, guided by homophily-based sampling strategies to ensure structural consistency. This method avoids unnecessary computation by focusing only on key nodes and regions of the graph. It achieves a balance between computational efficiency and augmentation quality.
>     - **GraphSMOTE**, in contrast, synthesizes new node embeddings while also introducing additional graph-level computations. The dense matrix operations, combined with graph-wide connectivity updates, result in significantly higher memory and computational requirements.
> - **Conclusion.** The differences in memory performance among GraphENS, GraphSHA, and GraphSMOTE are due to their fundamental algorithmic designs. While GraphENS and GraphSHA employ localized or selective augmentation, GraphSMOTE’s global operations and dense computations lead to higher memory requirements. This is why it encountered memory issues on the large-scale ogbn-arXiv. These findings emphasize the trade-offs between augmentation scope and computational efficiency.
>
> We appreciate the reviewer’s suggestion and will work toward further optimizing and analyzing these algorithms in future updates.

---

> ### Author Response · Authors · 2024-11-18
> **Conclusion**
>
> Thank you again for your time and valuable insights, and hope our responses can address all your concerns! We are excited about the impact IGL-Bench may have on the community and are grateful for your support in advancing this work.
>
> **We would be even encouraging if you could kindly consider raising the ratings!**
>
> ---
>
> **References:**
>
> **[1]** GraphSHA: Synthesizing harder samples for class-imbalanced node classification. Li W Z, et al. Proceedings of the 29th ACM SIGKDD Conference on Knowledge Discovery and Data Mining, 2023.
>
> **[2]** The class imbalance problem: A systematic study. Japkowicz N, et al. Intelligent data analysis, 2002.
>
> **[3]** Disentangling label distribution for long-tailed visual recognition. Hong Y, et al. Proceedings of the IEEE/CVF conference on computer vision and pattern recognition, 2021.
>
> **[4]** Mastering long-tail complexity on graphs: characterization, learning, and generalization. Proceedings of the 30th ACM SIGKDD Conference on Knowledge Discovery and Data Mining, 2024.
>
> **[5]** Decaf: Deep extreme classification with label features. Mittal A, et al. Proceedings of the 14th ACM International Conference on Web Search and Data mining, 2021.
>
> **[6]** A review on multi-label learning algorithms. Zhang M L, Zhou Z H. IEEE Transactions on Knowledge and Data Engineering, 2013.
>
> **[7]** GSLB: the graph structure learning benchmark. Li Z, et al. Advances in Neural Information Processing Systems, 2024.
>
> **[8]** OpenGSL: A comprehensive benchmark for graph structure learning. Zhiyao Z, et al. Advances in Neural Information Processing Systems, 2024.

---

> ### Author Response · Authors · 2024-11-25
>
> Thank you very much for taking the time to review our work thoroughly. We truly appreciate your constructive feedback, which helped us improve the quality of our manuscript. Your support and positive evaluation mean a lot to us. We will spare no effort to incorporate your suggestions, the added experimental results, and in-depth analysis into the camera-ready version. We look forward to your follow-up attention!

---

### Official Review · Reviewer_6GyC · 2024-11-04

**Soundness:** 3
**Presentation:** 3
**Contribution:** 3
**Rating:** 8
**Confidence:** 3

**Summary:**

This paper addresses the gap in graph learning by introducing a benchmark named IGL-Bench for imbalanced graph learning. The benchmark covers 24 algorithms across 17 datasets, tackling node-level and graph-level tasks that are affected by class and topology imbalances. The benchmark includes standardized data processing, evaluation protocols, and metrics across different datasets. Evaluation on effectiveness, robustness, boundary clarity, and efficiency of IGL methods under imbalanced conditions are further carried out. The benchmark findings reveal key insights into the strengths and limitations of current IGL algorithms, particularly around scalability, robustness, and their dependency on graph topology properties.

**Strengths:**

- Originality: The paper introduces a comprehensive benchmark for IGL to fill the gap in graph learning.
- Quality: Experiments valiadate the quality of the benchmark.
- Clarity: The paper is well written and organized. Easy for general audience to follow.
- Significance: The benchmark provides an easy platform for developing or comparing IGL methods, especially as graph learning expands into real-world, imbalanced datasets.

**Weaknesses:**

- Complex visualizations: Certain visualizations, such as multi-metric plots across multiple imbalance ratios, could benefit from clearer explanations to aid interpretability for readers less familiar with the metrics.
- New dataset: No new dataset is introduced in the paper.

**Questions:**

- Can the authors elaborate on the specific challenges encountered in standardizing data splits and evaluation protocols across such a diverse set of IGL methods?
- How does IGL-Bench handle algorithm hyperparameter tuning, particularly for methods that might be sensitive to different types of imbalance?
- Are there future plans to study the scalability within IGL-Bench, e.g., for larger, more complex graph datasets?

---

> ### Author Response · Authors · 2024-11-18
> **Overall Response to Reviewer 6GyC**
>
> Thank you for your thoughtful and encouraging feedback on our paper. We greatly appreciate your recognition of the originality and significance of the IGL-Bench benchmark, as well as the validation of its quality through experiments. We are also glad that you found the paper clear and well-organized, and that the benchmark’s potential as a platform for advancing IGL research was highlighted. We respond to your concerns and questions as follows.
>
> **[Preview]**
>
> - **Weakness #1:** Complex visualizations.
> - **Weakness #2:** Lack new dataset.
> - **Question #1:** Declare specific challenges.
> - **Question #2:** Hyperparameter tuning.
> - **Question #3:** Study on scalable and complex graphs.
> - **Conclusion.**

---

> ### Author Response · Authors · 2024-11-18
> **Response to Weakness #1**
>
> >**Weakness #1:** Complex visualizations: Certain visualizations, such as multi-metric plots across multiple imbalance ratios, could benefit from clearer explanations to aid interpretability for readers less familiar with the metrics.
>
> **A:** Thank you for your insightful feedback. We provide a more thorough explanation, particularly Figure 3, Figure 4, Figure 5, and Figure 6. Specifically,
>
> - **Figure 3:**
>     - **Settings.** This figure presents the robustness analysis of **node-level algorithms** under varying **class-imbalance levels** (**Balanced, Low, and High**) on **Cora** (homophilic). Accuracy and its relative decrease compared to the balanced split are shown. Similar comparisons for CiteSeer (homophilic), Chameleon (heterophilic), and Squirrel (heterophilic) are in Figures D.2, D.3, and D.4. ([Link](https://openreview.net/pdf?id=uTqnyF0JNR#page=48))
>     - **Interpret. (1) Bar Chart.** A single bar (same color) compares between all IGL algorithms. **A set of bars** (for one algorithm) shows robustness across **different imbalance levels**. **(2) Line Plot.** The slope indicates how well the algorithm maintains performance (starting from 100%) under extreme imbalance (**a flatter slope shows better control**). **Inflection points reveal preferences for certain imbalance levels**. Both charts should be analyzed together.
> - **Figure 4:**
>     - **Settings.** This figure analyzes the robustness of **node- and graph-level algorithms across different imbalance levels**, showing performance **(Accuracy) with standard deviation (shaded area)**. Figure 4 provides an overview, with further results in Figures D.5, D.6, D.7, and D.8 ([Link](https://openreview.net/pdf?id=uTqnyF0JNR#page=50)), covering nearly all datasets in our benchmark.
>     - **Interpret. (1) Vertical comparison.** Comparing data points on the vertical axis at **a fixed imbalance level** reveals the strengths and weaknesses of IGL algorithms. **(2) Horizontal comparison.** Examining **a single algorithm across imbalance levels** shows its robustness. **(3) Inflection and crossover points.** Inflection points indicate an algorithm’s preference for specific imbalance levels, while crossover points highlight differing robustness sensitivities among algorithms.
> - **Figure 5:**
>     - **Settings.** This figure visualizes learned node and graph representations from IGL algorithms using **t-SNE**, showing classifier boundaries with **samples colored by predicted class labels** and **the Silhouette Score**. Figure 5 previews **Cora** and **COLLAB**, while Figures D.9, D.10, D.11, and D.12 ([Link](https://openreview.net/pdf?id=uTqnyF0JNR#page=52)) present results for most datasets in our benchmark.
>     - **Interpret. (1) Classification boundaries.** The distribution of sample points (in different colors) reflects the classifier’s effectiveness. **Clear boundaries indicate strong performance on imbalanced graph data**, while overlapping boundaries suggest difficulty in distinguishing similar samples. **(2) Silhouette score.** This score quantitatively measures clustering quality, ranging from -1 to 1. **Higher values** indicate **tighter clustering**, reflecting the algorithm's ability to group similar samples and separate dissimilar ones.
> - **Figure 6:**
>     - **Settings.** This figure analyzes the time and space efficiency of **node- and graph-level IGL algorithms**, showing **training time (s)**, **training space (MB)**, and algorithm **performance (Accuracy)**. Figure 6 provides an overview, with Figures D.14 to D.21 ([Link](https://openreview.net/pdf?id=uTqnyF0JNR#page=54)) covering most datasets and IGL algorithms under various imbalance settings.
>     - **Interpret. (1) Training time.** This measures the duration from **training start to convergence**. An algorithm with shorter training time and higher accuracy is preferable, with **superior** algorithms appearing **near the top-left** of the visualization. **(2) Training space.** This indicates the **peak GPU memory** needed during training. Algorithms requiring **less space** and achieving higher accuracy are also preferable, shown **near the top-left**. Algorithms exceeding available memory are marked with a red “❌” (Out-of-Memory, OOM).
>
> We will add the above explanations in the revised paper. Thank you once again for your valuable feedback and thoughtful suggestions. We hope our further explanations have addressed your concerns.

---

> ### Author Response · Authors · 2024-11-18
> **Response to Weakness #2**
>
> > **Weakness #2:** New dataset: No new dataset is introduced in the paper.
>
> **A:** We appreciate the reviewer's comment. Though we choose “datasets and benchmarks” as the primary area, the **core contributions** of our work are proposing **the first comprehensive benchmark for Imbalanced Graph Learning (IGL)**, which do not necessarily require the introduction of new datasets. Specifically, our contributions are as follows:
>
> - **Algorithm Taxonomy and Quantification.** Our work is the **first** to systematically categorize existing IGL algorithms, providing **a clear framework for understanding various types of imbalanced issues**. We define each category, explain their unique challenges, and **propose quantification methods for imbalances ([Table 1](https://openreview.net/pdf?id=uTqnyF0JNR#page=4))**. This contribution provides a foundation for researchers to understand and address different IGL problems effectively.
> - **Dataset Preparation Rule.** We propose **standardized guidelines for dataset preparation**, including data processing approaches and **imbalanced data-splitting strategies**. As highlighted in the paper, the variability in dataset usage and **the inconsistency in how imbalances are addressed across studies** have made it difficult to compare results effectively. By introducing a clear framework for dataset preparation, we ensure that future research can produce results that are directly comparable and reproducible.
> - **Experiment Conduction Protocol.** Another key contribution is the establishment of a consistent experimental protocol. The lack of **standardization in experimental setups**, such as parameter settings, initialization procedures, and convergence criteria, has been a major barrier to reproducibility in IGL studies. Our work provides a comprehensive protocol to standardize these aspects, enabling more reliable and comparable results across different studies.
> - **Performance Evaluation Standard.** We also address the inconsistency in performance **evaluation metrics** across studies. While **effectiveness** is often the primary focus, we argue that understanding the **efficiency and complexity** of algorithms is equally important. Our paper introduces a more comprehensive evaluation standard that includes these dimensions, ensuring that IGL methods can be better assessed in terms of both their accuracy and practical feasibility.
>
> Our work lays the foundation for future studies by standardizing **how datasets are prepared and how experiments should be conducted, ensuring that subsequent research can build on a solid, reproducible code base.**
>
> We hope this clarifies the scope and importance of our contributions, and we will make sure to emphasize these points more clearly in the revised manuscript. Thank you again for your feedback!

---

> ### Author Response · Authors · 2024-11-18
> **Response to Question #1**
>
> > **Question #1:** Can the authors elaborate on the specific challenges encountered in standardizing data splits and evaluation protocols across such a diverse set of IGL methods?
>
> **A:** Thank you for your insightful question! Standardizing data splits and evaluation protocols for diverse IGL methods is super challenging due to several factors:
>
> - **Data Diversity.** Graph datasets used in IGL research come from varied domains, each with unique characteristics such as levels of homophily, heterophily, and differences in graph sizes. This made it difficult to establish a splitting strategy that would be unbiased and representative for all datasets. Splits that work well for datasets with strong homophily may be ineffective for those with heterophily, requiring careful adjustments to maintain fairness while preserving each dataset’s properties.
> - **Complex Imbalance Types.** The imbalances in graph data are multifaceted, including class-imbalance and topology-imbalance. Addressing these required us to design evaluation protocols that could assess methods effectively across different imbalance scenarios, such as local and global topology issues. This increased the complexity, as we needed metrics that capture the nuances of both types of imbalance, ensuring a comprehensive evaluation of algorithm performance.
> - **Consistency and Fairness.** Previous IGL studies often varied in terms of data preprocessing, hyperparameter tuning, and performance metrics, making results hard to compare. We tackled this by normalizing these elements across our benchmark, using uniform metrics like Accuracy, Balanced Accuracy, Macro-F1, and AUC-ROC. This involved extensive experimentation to align the evaluation processes and ensure fair and reproducible comparisons across all included methods.
>
> Due to these challenges, there has been no unified, comprehensive benchmark for IGL so far to the best of our knowledge. Each of these challenges has significantly hindered progress toward developing such a standardized benchmarking framework. Our efforts were necessary to provide a robust, reproducible framework that fairly evaluates the diverse approaches to imbalanced graph learning, while promoting consistency and comparability within the research community.

---

> ### Author Response · Authors · 2024-11-18
> **Response to Question #2**
>
> > **Question #2:** How does IGL-Bench handle algorithm hyperparameter tuning, particularly for methods that might be sensitive to different types of imbalance?
>
> **A:** Thanks for your questions. Following the standard practice in most related works, we conducted comprehensive hyperparameter searches for all algorithms within IGL-Bench across various imbalance settings. This ensures the results reflect the best possible performance of each algorithm. Specifically, IGL-Bench adopts a comprehensive and systematic approach:
>
> - **Integrated Efficient Tuning Tools.** Our package architecture includes the **Weights & Biases (wandb) framework ([Figure E.1](https://openreview.net/pdf?id=uTqnyF0JNR#page=56), top right in “Package Utils”)**, which provides comprehensive training management. It enables efficient logging of results and supports multiple hyperparameter search strategies, such as Bayesian optimization and grid search. This integration allows IGL-Bench to efficiently perform hyperparameter tuning for benchmarked algorithms, while also offering flexibility for users to tune their custom algorithms and datasets, **with only several lines of codes and configurations.**
> - **Targeted Tuning for Different Imbalance Types.** IGL-Bench carefully adjusts hyperparameters based on the specific imbalance type an algorithm is sensitive to. For **class imbalance**, we tune parameters that **affect class distribution**, such as class weights, sampling strategies, and loss function configurations, based on their **original settings**. For **topology imbalance**, we tune parameters that **influence graph structure**, such as neighborhood size and the number of graph convolution layers, to mitigate issues like over-squashing and under-reaching, while adhering to their **original settings**. **We list all the hyperparameter search space for all IGL algorithms in [Appendix C.3](https://openreview.net/pdf?id=uTqnyF0JNR#page=31).**
> - **Scenario-Specific Optimization Workflow.** To ensure optimal performance across various imbalance conditions, we evaluate and fine-tune each algorithm under a range of imbalance ratios, from mild to extreme imbalance. This allows us to observe and account for the performance variability of algorithms under different conditions, thereby understanding their robustness more thoroughly.
> - **Automation and Scalability.** The automated hyperparameter tuning via wandb ensures consistent and fair comparisons across all algorithms, while also being scalable for future extensions. This design allows researchers to reproduce our results or customize the tuning process for their specific needs.
>
> By implementing these strategies, IGL-Bench effectively manages algorithm sensitivities to different types of imbalance, providing a reliable and adaptable evaluation framework.

---

> ### Author Response · Authors · 2024-11-18
> **Response to Question #3**
>
> > **Question #3:** Are there future plans to study the scalability within IGL-Bench, e.g., for larger, more complex graph datasets?
>
> **A:** We thank the reviewer for highlighting the important aspect of scalability in IGL-Bench. We recognize that scalability is a crucial factor, especially as real-world applications often involve larger and more complex graph datasets.
>
> **We have already integrate large-scale** graph datasets ogbn-arXiv and ogbg-molhiv for **node-level and graph-level tasks** respectively. However, we observed the limitations and low scalability of existing IGL algorithms when dealing with these large-scale and complex graph data.
>
> We found several recent works proposed scalable IGL algorithms to handle such graph data [1, 2]. Addressing this, **we indeed have plans for future extensions of IGL-Bench aimed at enhancing its ability to evaluate and benchmark algorithms on larger-scale graphs.**
>
> - **Future Integration of Larger-Scale Datasets.** We are actively exploring the integration of more extensive and complex graph datasets that reflect real-world scenarios, such as the Open Graph Benchmark [3]. These datasets will include not only larger node and edge counts but also more intricate structures, such as **heterogeneous and dynamic graphs**. This expansion will help researchers understand how well current algorithms scale and identify areas that need further optimization.
> - **Efficiency and Scalability Analysis.** As outlined in our paper, we have conducted preliminary experiments to evaluate the time and space efficiency of algorithms. Moving forward, we plan to extend this analysis by benchmarking algorithms on ultra-large-scale datasets. We also aim to explore performance trade-offs between effectiveness and computational cost, providing deeper insights into the practical applicability of these algorithms.
> - **Development of Scalability-Focused Metrics.** In addition to expanding our dataset library, we plan to introduce metrics and evaluation strategies specifically tailored for scalability assessment. This includes measuring algorithm performance under increasing data volume, memory footprint analysis, and runtime complexity under various graph settings.
> - **Leveraging Graph Foundation Models.** We are also considering incorporating graph foundation models [4-6], which are designed for efficient representation learning on large graphs. This will further enhance IGL-Bench’s capability to assess and compare state-of-the-art scalable graph learning algorithms.
>
> In summary, **we are committed to extending IGL-Bench’s scalability evaluation capabilities**, ensuring that it remains a comprehensive and practical resource for the research community as the field of imbalanced graph learning continues to evolve.

---

> ### Author Response · Authors · 2024-11-18
> **Conclusion**
>
> Thank you again for your time and valuable insights, and hope our responses can address all your concerns! We are excited about the impact IGL-Bench may have on the community and are grateful for your support in advancing this work.
>
> **We would be more encouraging if you could kindly consider raising your ratings!**
>
> ---
>
> **References:**
>
> **[1]** Overcoming graph topology imbalance for inductive and scalable semi-supervised learning. Dornaika F, et al. Applied Soft Computing, 2024.
>
> **[2]** SHINE: A Scalable Heterogeneous Inductive Graph Neural Network for Large Imbalanced Datasets. Van Belle R, et al. IEEE Transactions on Knowledge and Data Engineering, 2024.
>
> **[3]** Open graph benchmark: Datasets for machine learning on graphs. Hu W, et al. Advances in neural information processing systems, 2020.
>
> **[4]** Towards graph foundation models: A survey and beyond. Liu J, et al. arXiv, 2023.
>
> **[5]** Graph foundation models. Mao H, et al. arXiv, 2024.
>
> **[6]** Opengraph: Towards open graph foundation models. Xia L, et al. arXiv, 2024.

---

> ### Author Response · Authors · 2024-11-22
> **Friendly Reminder**
>
> **Dear Reviewer 6GyC,**
>
> This is a friendly reminder that we greatly value your input on our rebuttal. Your comments have been extremely helpful in improving our paper.
>
> With less than one week (ends on **Nov. 26th** AoE) remaining before the conclusion of the discussion phase, we wish to extend a respectful request for your feedback about our rebuttal.
>
> We highly value your insights and are eagerly awaiting your revised evaluation. Should you find our rebuttal to be informative and beneficial, we would deeply appreciate your recognition. Additionally, if you have any further questions or need additional clarification, please feel free to contact us. We are fully dedicated to engaging and providing support during this important discussion phase.
>
> Thanks for your time again.
>
> Best regards,
>
> **Authors of Submission #6858.**

---

> ### Comment · Reviewer_6GyC · 2024-11-25
> **Response to authors**
>
> Thanks for your detailed responses, which address most of my points and help me better evaluate the contributions of the paper. I believe including the additional materials, e.g., visualization and scalability analysis, into the final version could help enhance the quality of the paper.
>
> Overall, I think this is a good work, providing a comprehensive benchmark for a new setting. I would like to increase my rating and favor the paper to be accepted.

---

> ### Author Response · Authors · 2024-11-25
>
> Thank you very much for your positive feedback and for taking the time to provide detailed comments on our work. We are glad that our responses addressed your concerns and helped clarify the contributions of our paper. We greatly appreciate your suggestion to include additional clarifications, such as visualizations and scalability analysis, to further enhance the paper's quality. We will make it more thorough and easy-to-follow in the camera-ready version!

---

### Author Response · Authors · 2024-11-18
**Global Response**

We sincerely thank all reviewers for their thoughtful and constructive feedback on our paper. We are delighted that the reviewers recognize the significance of our benchmark to the Imbalanced Graph Learning (IGL) research community. Specifically, reviewers acknowledged:

- **Pioneering Benchmark.** The introduction of IGL-Bench as the first comprehensive benchmark for imbalanced graph learning (IGL)  fills a critical gap in the field. Establishing such a benchmark is essential for tracking progress within the research community. (Reviewer `6GyC`, Reviewer `t4gw`)
- **Thorough  and High-Quality Experimental Design.** The reviewers believe that the experimental design is thorough and high-quality, validating the effectiveness of the benchmark. The research questions and the authors’ insights posed are highly relevant and of interest to the research community, offering a comprehensive evaluation of IGL methods. (Reviewer `6GyC`, Reviewer `t4gw`, Reviewer `mZgm`)
- **Reproducibility and Accessibility.** The provision of an open-source package and  a standardized  platform facilitate reproducible evaluation,  encouraging a more consistent evaluation process for future work. (Reviewer `6GyC`, Reviewer `t4gw`, Reviewer `mZgm`)
- **Clarity and Presentation.** The reviewers believe that the paper is well-written, well-organized, and easy to follow, making it suitable for both general and specialized audiences. (Reviewer `6GyC`, Reviewer `t4gw`, Reviewer `mZgm`)

We have provided detailed responses to each reviewer’s comments. Additional experimental results and further clarifications are also included in the **[revised PDF (Appendix G)](https://openreview.net/pdf?id=uTqnyF0JNR#page=59)**.

Once again, we are grateful for the reviewers’ valuable insights, which have greatly helped us improve our work. We have made every effort to address all concerns thoroughly. Thank you for considering our rebuttal, and we kindly request that you reconsider the ratings of our paper, if possible.

---

### Comment · Area_Chair_jUmt · 2024-11-23

Dear Reviewers,

The authors have uploaded their rebuttal. Please take this opportunity to discuss any concerns you may have with the authors.

AC

---

### Meta-Review · Area_Chair_jUmt · 2024-12-20

**Metareview:**

This paper introduces IGL-Bench, a benchmark for Imbalanced Graph Learning (IGL) that evaluates 24 algorithms across 17 datasets, addressing both class and topology imbalances in node- and graph-level tasks. The benchmark provides a unified data processing pipeline, evaluation protocols, and metrics, allowing researchers to assess the effectiveness, robustness, and efficiency of IGL methods. Additionally, the paper offers an open-source codebase to promote reproducibility and further research in the field.

The paper addresses a significant gap in graph learning by introducing a comprehensive benchmark for Imbalanced Graph Learning (IGL), essential for standardized evaluation in this rapidly evolving field. Through thorough experiments and analysis, it provides valuable insights into the strengths and limitations of current IGL algorithms, laying a solid foundation for future research. The open-source codebase and standardized evaluation framework enhance reproducibility, ensuring that the benchmark remains accessible and impactful for the broader research community. As a result, the reviewers unanimously voted to accept this paper.

**Additional Comments On Reviewer Discussion:**

The authors have made significant efforts in addressing the reviewers’ concerns, with detailed responses and clarifications to improve the paper. They have provided additional experimental results and insights, particularly in areas where the reviewers raised issues, such as the clarity of visualizations, the inclusion of new methods, and the scalability of the benchmark. The authors also emphasized the value of their work in filling a critical gap in the Imbalanced Graph Learning (IGL) field, offering a comprehensive and reproducible evaluation platform. Overall, the authors have strengthened the paper by thoroughly addressing the feedback, enhancing both its clarity and robustness.

---

### Decision · Program_Chairs · 2025-01-22

Accept (Spotlight)